

# The edge of chaos:
# Quantum field theory and deep neural networks

**Kevin T. Grosvenor[1,2] and Ro Jefferson[3]**

**1** Max-Planck-Institut für Physik komplexer Systeme and Würzburg-Dresden,
Cluster of Excellence ct.qmat, Nöthnitzer Str. 38, 01187 Dresden, Germany
**2** Instituut-Lorentz, Universiteit Leiden, P.O. Box 9506, 2300 RA Leiden,
The Netherlands
**3** Nordita KTH Royal Institute of Technology and Stockholm University,
Hannes Alfvéns väg 12, 106 91 Stockholm, Sweden

## Abstract

We explicitly construct the quantum field theory corresponding to a general class of deep neural networks encompassing both recurrent and feedforward architectures. We first consider the mean-field theory (MFT) obtained as the leading saddlepoint in the action, and derive the condition for criticality via the largest Lyapunov exponent. We then compute the loop corrections to the correlation function in a perturbative expansion in the ratio of depth $T$ to width $N$, and find a precise analogy with the well-studied $O(N)$ vector model, in which the variance of the weight initializations plays the role of the 't Hooft coupling. In particular, we compute both the $\mathcal{O}(1)$ corrections quantifying fluctuations from typicality in the ensemble of networks, and the subleading $\mathcal{O}(T/N)$ corrections due to finite-width effects. These provide corrections to the correlation length that controls the depth to which information can propagate through the network, and thereby sets the scale at which such networks are trainable by gradient descent. Our analysis provides a first-principles approach to the rapidly emerging NN-QFT correspondence, and opens several interesting avenues to the study of criticality in deep neural networks.

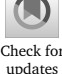

# 1 Introduction

The remarkable empirical success of deep neural networks in domains ranging from image classification to natural language processing has far outpaced our theoretical understanding. In practice, these networks are generally treated as black boxes, with state-of-the-art performance relying primarily on heuristics and trial-and-error rather than on any fundamental theory [1, 2]. Indeed, this state of affairs has even led some experts in the field to compare modern machine learning to alchemy [3]. As has been pointed out in, e.g., [4], top-down insights from fields such as cognitive science may be fruitfully complemented by bottom-up approaches from theoretical physics towards uncovering the fundamental principles governing learning and intelligence in minds and machines.

Recently, physicists have begun exploring the parallels between deep neural networks and quantum field theory, leading to a number of interesting works on what is sometimes called the NN-QFT correspondence [2, 5–9].[1] One key aspect of this approach is the fact that many modern architectures admit a Gaussian process limit: as the number of neurons (per layer, i.e., the width) $N \to \infty$, the network is described by a Gaussian distribution, and hence may be modelled as a free (non-interacting) quantum field theory. This is essentially a consequence

---

[1]We use the phrase "quantum field theory" to make manifest the precise analogies with well-studied techniques in QFT that we explore below, but we emphasize that there is nothing intrinsically "quantum" about the networks under consideration. It would perhaps be more accurate to say "statistical field theory," or one may simply think of this as QFT in Euclidean signature, cf. sec. 5. We also thank Dan Roberts and Sho Yaida for pointing out that previous works that we have included under the "NN-QFT" umbrella did not involve a *bona fide* field theory in the sense of a well-defined continuum limit, nor a direct correspondence between elements of a neural network and elements of a QFT (see appendix A), but we have here used the term broadly as in [7] to refer to the general philosophy of establishing parallels between neural networks and QFT, with the aim of using the latter to understand the former. See also [10] for an orthogonal approach from quantum mechanics.

of the central limit theorem (CLT); see for example [7] for a brief summary and plentiful references of the Gaussian process limit in this context, including the original thesis [11].

While the infinite-width limit provides an analytically tractable approximation that has led to important progress (see, e.g., [12–15]), it fails to capture crucial aspects of real-world networks which must of necessity be of finite width. For example, the lack of interactions – i.e., intralayer correlations – in the Gaussian limit implies that representations in these idealized networks do not evolve during gradient-based learning [2, 16]. (Note that this does not contradict the well-known universal approximation theorem for infinite-width networks: the latter merely states that there exists some network (that is, a particular set of weights and biases) that approximates a given function, but says nothing about the learning dynamics. The statement that the representation (specifically, the neural tangent kernel) does not evolve at infinite width means that if the network does not happen to start in the correct state, it will never evolve to it; see sections 6.3.3, 10.1.2, or chapter 11 of [2] for details.) Additionally, for networks exhibiting an order-to-chaos phase transition (see below), the location of the critical point predicted by the central limit theorem differs from empirical observations [17, 18]. Understanding networks away from the infinite-width limit is therefore of central importance for any theory of deep learning with practical applications.

Conveniently, the art of carefully backing away from the $N \to \infty$ limit has been perfected by physicists in the form of quantum field theory. The machinery of Feynman diagrams, which we shall later employ, provides a compact and efficient means of computing perturbative contributions from interactions. In the most basic terms, the idea is to solve a complicated (non-Gaussian) theory by perturbing about the free (Gaussian) theory in terms of some small parameter. This allows one to effectively turn on interactions – i.e., finite-width effects – in a controlled manner, which in statistical language corresponds to the inclusion of higher cumulants. In finite-width networks, such cumulants appear in subsequent layers in a manner precisely analogous to what one observes along the renormalization group (RG) flow, due to the marginalization over neurons in previous layers (i.e., tracing-out hidden degrees of freedom) [2, 19].

Another important aspect that many neural networks share with models well-studied in physics is the phenomenon of criticality, by which we mean a continuous phase transition separating an ordered and disordered or chaotic phase. The idea that the *edge of chaos* holds key advantages for computation and learning dates back to at least the classic paper by Langton [20], and was further developed in the influential work by Bertschinger, Legenstein, and Natschläger [21, 22]. It has since appeared in many fields ranging from theoretical neuroscience and neurophysiology [23–25], to biological and complex systems [26, 27], and was also explored in an early model of bioplausible neural networks in [28]. More recently, [17, 18, 29–32] demonstrated that networks initialized at criticality are trainable to far greater depths than those lying further into either phase. To see this, one identifies a correlation length that sets the scale at which correlations between local degrees of freedom – e.g., the activations of two different neurons, or the spins of two different magnetic dipoles – decay with separation or depth through the network. Intuitively, both the ordered and chaotic phases are bad for learning, since correlations about the input data – i.e., the structure one is attempting to learn – are energetically damped or washed-out by noise, respectively. However, a critical point is characterized by a divergent correlation length, which allows the persistence of structured information on all scales. We refer the interested reader to [1, 33, 34] for topical reviews of this phenomenon, or to [35] for a beautiful visual demonstration in the case of the 2d Ising model.

In this work, using techniques from statistical field theory [36, 37], we *explicitly construct* the quantum field theory corresponding to a general class of deep neural networks encompassing both recurrent (RNN) and feedforward (multilayer perceptron, MLP) architectures, in order to leverage well-known tools in perturbative QFT to compute the finite-width correc-

tions to the correlation length. An original motivation was to determine whether these $1/N$ effects would shift the location of the critical point, as one expects on both theoretical and empirical grounds. Perhaps surprisingly, while the effective correlation length does appear to receive corrections from all orders in the perturbative expansion, the location of the critical point itself appears unchanged. This conclusion should however be regarded with skepticism, since the theory breaks down at the critical point itself. As is generally the case in QFT, the perturbative analysis is only valid at weak coupling, which we shall see corresponds to a constraint on the variance of the weight initializations, $\sigma_w^2$. Consequently, the resulting theory is only strictly valid in a subregion of phase space that shrinks as we approach the edge of chaos from the ordered phase. Nonetheless, the theory exhibits a remarkable similarity with the well-known $O(N)$ vector model [38–44] that we elaborate on in more detail below, and offers a new perspective on the nascent NN-QFT correspondence that may be fruitfully developed further.

**Relation to other work**

As this work was nearing completion, the excellent book [2] appeared, which shares a similar philosophy of applying ideas from physics to understand the structure and properties of deep neural networks, and in particular also considers perturbative corrections in depth/width; see also the previous work [5]. Relative to our approach, the main difference is that they did not develop a correspondence with any QFT; rather, the finite-width corrections in their case are seen to arise by starting with the Gaussian distribution of preactivations in the first layer, and fixing the couplings in subsequent layers to track the appearance of effective interactions, i.e., higher cumulants that appear as a result of marginalizing over hidden degrees of freedom, in the analogy with RG mentioned above. In contrast, we are concerned with the bulk[2] behavior of the network after the layer-to-layer Hamiltonian has reached a steady state (which must occur some finite depth after the input layer), and the exact form of the interactions are included in the full action of the QFT by construction. A central feature shared by our analyses is that the ratio of depth to width, $T/N$, is the perturbative expansion parameter; accordingly, we likewise work in the limit $T, N \to \infty$ with $T/N \ll 1$ fixed (though in our case this quantity, specifically $T$, is dimensionful, and hence one must include an order-1 constant, as will be discussed in detail below). As observed in [2], this is the regime of most deep neural networks used in practice. We view our approaches as complementary, and warmly recommend [2] for readers interested in a pedagogical treatment of many ideas at this interface of theoretical physics and machine learning. We note that finite-width corrections to the feature kernel were also considered recently in [45].

Another RG-inspired approach appeared in the earlier work [7], which was further developed in two more papers [8,9] that appeared while this project was underway (see also [46]). As alluded above, [7] explores the relation between QFTs and the Gaussian process (i.e., $N \to \infty$) limit of neural networks in detail. The basic idea from Wilsonian effective field theory is to write down the most general action consistent with the assumed symmetries and properties of the model (e.g., translation symmetry, locality), and fit the associated couplings (that is, coefficients of the interaction terms) by experiment. In particular, they posit an initially Gaussian action, and show that the deviations from the infinite-width limit can be accurately modelled by the addition of a quartic interaction term, whose contribution is suppressed by $1/N$. The irrelevance of higher (e.g., six-point) interactions can be understood from an RG perspective, which was developed more thoroughly in [8]. These authors also introduced the phrase "neural network phenomenology" to emphasize the fact that the form of the action in

---

[2]That is, we work away from the input/output layers. At a technical level, we shall assume time- (i.e., layer-) translation invariance in order to facilitate obtaining closed-form expressions for the correlation functions.

all of the above approaches is posited on general grounds, with couplings determined either by experiment or by tracking effective interactions. In contrast, the key difference in the present work is again that we explicitly construct the action of the field theory from the (stochastic) differential equation governing the network dynamics, so that the precise form of the interactions follows directly. It would be very interesting to understand the relation between these approaches in more detail, and in particular whether applying the RG techniques in [7,8] to the type of bottom-up model we develop here would lead to further insights into criticality in deep neural networks.

Additionally, as we will employ Feynman diagrams to compute interactions in perturbation theory, let us mention that these have previously appeared in the neural network literature in [6], and are also reviewed in [7]. The present case differs in that the diagrammatic expansion closely resembles that in the $O(N)$ vector model, where the dominant correction stems from an infinite series of so-called cactus diagrams (see subsec. 4.3.1) which we develop in a similar manner to the double-line notation originally introduced by 't Hooft [38] (see chapter 8 of [40] for a pedagogical treatment). Interestingly, we also find that the weight variance $\sigma_w^2$ plays the role of the 't Hooft coupling $\lambda := g_{YM}^2 N$ in Yang-Mills theory, where $g_{YM}$ is the (Yang-Mills) coupling constant that appears in the action. There, one considers the $N \to \infty$ limit with fixed 't Hooft coupling $\lambda$, the size of which controls the celebrated AdS/CFT correspondence [47,48]. Perturbative quantum field theory of the kind we explore here is obtained in the weak-coupling limit ($\lambda < 1$), whereas strong coupling ($\lambda \gtrsim 1$) gives the low-energy supergravity limit of string theory on AdS spacetime. As alluded above, weak coupling is required in order for the infinite series of diagrams at each order in perturbation theory to converge, which imposes a limitation on the region of phase space in which the theory is strictly valid. We regard the analogy with $O(N)$ theory at weak coupling as an interesting potential avenue for future research on the NN-QFT correspondence.

Turning to the titular edge of chaos, we are inspired by the aforementioned works [17, 18, 29–32] examining criticality in various deep network architectures. However, while many of these papers used the phrase "mean-field theory", they did not actually rely on any MFT analysis: as mentioned above, Gaussianity arises simply as a consequence of the central limit theorem (CLT). In fact, one important lesson of our analysis is that the correlation functions obtained from the CLT do *not* agree with the MFT for random networks, even in the infinite-width limit. For the aforementioned works, the distinction is inconsequential, since both approximations appear to yield the same condition for criticality,[3] but it is essential to the development of a rigorous NN-QFT correspondence. Prior to considering finite-width corrections in sec. 4, we shall first show in sec. 3 that we recover the criticality condition in the previous works from a true MFT treatment, which we will subsequently reaffirm as the tree-level contribution in perturbative quantum field theory in sec. 4. For readers unfamiliar with statistical field theory, we emphasize that the use of MFT in this context is itself not a novel development: the basic approach we employ was pioneered in the classic paper [37], and we have benefited greatly from the pedagogical review [36].[4] Nonetheless, while the basic starting point of our work is not fundamentally new, we have considerably developed the perturbative analysis, and applied a variety of old ideas in novel ways.

The remainder of this paper is organized as follows: in sec. 2, we construct the QFT corresponding to the general class of networks under consideration using standard techniques

---

[3]For this reason, it is unclear whether the CLT result for the critical condition in [17] and related works effectively includes the $\mathcal{O}(1)$ corrections we compute in sec. 4, since we expect that the latter do not alter the Gaussianity of the theory; see sec. 5 for further discussion.

[4]For other applications of MFT to artificial intelligence, see for example [49] and references therein.

in statistical field theory [36]. Specifically, we will obtain the partition function[5] describing self-averaging random networks, and derive equations for the correlation functions in the MFT approximation. In sec. 3, we will construct the double-copy MFT system in order to examine the growth of correlations between different samples of networks from the ensemble. The edge of chaos is then determined by the point at which the largest Lyapunov exponent becomes positive, which yields precisely the criticality condition discussed above. In sec. 4, we go beyond the MFT regime, and consider the perturbative expansion of the full (single-copy) QFT. We first explicitly compute the tree-level propagator – that is, the two-point correlation function in the absence of quantum corrections[6] – identify the correlation length, and show that the condition for criticality precisely agrees with the previous MFT result, as expected. We then compute the first two correction terms to the two-point correlation function, which contribute at $\mathcal{O}(1)$ and $\mathcal{O}(T/N)$, respectively, for both linear (subsec. 4.3) and non-linear (subsec. 4.4) models.[7] Here the use of Feynman diagrams vastly simplifies the computations: we will develop an elegant diagrammatic recursion relation, which enables us to sum the infinite series of diagrams at each order to yield a finite contribution. With the loop-corrected propagator in hand, we are then able to identify the finite-width corrections to the effective correlation length in the weak-coupling regime. We close in sec. 5 with some discussion and comments on future directions. For convenience, we have summarized the elements of the nascent *NN-QFT dictionary* in appendix A. Appendix B contains some identities used in evaluating the Feynman diagrams encountered in sec. 4.4, while appendix C contains explicit expressions for the loop-corrected correlation length for nonlinear models which were deemed too lengthy to include in the main text.

## 2 The statistical field theory approach

Rather than writing down a general action and fixing the couplings to match empirical results, we will instead employ a bottom-up approach, in which we start with the differential equation describing an RNN – which includes vanilla feedforward neural networks (i.e., multilayer perceptrons, MLPs) as a special case – and explicitly construct the corresponding quantum (statistical) field theory. As mentioned in the introduction, the approach we follow has a long and diverse history, and we shall draw heavily from the pedagogical review [36]. Indeed, the main technical differences are the inclusion of the data-dependent term, and the explicit constant $\gamma$ in (2). That said, the presentation in [36] focuses on many other topics that are not relevant to our goal, and we have distilled the full derivation in order to make the paper self-contained, as well as to establish our notation.

The basic strategy is as follows: starting with the stochastic differential equation describing the update rule for the network, the probability for a particular sequence of network states is obtained by marginalizing over the stochasticity. We then introduce an auxiliary field to impose the constraint that the network continue to satisfy the update rule while allowing the fields to fluctuate (i.e., to go off-shell). Upon adding source terms, we obtain the partition function in a form familiar to physicists, from which we can in principle compute any quantity

---

[5]For our machine learning readers, this is essentially the moment-generating function, with which we can – in principle – compute any observable we wish to know; see [50] for a pedagogical exploration of these relationships.

[6]Again, to preempt any sensationalist headlines that "quantum physics explains deep learning", we remind the reader that there is nothing fundamentally quantum at work: the language is meant to be evocative of the loop corrections with which physicists are intimately familiar, but these are purely classical corrections due to the cumulants associated with turning on interactions in a controlled manner.

[7]Our analysis is phrased at the more general level of recursive neural networks (RNNs), which includes feedforward networks as a simple case. In the former, the total time $T$ serves as an infra-red (IR) cutoff, while for feedforward networks this role is played by the total length $L$, as in [2].

of interest, e.g., correlation functions. In subsec. 2.2, we make the theory explicit by introducing an ansatz for the trainable parameters, which are similarly elevated to fluctuating field variables. The MFT in subsec. 2.3 is then simply the leading-order saddlepoint approximation of the resulting QFT. Various technical assumptions introduced along the way are summarized in appendix A.1. Throughout this work, we have endeavored to make each step in the construction as transparently explicit as possible, in the hopes that the reader will find the length more pedagogical than daunting.

## 2.1  Constructing the partition function

Our starting point is the continuous-time formulation of a recurrent neural network (RNN) as a stochastic differential equation (SDE) [51]

$$dh = f(h, x)\, dt + g(h, x)\, dS, \tag{1}$$

where $h \in \mathbb{R}^{N_h}$ is a vector of hidden states, $x \in \mathbb{R}^{N_x}$ is a vector of inputs, and $dS$ is a $N_r$-dimensional vector of stochastic increments. This expression represents the most general class of SDE, and hence provides a powerful starting point for our approach. The stochasticity or noise is controlled by the diffusion coefficient $g \in \mathbb{R}^{N_h \times N_r}$, while the deterministic update is controlled by the drift coefficient $f \in \mathbb{R}^{N_h}$. For the moment we shall keep the former arbitrary, while for the latter we shall consider

$$f(h, x) = (A - \gamma)h + Bx + W\phi(h) + U\varphi(x) + b, \tag{2}$$

which is consistent with the standard convention used in the literature (e.g., [29, 36]), where $A, B, W \in \mathbb{R}^{N_h \times N_h}$, $U \in \mathbb{R}^{N_h \times N_x}$, $\gamma > 0$ is some fixed $\mathcal{O}(1)$ constant, and $b \in \mathbb{R}^{N_h}$ are trainable parameters, and $\phi$ is some nonlinear activation function. Here $W$ is the matrix of weights that connect each neuron to all of those in the next layer/time-step, and $b$ is the vector of biases. $U$ can be thought of as analogous to $W$, and controls the weight or importance of the input data $x$, while $A$ and $B$ are additional parameters included for generality. The constant $\gamma$ is needed to absorb the dimensions of $dt$ (that is, the total time $T$ will be endowed with units of $1/\gamma$ below). However, it has a more practical interpretation as determining the specific model under study. For example, to recover a multilayer perceptron with preactivations at layer $\ell$ defined as $h^\ell = W_{ij}^\ell \phi(h_j^{\ell-1}) + b^\ell$, we set $\gamma = 1$ and $A = B = U = [0]$, and view the discrete timesteps as subsequent layers $\ell \in [0, T-1]$.[8] We shall eventually specify to $\phi = \tanh$ as our activation function for concreteness, but the analysis may be performed for any suitable activation function that satisfies some basic properties, as discussed in appendix A.1. For simplicity, we shall henceforth take $N := N_h = N_x = N_r$.[9]

The SDE (1) is interpreted under the Itô discretization convention as

$$h_t - h_{t-1} = f(h_{t-1}, x_{t-1})\eta + g(h_{t-1}, x_{t-1})\xi_t, \tag{3}$$

where $\eta := dt$ and $\xi_t := \frac{dS_t}{dt}\, dt$. Note that while we have written $g$ as a function of the state and data for generality, we will henceforth specify to the case where $g$ is independent of either, and is instead some external function of time, i.e., $g_{t-1}$. This is physically more natural insofar as the stochasticity should not depend on the current state of the system, and further ensures

---

[8]As we shall see, $A$ and $B$ play relatively little role, and in fact we shall set them to zero for simplicity in the perturbative analysis in sec. 4, while the constant $\gamma$ is of key importance. Additionally, note that the input/output layers are not strictly part of this analysis, as explained in the introduction (i.e., one may think of the true length of the network as at least $T + 2$).

[9]It is not necessary for the network to have constant width, but the width of each layer must be sufficiently large, and go to infinity at the same rate.

that it does not cause identically-prepared copies of the system in the ostensibly ordered phase to diverge, cf. sec. 3.

The field-theoretic approach begins by constructing a moment generating functional for the state of the system, i.e., the partition function. Proceeding with (1), the first step is to observe that if the noise is drawn independently for each timestep, then the probability of a particular path $h(t)$ (by which we mean the sequence of states $h_0, \dots, h_T$) may be written as[10]

$$p(h(t)|a) = \int \prod_{t=0}^{T} d\xi_t \, \rho(\xi_t) \, \delta(h_t - y_t(\xi_t, h_{t-1})), \tag{4}$$

where $\rho(\xi_t)$ is the probability density function for the noise increment $\xi_t$ (e.g., a Gaussian) with normalization $\prod_{t=0}^{T} \int d\xi_t \, \rho(\xi_t) = 1$, and $y_t(\xi_t, h_{t-1})$ is the solution to (3),[11]

$$y_t(\xi_t, h_{t-1}) = h_{t-1} + f(h_{t-1}, x_{t-1})\eta + g_{t-1}\xi_t + a\delta_{t,0}, \tag{5}$$

where we have included the possibility of specifying a non-zero initial condition in the form of the Kronecker delta term. The Dirac delta in (4) thus imposes that the state satisfies the given SDE (3) at time $t$, given the noise $x_t$ and the solution at the previous timestep $h_{t-1}$ (i.e., the process is Markovian).

Now, given the integral expression for the Dirac delta function,

$$\delta(x) = \int_{-\infty}^{\infty} \frac{dk}{2\pi} e^{ikx} = \int_{-i\infty}^{i\infty} \frac{d\tilde{z}}{2\pi i} e^{\tilde{z}x}, \tag{6}$$

where $\tilde{z} = ik \in \mathbb{C}$, we can express (4) as an integral over the auxiliary variable $\tilde{z}$ (also called the *response field*, for reasons which will become apparent below). Introducing the shorthand $f_{t-1} := f(h_{t-1}, x_{t-1})$, we have

$$
\begin{aligned}
p(h(t)|a) &= \prod_{t=0}^{T} \int d\xi_t \, \rho(\xi_t) \int_{-i\infty}^{i\infty} \frac{d\tilde{z}_t}{2\pi i} e^{\tilde{z}_t(h_t - y_t(\xi_t, h_{t-1}))} \\
&= \prod_{t=0}^{T} \int d\xi_t \, \rho(\xi_t) \int_{-i\infty}^{i\infty} \frac{d\tilde{z}_t}{2\pi i} \exp\left[\tilde{z}_t\left(h_t - h_{t-1} - f_{t-1}\eta - a\delta_{t,0}\right) - \tilde{z}_t g_{t-1}\xi_t\right] \\
&= \prod_{t=0}^{T} \int_{-i\infty}^{i\infty} \frac{d\tilde{z}_t}{2\pi i} \exp\left[\tilde{z}_t\left(h_t - h_{t-1} - f_{t-1}\eta - a\delta_{t,0}\right) + K_\xi(-\tilde{z}_t g_{t-1})\right],
\end{aligned}
\tag{7}
$$

where $K_\xi(-\tilde{z}_t g_{t-1}) = \ln\langle e^{-\tilde{z}_t g_{t-1}\xi}\rangle_\xi = \ln \int d\xi_t \, \rho(\xi_t) e^{-\tilde{z}_t g_{t-1}\xi_t}$ is the cumulant generating function of $\xi_t$. Importantly, note that the effects of noise are now entirely encapsulated by the associated cumulants! The moment generating or partition function is then obtained by adding

---

[10]Strictly speaking, we should view this as a conditional distribution on a particular sequence of inputs $x(t)$, since this will be treated as external data below.

[11]Note that the existence of an explicit solution, i.e., $y_t = h_t$, is a beneficial feature of the Itô convention. If we had instead selected the Stratonovich convention, $f(x) = f\left(\frac{x_t + x_{t-1}}{2}\right)$, we would have only an implicit solution, since $h_t$ would appear on both the left- and right-hand sides.

source terms $j_t h_t \eta$ to the action and integrating over all paths $h$, i.e.,

$$
\begin{aligned}
Z[j] &= \Big\langle \prod_{t=0}^{T} e^{j_t h_t \eta} \Big\rangle_h = \prod_{t=0}^{T} \int dh_t \, p(h(t)|a) \, e^{j_t h_t \eta} \\
&= \prod_{t=0}^{T} \int dh_t \int_{-i\infty}^{i\infty} \frac{d\tilde{z}_t}{2\pi i} \exp\Big[ \tilde{z}_t \big(h_t - h_{t-1} - f_{t-1}\eta - a\delta_{t,0}\big) + j_t h_t \eta + K_\xi(-\tilde{z}_t g_{t-1}) \Big] \\
&= \prod_{t=0}^{T} \Big\{ \int dh_t \int_{-i\infty}^{i\infty} \frac{d\tilde{z}_t}{2\pi i} \Big\} \times \\
&\quad \exp \sum_{t=0}^{T} \Big[ \tilde{z}_t \big(h_t - h_{t-1} - f_{t-1}\eta - a\delta_{t,0}\big) + j_t h_t \eta + K_\xi(-\tilde{z}_t g_{t-1}) \Big].
\end{aligned}
\tag{8}
$$

From the first line, one immediately sees that differentiating $Z[j]$ with respect to $j_t \eta$ yields the moments of $h_t$, e.g., $\partial_{j_t \eta} Z[j]\big|_{j_t = 0} = \langle h_t \rangle_h$ and so on.

We now take the continuous-time limit $\eta \to 0$, and introduce the standard notation $\lim_{\eta \to 0} \prod_{t=0}^{T} \int_{-\infty}^{\infty} dh_t =: \int \mathcal{D}h$, and $\lim_{\eta \to 0} \prod_{t=0}^{T} \int_{-i\infty}^{i\infty} \frac{d\tilde{z}_t}{2\pi i} =: \int \mathcal{D}\tilde{z}$ for the path-integrals over the real $h$ and complex $\tilde{z}$ fields.[12] Within the exponential, we have $\lim_{\eta \to 0} \sum_t h_t \eta = \int dt\, h(t)$ and $\lim_{\eta \to 0} \sum_t \frac{h_t - h_{t-1}}{\eta} \eta = \int dt\, \partial_t h(t)$, as well as the convention $\lim_{\eta \to 0} \frac{\delta_{t,0}}{\eta} = \delta(t)$, hence

$$
Z[j] = \int \mathcal{D}h \int \mathcal{D}\tilde{z} \, \exp\Big\{ \int dt\, [\tilde{z}(t)(\partial_t h(t) - f(t) - a\delta(t)) + j(t)h(t)] + K_S(-\tilde{z}g) \Big\}, \tag{9}
$$

where $K_S(-\tilde{z}g)$ is the continuum-limit of the cumulant generating function:

$$
\begin{aligned}
\sum_t K_\xi(-\tilde{z}_t g_{t-1}) &= \ln \prod_t \int d\xi_t \, \rho(\xi_t) e^{-\tilde{z}_t g_{t-1} \xi_t} \\
&= \ln \prod_t \Big\{ \int d\xi_t \, \rho(\xi_t) \Big\} \exp\Big( -\sum_t \tilde{z}_t g_{t-1} \frac{dS_t}{dt} \eta \Big) \\
&\stackrel{\eta \to 0}{=} \ln \int \mathcal{D}\xi \, \exp\Big\{ -\int dt\, \tilde{z}(t) g(t) \frac{dS}{dt} \Big\} \\
&= \ln \int \mathcal{D}\xi \, \exp\Big\{ -\int dS\, \tilde{z}(t) g(t) \Big\} =: K_S(-\tilde{z}g),
\end{aligned}
\tag{10}
$$

where we have absorbed the probability density function into the path-integral measure, i.e., $\int \mathcal{D}\xi := \lim_{\eta \to 0} \prod_t \int d\xi_t \, \rho(\xi_t)$; in the case of Gaussian noise for example, this would reduce to the standard multidimensional Gaussian measure. Note that in these expressions, $f_{t-1} \to f(t)$ and $g_{t-1} \to g(t)$, since the subscript denotes the shift $\lim_{\eta \to 0}(t-\eta) = t$. Moments are now obtained from the continuum partition function (9) via functional differentiation with respect to the source; that is, the $n^{\text{th}}$ moment is given by the $n$-point correlator

$$
\langle h(s_1) \ldots h(s_n) \rangle = \frac{\delta^n}{\delta j(s_1) \ldots \delta j(s_n)} Z[j] \Big|_{j=0}, \tag{11}
$$

where the operator insertions $h(s_i)$, $i \in \{1, \ldots, n\}$ need not be time-ordered.

---

[12]Formally, this simply amounts to recovering (1) from the Itô discretization (3). Note that while there is no obvious continuum limit in the neural index, there *is* a sensible continuum limit in the temporal/layer index (recall that we work at $T \to \infty$), and it is the latter we are considering here, hence the $N$-component fields that give rise to the analogies with the $O(N)$ vector model below; we thank Dan and Sho for discussions on this point.

What about the correlator of the auxiliary field $\tilde{z}$? Observe that we may add a source term $\tilde{j}(t)\tilde{z}(t)$ to (9) to obtain the moment generating function for both $h$ and $\tilde{z}$:

$$Z[j,\tilde{j}] = \int \mathcal{D}h \int \mathcal{D}\tilde{z} \exp\left\{\int dt\,[\tilde{z}(t)(\partial_t h(t) - f(t)) + j(t)h(t) + \tilde{j}(t)\tilde{z}(t)] + K_S(-\tilde{z}g)\right\}, \quad (12)$$

where for compactness we will henceforth suppress the initial condition $a\delta(t)$ (note that this can be absorbed into the source term for $\tilde{z}$). This corresponds to adding a time-dependent driving term to the SDE (1), i.e.,

$$dh = (f(h,x) - \tilde{j}(t))\,dt + g(h,x)\,dS\,. \quad (13)$$

Therefore, the auxiliary field can be interpreted as an external perturbation to the system, the response to which is measured by $\tilde{z}$ (hence the name). The nonphysical nature of the auxiliary field $\tilde{z}$ is encoded by the absence of a kinetic term, which implies that these fields do not propagate, i.e., all $n$-point functions of $\tilde{z}$ vanish. To see this, observe that since (4) is a properly normalized distribution,[13] the partition function (9) in the absence of sources retains this normalization, i.e., $Z[j=0] = 1$. Since the perturbation $\tilde{j}$ only appears on the right-hand side of (13), it simply shifts the solution $y_t$ (5), and does not affect this normalization (i.e., the construction of $Z$ holds for any solution). Hence $Z[j=0,\tilde{j}] = 1$ for all $\tilde{j}$, which immediately implies that all self-correlations of the auxiliary field vanish:

$$\langle\tilde{z}(s_1)\ldots\tilde{z}(s_n)\rangle = \frac{\delta^n}{\delta\tilde{j}(s_1)\ldots\delta\tilde{j}(s_n)}Z[0,\tilde{j}]\bigg|_{\tilde{j}=0} = 0\,. \quad (14)$$

## 2.2 Self-averaging random networks

Recall that the function $f(t)$ appearing in the partition function (12) depends on the trainable parameters $A, B, W, U, b$, cf. (2). Henceforth we shall consider a class of so-called random neural networks, in which these parameters are initialized as independent Gaussian variables,

$$\begin{aligned}
X_{ij} &\sim \mathcal{N}(\mu_x, \sigma_x^2/N)\,, & X = [X_{ij}] \in \{A, B, W, U\}\,, \\
b_i &\sim \mathcal{N}(\mu_b, \sigma_b^2)\,, & b = [b_i]\,,
\end{aligned} \quad (15)$$

i.e.,

$$\rho(X_{ij}) = \sqrt{\frac{N}{2\pi\sigma_x^2}}\,e^{-\frac{N}{2}\left(\frac{X_{ij}-\mu_X}{\sigma_x}\right)^2}\,, \quad (16)$$

and similarly for $\rho(b_i)$. Note that as usual, the variances of $X_{ij}$ are scaled by the width, to ensure that the activations at subsequent layers remain $\mathcal{O}(1)$ at large $N$. The partition function then describes an ensemble of networks, and we expect that in the large-$N$ limit, the observables of interest are captured by the ensemble average $\langle Z[j]\rangle_{X,b} =: \bar{Z}[j]$. The idea is that the network is *self-averaging*, in that while the particular instantiation varies from one realization of $X_{ij}$, $b_i$ to the next, the physical properties of the ensemble should approach that described by the average over the (random) couplings, provided the fluctuations around the mean values are sufficiently small. This statement is closely related to the central limit theorem, and becomes exact as $N \to \infty$ [36].[14] The ensemble average is then

$$\bar{Z}[j,\tilde{j}] = \langle Z[j,\tilde{j}]\rangle_{X,b} = \prod_{ij}\int dX_{ij}\int db_i\,\rho(X_{ij})\rho(b_i)Z[j,\tilde{j}] =: \int \mathcal{D}X \int \mathcal{D}b\,Z[j,\tilde{j}]\,, \quad (17)$$

---

[13]Integrating over all paths, we have $\prod_t \int dh_t\,p(h_0,\ldots,h_T) = \prod_t \int d\xi_t\,\rho(\xi_t)\underbrace{\int dh_t\,\delta(h_t - y_t)}_{1} = 1$.

[14]As we will discuss in more detail below, the CLT does *not* appear to yield equivalent results to MFT in the present case, even in the infinite-width limit.

where $\rho(X_{ij})$ for each $X \in \{A, B, W, U\}$ are the normalized Gaussian probability density functions (16), and similarly for $\rho(b_i)$, which we have absorbed into the measures $\mathcal{D}X$, $\mathcal{D}b$ for compactness. Substituting the rule (2) into the partition function (12) and integrating over the disorder $X$ and the bias $b$, we obtain

$$\bar{Z}[j, \tilde{j}] = \int \mathcal{D}h \int \mathcal{D}\tilde{z} \, e^{S_0 + S_{\text{source}} + S_{\text{int}}}, \tag{18}$$

where

$$
\begin{aligned}
S_0 &= \int dt \, \tilde{z}(t)(\partial_t + \gamma)h(t) + K_S(-\tilde{z}g), \\
S_{\text{source}} &= \int dt \, [j(t)h(t) + \tilde{j}(t)\tilde{z}(t)],
\end{aligned}
\tag{19}
$$

and the integration over $X \in \{A, B, W, U\}$ and $b$ is contained in the interaction term

$$
\begin{aligned}
e^{S_{\text{int}}} &= \int \mathcal{D}A \int \mathcal{D}B \int \mathcal{D}W \int \mathcal{D}U \int \mathcal{D}b \\
&\times \exp\left\{ -\int dt \, \tilde{z}_i(t) \left[ A_{ij}h_j(t) + B_{ij}x_j(t) + W_{ij}\phi(h_j(t)) + U_{ij}\varphi(x_j(t)) + b_i \right] \right\} \\
&= \left\langle e^{-A_{ij}\int dt \, \tilde{z}_i h_j} \right\rangle_A \left\langle e^{-B_{ij}\int dt \, \tilde{z}_i x_j} \right\rangle_B \left\langle e^{-W_{ij}\int dt \, \tilde{z}_i \phi(h_j)} \right\rangle_W \left\langle e^{-U_{ij}\int dt \, \tilde{z}_i \varphi(x_j)} \right\rangle_U \left\langle e^{-b_i\int dt \, \tilde{z}_i} \right\rangle_b,
\end{aligned}
\tag{20}
$$

where in going to the third line, the exponential has factorized into a product of moment generating functions for $X_{ij}$ and $b_i$, with an implicit sum over repeated neuron indices $i, j$. Due to the i.i.d. condition (15), each of these is a simple product of Gaussian integrals, e.g.,

$$\left\langle e^{-W_{ij}\int dt \, \tilde{z}_i \phi(h_j)} \right\rangle_W = \prod_{ij} \sqrt{\frac{N}{2\pi\sigma_w^2}} \int dW_{ij} \exp\left\{ -\frac{N}{2}\left(\frac{W_{ij}-\mu_w}{\sigma_w}\right)^2 - W_{ij}\int dt \, \tilde{z}_i \phi(h_j) \right\}. \tag{21}$$

For compactness, let us define $y_{ij} := \int dt \, \tilde{z}_i \phi(h_j)$. Then completing the square in the exponent, we have

$$
\begin{aligned}
\left\langle e^{-W_{ij}\int dt \, \tilde{z}_i \phi(h_j)} \right\rangle_W &= \prod_{ij} \sqrt{\frac{N}{2\pi\sigma_w^2}} \int dW_{ij} \exp\left\{ -\frac{N}{2}\left(\frac{W_{ij}-\mu_w}{\sigma_w}\right)^2 - W_{ij}y_{ij} \right\} \\
&= e^{\frac{1}{2}\sum_{ij}\left(\frac{\sigma_w^2}{N}y_{ij}^2 - 2\mu_w y_{ij}\right)} \prod_{ij} \sqrt{\frac{N}{2\pi\sigma_w^2}} \int dW_{ij} \exp\left\{ -\frac{N}{2\sigma_w^2}\left[ W_{ij} + \left(\frac{\sigma_w^2}{N}y_{ij} - \mu_w\right) \right]^2 \right\} \\
&= \exp\sum_{ij}\left\{ \frac{\sigma_w^2}{2N}\int dt_1 dt_2 \, \tilde{z}_i(t_1)\tilde{z}_i(t_2)\phi(h_j(t_1))\phi(h_j(t_2)) - \mu_w\int dt \, \tilde{z}_i(t)\phi(h_j(t)) \right\},
\end{aligned}
\tag{22}
$$

where in going to the last line, the Gaussian integrals have each evaluated to unity, and we have inserted the definition of $y_{ij}$ above. Note that the effect of integrating over the disorder $W$ is the introduction of a coupling term of order $\phi^2$. The integrations over $A, B, U, b$ are all formally identical to (22). For simplicity, we shall henceforth take $\mu_A = \mu_B = \mu_w = \mu_u = \mu_b = 0$, so that terms linear in $h$, $\tilde{z}$, $\phi$ and $\varphi$ drop out. Note however that we retain the ability to effectively shift the mean of $A$ by tuning the constant $\gamma$ above. The interaction term (20) is then

$$
\begin{aligned}
S_{\text{int}} = \frac{1}{2N}\int dt_1 dt_2 \sum_i \tilde{z}_i(t_1)\tilde{z}_i(t_2)\sum_j \Big[ &\sigma_b^2 + \sigma_A^2 h_j(t_1)h_j(t_2) + \sigma_B^2 x_j(t_1)x_j(t_2) \\
&+ \sigma_w^2 \phi(h_j(t_1))\phi(h_j(t_2)) + \sigma_u^2 \varphi(x_j(t_1))\varphi(x_j(t_2)) \Big].
\end{aligned}
\tag{23}
$$

Despite the name however, observe that the system has been reduced to $N$ independent subsystems ($\tilde{z}_i^2$, $i \in \{1, \ldots, N\}$) with identical couplings given by the sum over $j$. The latter is a collection of $N$ weakly correlated contributions, which we may represent by introducing the bi-local field variables $\mathfrak{A}, \mathfrak{B}, \mathfrak{W}, \mathfrak{U}$:

$$\mathfrak{A}(t_1, t_2) := \frac{\sigma_A^2}{N} \sum_j h_j(t_1) h_j(t_2), \qquad \mathfrak{B}(t_1, t_2) := \frac{\sigma_B^2}{N} \sum_j x_j(t_1) x_j(t_2),$$

$$\mathfrak{W}(t_1, t_2) := \frac{\sigma_w^2}{N} \sum_j \phi(h_j(t_1)) \phi(h_j(t_2)), \qquad \mathfrak{U}(t_1, t_2) := \frac{\sigma_u^2}{N} \sum_j \varphi(x_j(t_1)) \varphi(x_j(t_2)). \tag{24}$$

In the large-$N$ (i.e., infinite width) limit, the value of these contributions will approach their respective means. The *mean-field theory* approximation to the ensemble average is then obtained as the leading-order saddlepoint contribution from $\mathfrak{A}, \mathfrak{B}, \mathfrak{W}, \mathfrak{U}$, treated as independent fields that we allow to fluctuate.

To proceed, we enforce the constraints (24) via delta functions as in (4),

$$e^{S_{\text{int}}} = \int \mathcal{D}\mathfrak{A}\, \mathcal{D}\mathfrak{B}\, \mathcal{D}\mathfrak{W}\, \mathcal{D}\mathfrak{U}$$

$$\times \exp\left\{ \frac{1}{2} \int dt_1\, dt_2 \sum_i \tilde{z}_i(t_1) \tilde{z}_i(t_2) \left[ \sigma_b^2 + \mathfrak{A}(t_1, t_2) + \mathfrak{B}(t_1, t_2) + \mathfrak{W}(t_1, t_2) + \mathfrak{U}(t_1, t_2) \right] \right\}$$

$$\times \delta\left( \sum_j h_j(t_1) h_j(t_2) - \frac{N}{\sigma_A^2} \mathfrak{A}(t_1, t_2) \right) \delta\left( \sum_j x_j(t_1) x_j(t_2) - \frac{N}{\sigma_B^2} \mathfrak{B}(t_1, t_2) \right)$$

$$\times \delta\left( \sum_j \phi(h_j(t_1)) \phi(h_j(t_2)) - \frac{N}{\sigma_w^2} \mathfrak{W}(t_1, t_2) \right) \delta\left( \sum_j \varphi(x_j(t_1)) \varphi(x_j(t_2)) - \frac{N}{\sigma_u^2} \mathfrak{U}(t_1, t_2) \right),$$

and introduce an integration over auxiliary fields $\tilde{A}, \tilde{B}, \tilde{W}, \tilde{U}$ as in (7):

$$e^{S_{\text{int}}} = \int \mathcal{D}\mathfrak{A}\, \mathcal{D}\mathfrak{B}\, \mathcal{D}\mathfrak{W}\, \mathcal{D}\mathfrak{U} \int \mathcal{D}\tilde{A}\, \mathcal{D}\tilde{B}\, \mathcal{D}\tilde{W}\, \mathcal{D}\tilde{U}$$

$$\times \exp \int dt_1\, dt_2 \left\{ \frac{1}{2} \sum_i \tilde{z}_i(t_1) \tilde{z}_i(t_2) \left[ \sigma_b^2 + \mathfrak{A}(t_1, t_2) + \mathfrak{B}(t_1, t_2) + \mathfrak{W}(t_1, t_2) + \mathfrak{U}(t_1, t_2) \right] \right.$$

$$- \frac{N}{\sigma_A^2} \tilde{A}(t_1, t_2) \mathfrak{A}(t_1, t_2) - \frac{N}{\sigma_B^2} \tilde{B}(t_1, t_2) \mathfrak{B}(t_1, t_2) - \frac{N}{\sigma_w^2} \tilde{W}(t_1, t_2) \mathfrak{W}(t_1, t_2)$$

$$- \frac{N}{\sigma_u^2} \tilde{U}(t_1, t_2) \mathfrak{U}(t_1, t_2) + \tilde{A}(t_1, t_2) \sum_j h_j(t_1) h_j(t_2) + \tilde{B}(t_1, t_2) \sum_j x_j(t_1) x_j(t_2)$$

$$\left. + \tilde{W}(t_1, t_2) \sum_j \phi(h_j(t_1)) \phi(h_j(t_2)) + \tilde{U}(t_1, t_2) \sum_j \varphi(x_j(t_1)) \varphi(x_j(t_2)) \right\}. \tag{25}$$

Despite its unwieldy form, this expression preserves the independence of the $N$ systems we observed in (23), since the (auxiliary) fields couple only to sums over $j \in \{1, \ldots, N\}$ (i.e., we have the same integral over $\tilde{z}_j, h_j$ for all $j$). As alluded above, the remaining parts of the action (19) share this property, as one sees by writing out the implicit summations; e.g.,

$$S_0 = \sum_i \left\{ \int dt\, \tilde{z}_i(t) (\partial_t + \gamma) h_i(t) + \ln \int \mathcal{D}\xi\, e^{-\int dB\, \tilde{z}_i g_i} \right\}, \tag{26}$$

and similarly for $S_{\text{source}}$.[15] Therefore, the generating function for $h, \tilde{z}$ factorizes into a product of $N$ factors – which we shall denote $\bar{Z}_i[j, \tilde{j}]$ – allowing us to express the total average partition

---

[15] The cumulant term requires moving the summation through both the exponential and the log, i.e., $\ln \int e^{\sum_i y_i} = \ln \int \prod e^{y_i} = \ln \prod \int e^{y_i} = \sum \ln \int e^{y_i}$.

function in the form[16]

$$
\begin{aligned}
\bar{Z} = \int \mathcal{D}\mathfrak{A}\,\mathcal{D}\mathfrak{B}\,\mathcal{D}\mathfrak{W}\,\mathcal{D}\mathfrak{U} \int \mathcal{D}\tilde{A}\,\mathcal{D}\tilde{B}\,\mathcal{D}\tilde{W}\,\mathcal{D}\tilde{U} \exp\Bigg\{ & \int dt_1\,dt_2 \Big[ -\frac{N}{\sigma_A^2}\tilde{A}(t_1,t_2)\mathfrak{A}(t_1,t_2) \\
& -\frac{N}{\sigma_B^2}\tilde{B}(t_1,t_2)\mathfrak{B}(t_1,t_2) - \frac{N}{\sigma_w^2}\tilde{W}(t_1,t_2)\mathfrak{W}(t_1,t_2) - \frac{N}{\sigma_u^2}\tilde{U}(t_1,t_2)\mathfrak{U}(t_1,t_2) \Big] \\
& + N \ln \bar{Z}_i[j,\tilde{j}] \Bigg\},
\end{aligned}
\tag{27}
$$

where for compactness we have suppressed the sources for the fields $\mathfrak{X} \in \{\mathfrak{A},\mathfrak{B},\mathfrak{W},\mathfrak{U}\}$ and auxiliaries $\tilde{X} \in \{\tilde{A},\tilde{B},\tilde{W},\tilde{U}\}$, and the generating function for $h_i, \tilde{z}_i$ is

$$
\begin{aligned}
\bar{Z}_i[j,\tilde{j}] := \int \mathcal{D}h_i \int \mathcal{D}\tilde{z}_i \exp\Bigg\{ & S_0[h_i,\tilde{z}_i] + S_{\text{source}}[j_i,\tilde{j}_i] \\
& + \int dt_1\,dt_2 \Big( \frac{1}{2}\big[\sigma_b^2 + \mathfrak{A}(t_1,t_2) + \mathfrak{B}(t_1,t_2) + \mathfrak{W}(t_1,t_2) + \mathfrak{U}(t_1,t_2)\big]\tilde{z}_i(t_1)\tilde{z}_i(t_2) \\
& \quad + \tilde{A}(t_1,t_2)h_i(t_1)h_i(t_2) + \tilde{B}(t_1,t_2)x_i(t_1)x_i(t_2) \\
& \quad + \tilde{W}(t_1,t_2)\phi(h_i(t_1))\phi(h_i(t_2)) + \tilde{U}(t_1,t_2)\varphi(x_i(t_1))\varphi(x_i(t_2)) \Big) \Bigg\}.
\end{aligned}
\tag{28}
$$

(Note the subscript $i$ on the Gaussian integration measures, in contrast to $\int \mathcal{D}h := \prod_i \int \mathcal{D}h_i$). Thus, we have reduced the initially complicated system of $N$ interacting units to that of a single unit under the influence of external fields $\mathfrak{X},\tilde{X}$. Henceforth, we may drop the source term $S_{\text{source}}[h_i,\tilde{z}_i]$, and consider the field theory for $\mathfrak{X},\tilde{X}$ in their own right. We emphasize that up to this point, the result (27) is exact.

## 2.3 Mean-field theory approximation

We now perform the aforementioned saddlepoint approximation, from which the MFT is obtained at leading order. Let us express the partition function schematically as $\bar{Z} = \int \mathcal{D}\mu\, e^{-S[\mu]}$. Provided the exponential decays sufficiently rapidly, the integral will be dominated by the minimum value $\mu_0$. The saddlepoint approximation simply consists of Taylor expanding $S[\mu]$ about the minimum:

$$
\bar{Z} = e^{-S[\mu_0]} \int \mathcal{D}\mu\, e^{-\frac{1}{2}(\mu-\mu_0)^2 S''[\mu_0]+\cdots},
\tag{29}
$$

where the prime denotes (functional) differentiation with respect to $\mu$, and the linear term vanishes by definition, i.e., $S'[\mu_0] = 0$. Note that while this may be a very poor approximation to $S[\mu]$ when $|\mu - \mu_0| \gg 0$, it will still give an accurate approximation to $\bar{Z}$ due to the exponential suppression of higher-order terms. If we take only the leading-order contribution, then the partition function is given entirely by the prefactor, $e^{-S[\mu_0]}$. It then remains simply to determine $\mu_0$.

In the present case, the minimality condition yields eight equations, one for each field $\mathfrak{X},\tilde{X}$.

---

[16]Note that while we require $N$ to be sufficiently large for the Gaussian distributions to be valid, the factorization itself holds even at finite $N$, since it relies only on each term in the summations over $\tilde{z}_i^2$, $\phi(h_i)^2$, and $\varphi(x_i)^2$ being identical, which is true by virtue of the integrals over $h_i, \tilde{z}_i$ in (28). No uniformity condition is required on the external data $x$, cf. footnote 10.

Starting with $\mathfrak{A}$, we have[17]

$$
\begin{aligned}
\frac{\delta S}{\delta \mathfrak{A}(t_1, t_2)} &= -\frac{N}{\sigma_A^2}\tilde{A}(t_1, t_2) + \frac{N}{\bar{Z}_i}\frac{\delta \bar{Z}_i}{\delta \mathfrak{A}(t_1, t_2)} \\
&= -\frac{N}{\sigma_A^2}\tilde{A}(t_1, t_2) + \frac{N}{2}\langle \tilde{z}_i(t_1)\tilde{z}_i(t_2)\rangle_i,
\end{aligned}
\tag{30}
$$

where $S$ is the full action appearing in (27), and in the second equality, we have observed that $\mathfrak{A}$ acts as a source for the $\tilde{z}^2$ term in (28). The subscript on the correlator is understood to denote the expectation value with respect to the "single neuron" partition function (28), with all sources (i.e., $\mathfrak{X}, \tilde{X}$) held fixed. Repeating this procedure with respect to the remaining fields and setting each variation to zero, we obtain the following system of equations for the minima $X_0$ (denoted with an additional subscript 0 on the expectation value):[18]

$$
\begin{aligned}
\mathfrak{A}_0(t_1, t_2) &= \sigma_A^2\langle h_i(t_1)h_i(t_2)\rangle_{i,0} =: \sigma_A^2 C_{hh}(t_1, t_2), \\
\mathfrak{B}_0(t_1, t_2) &= \sigma_B^2\langle x_i(t_1)x_i(t_2)\rangle_{i,0} =: \sigma_B^2 C_{xx}(t_1, t_2), \\
\mathfrak{W}_0(t_1, t_2) &= \sigma_w^2\langle \phi(h_i(t_1))\phi(h_i(t_2))\rangle_{i,0} =: \sigma_w^2 C_{\phi\phi}(t_1, t_2), \\
\mathfrak{U}_0(t_1, t_2) &= \sigma_u^2\langle \varphi(x_i(t_1))\varphi(x_i(t_2))\rangle_{i,0} =: \sigma_u^2 C_{\varphi\varphi}(t_1, t_2), \\
\tilde{A}_0(t_1, t_2) &= \frac{\sigma_A^2}{2}\langle \tilde{z}_i(t_1)\tilde{z}_i(t_2)\rangle_{i,0} = 0, \\
\tilde{B}_0(t_1, t_2) &= \frac{\sigma_B^2}{2}\langle \tilde{z}_i(t_1)\tilde{z}_i(t_2)\rangle_{i,0} = 0, \\
\tilde{W}_0(t_1, t_2) &= \frac{\sigma_w^2}{2}\langle \tilde{z}_i(t_1)\tilde{z}_i(t_2)\rangle_{i,0} = 0, \\
\tilde{U}_0(t_1, t_2) &= \frac{\sigma_u^2}{2}\langle \tilde{z}_i(t_1)\tilde{z}_i(t_2)\rangle_{i,0} = 0,
\end{aligned}
\tag{31}
$$

where the correlators of $\tilde{z}$ vanish by (14), and we have introduced the compact notation $C_{yy}$ for the two-point correlators of $y_i \in \{h_i, x_i, \phi_i, \varphi_i\}$. Substituting these into the action, we obtain the leading-order saddlepoint approximation $\bar{Z}_* := e^{N \ln \bar{Z}_{i,0}} = \bar{Z}_{i,0}^N$ to the partition function (27), where

$$
\begin{aligned}
\bar{Z}_{i,0} = \int \mathcal{D}h_i \int \mathcal{D}\tilde{z}_i \exp\Big\{ S_0 + \frac{1}{2}\int \mathrm{d}t_1\,\mathrm{d}t_2\Big[ \sigma_b^2 + \sigma_A^2 C_{hh}(t_1, t_2) + \sigma_B^2 C_{xx}(t_1, t_2) \\
+ \sigma_w^2 C_{\phi\phi}(t_1, t_2) + \sigma_u^2 C_{\varphi\varphi}(t_1, t_2)\Big]\tilde{z}_i(t_1)\tilde{z}_i(t_2)\Big\},
\end{aligned}
\tag{32}
$$

where we have dropped the sources, and the integrals over auxiliary fields only appear at higher orders, cf. (29). By virtue of the decoupling into $N$ non-interacting terms, we have effectively reduced the problem to that of a single system $S_0(h_i, \tilde{z}_i)$ exposed to a common Gaussian noise in the form of the two-point functions $C_{aa}$. That no higher-point correlators appear is a manifestation of the fact that the MFT ignores non-Gaussian fluctuations.

---

[17]Note that the functional derivative is performed at a particular $t_1, t_2$, which results in delta functions that kill the integrals over time in (27) and (28).

[18]Recall from footnote 10 that we treat the data $x(t)$ as external, which is why it is not integrated over in the partition function. Thus while we have written the second line of this expression in general, the product $x_i(t_1)x_i(t_2)$ comes out of the expectation value, which evaluates to unity. Alternatively, one could consider evaluating the theory on data drawn from some ensemble.

We now wish to obtain expressions for the propagators of the MFT for $h_i, \tilde{z}_i$ described by (32), which requires us to choose an explicit form for the stochasticity as encoded by the cumulant generating function (10). For simplicity, we shall henceforth consider the case in which the stochastic increments are independently normally distributed according to $\xi_t \sim \mathcal{N}(0, \kappa\eta)$, where $\kappa$ is a constant parameter. The choice of standard deviation – in particular the appearance of the discrete timestep $\eta$ – is necessary to ensure a consistent continuum limit. That is, observe that in (8), $K_\xi$ must be proportional to $\eta$ in order to convert the sum to an integral. Explicitly, inserting $\rho(\xi_t) = \frac{1}{\sqrt{2\pi\kappa\eta}} e^{-\xi_t^2/(2\kappa\eta)}$ into (10), we have

$$
\begin{aligned}
K_S(-\tilde{z}g) &= \lim_{\eta\to 0} \ln \prod_t \int \frac{\mathrm{d}\xi_t}{\sqrt{2\pi\kappa\eta}}\, e^{-\frac{1}{2\kappa\eta}\xi_t^2 - \tilde{z}_t g_{t-1}\xi_t} \\
&= \lim_{\eta\to 0} \frac{\kappa}{2} \sum_t (\tilde{z}_t g_{t-1})^2 \eta = \frac{\kappa}{2} \int \mathrm{d}t\, \tilde{z}(t)^2 g(t)^2 .
\end{aligned}
\tag{33}
$$

Note that while this restricts the form of the action to quadratic order in $\tilde{z}$, we retain the freedom to choose the diffusion coefficient $g(t)$.

Substituting the Gaussian cumulant generating function (33) into the MFT action (32), we have

$$
\begin{aligned}
\bar{Z}_{i,0} &= \int \mathcal{D}h_i \int \mathcal{D}\tilde{z}_i \exp\Bigg\{ \int \mathrm{d}t \left[ \tilde{z}_i(t)(\partial_t + \gamma)h_i(t) + \frac{\kappa}{2}\tilde{z}_i(t)^2 g_i(t)^2 \right] \\
&\quad + \frac{1}{2}\int \mathrm{d}t_1\, \mathrm{d}t_2\, \tilde{z}_i(t_1) \Big[ \sigma_b^2 + \sigma_A^2 C_{hh}(t_1, t_2) + \sigma_B^2 C_{xx}(t_1, t_2) + \sigma_w^2 C_{\phi\phi}(t_1, t_2) \\
&\quad\quad\quad + \sigma_u^2 C_{\varphi\varphi}(t_1, t_2) \Big] \tilde{z}_i(t_2) \Bigg\} \\
&= \int \mathcal{D}h_i \int \mathcal{D}\tilde{z}_i \exp\left\{ -\frac{1}{2}\int \mathrm{d}t_1\, \mathrm{d}t_2\, y_i^{\mathsf{T}}(t_1)\, \Xi(t_1, t_2)\, y_i(t_2) \right\} ,
\end{aligned}
\tag{34}
$$

where we have defined

$$
y_i(t) := \begin{pmatrix} h_i(t) \\ \tilde{z}_i(t) \end{pmatrix} , \qquad \Xi(t_1, t_2) = \begin{pmatrix} \Xi_{hh}(t_1, t_2) & \Xi_{h\tilde{z}}(t_1, t_2) \\ \Xi_{\tilde{z}h}(t_1, t_2) & \Xi_{\tilde{z}\tilde{z}}(t_1, t_2) \end{pmatrix} ,
\tag{35}
$$

with the operator elements

$$
\begin{aligned}
\Xi_{hh}(t_1, t_2) &= 0 , \\
\Xi_{h\tilde{z}}(t_1, t_2) &= \delta(t_1 - t_2)(\partial_{t_2} - \gamma) , \\
\Xi_{\tilde{z}h}(t_1, t_2) &= -\delta(t_1 - t_2)(\partial_{t_2} + \gamma) , \\
\Xi_{\tilde{z}\tilde{z}}(t_1, t_2) &= -\sigma_b^2 - \sigma_A^2 C_{hh}(t_1, t_2) - \sigma_B^2 C_{xx}(t_1, t_2) - \sigma_w^2 C_{\phi\phi}(t_1, t_2) - \sigma_u^2 C_{\varphi\varphi}(t_1, t_2) \\
&\quad - \kappa\delta(t_1 - t_2)g(t_2)^2 ,
\end{aligned}
\tag{36}
$$

where the upper-right element appears via integration by parts of the kinetic term.[19]

The *propagators* are then obtained as the matrix elements of the Green function $G$ for the matrix operator $\Xi$, defined as the right-inverse

$$
\int \mathrm{d}s\, \Xi(t_1, s) G(s, t_2) = \delta(t_1 - t_2)\mathbb{1} .
\tag{37}
$$

---

[19]That is, $\int \mathrm{d}t\, \tilde{z}\partial_t h = \frac{1}{2}\int \mathrm{d}t\, \tilde{z}\partial_t h - \frac{1}{2}\int \mathrm{d}t\, (\partial_t \tilde{z})h$; the vanishing of the boundary term is consistent with the fact that the two-point functions of $\tilde{z}$ and $h$ are either decaying exponentially or identically zero, cf. (121), (122).

Denoting the components of $G$ as in (35), and using the previously-determined fact that $G_{\tilde{z}\tilde{z}} = 0$ (cf. (14)), this yields the matrix equation

$$\int ds \begin{pmatrix} \Xi_{h\tilde{z}}(t_1,s)G_{\tilde{z}h}(s,t_2) & 0 \\ \Xi_{\tilde{z}h}(t_1,s)G_{hh}(s,t_2) + \Xi_{\tilde{z}\tilde{z}}(t_1,s)G_{\tilde{z}h}(s,t_2) & \Xi_{\tilde{z}h}(t_1,s)G_{h\tilde{z}}(s,t_2) \end{pmatrix} = \delta(t_1-t_2)\mathbb{1}, \tag{38}$$

which has three non-trivial components; denoting $\partial_{t_i} =: \partial_i$, we have

$$\delta(t_1-t_2) = \int ds\,\delta(t_1-s)(\partial_s - \gamma)G_{\tilde{z}h}(s,t_2) = (\partial_1 - \gamma)G_{\tilde{z}h}(t_1,t_2),$$

$$\delta(t_1-t_2) = -\int ds\,\delta(t_1-s)(\partial_s + \gamma)G_{h\tilde{z}}(s,t_2) = -(\partial_1 + \gamma)G_{h\tilde{z}}(t_1,t_2),$$

$$0 = (\partial_1 + \gamma)G_{hh}(t_1,t_2) \tag{39}$$

$$+ \int ds\left[\sigma_b^2 + \sigma_A^2 C_{hh}(t_1,s) + \sigma_B^2 C_{xx}(t_1,s) + \sigma_w^2 C_{\phi\phi}(t_1,s) + \sigma_u^2 C_{\varphi\varphi}(t_1,s)\right.$$

$$\left. + \kappa\delta(t_1-s)g(s)^2\right]G_{\tilde{z}h}(s,t_2).$$

Note that $G_{hh} \equiv C_{hh}$: what this tells us is that the expression for the transition amplitude from $h(t_1)$ to $h(t_2)$ must be determined self-consistently from these equations, i.e., that the correlation $\langle h_i(t_1)h_i(t_2)\rangle$ depends on the correlations in the other variables, via the interactions induced by the disorder we integrated out above.

Let us now assume that the system exhibits time translation symmetry, so that $G(s,t) = G(s-t)$. This allows us to swap derivatives à la $\partial_s G(s-t) = -\partial_t G(s-t)$. Performing this little slight of hand on the first of the equations above yields

$$(\partial_2 + \gamma)G_{\tilde{z}h}(t_1-t_2) = -\delta(t_1-t_2). \tag{40}$$

If we then act on the third equation with $(\partial_2 + \gamma)$, we obtain

$$(\partial_1 + \gamma)(\partial_2 + \gamma)G_{hh}(t_1-t_2)$$

$$= \int ds\left[\sigma_b^2 + \sigma_A^2 G_{hh}(t_1-s) + \sigma_B^2 C_{xx}(t_1,s) + \sigma_w^2 C_{\phi\phi}(t_1,s) + \sigma_u^2 C_{\varphi\varphi}(t_1,s)\right.$$

$$\left. + \kappa\delta(t_1-s)g(s)^2\right]\delta(s-t_2) \tag{41}$$

$$= \sigma_b^2 + \sigma_A^2 G_{hh}(t_1-t_2) + \sigma_B^2 C_{xx}(t_1,t_2) + \sigma_w^2 C_{\phi\phi}(t_1,t_2) + \sigma_u^2 C_{\varphi\varphi}(t_1,t_2)$$

$$+ \kappa\delta(t_1-t_2)g(t_2)^2.$$

Given our assumption of (time) translation invariance, it is convenient to define $\tau := t_1 - t_2$. Then the off-diagonal terms are given by

$$(\partial_\tau - \gamma)\,G_{\tilde{z}h}(\tau) = (-\partial_\tau - \gamma)\,G_{h\tilde{z}}(\tau) = \delta(\tau), \tag{42}$$

while for the diagonal term, (41) becomes

$$\left(-\partial_\tau^2 + \gamma^2 - \sigma_A^2\right)G_{hh}(\tau) = \sigma_b^2 + \sigma_B^2 C_{xx}(\tau) + \sigma_w^2 C_{\phi\phi}(\tau) + \sigma_u^2 C_{\varphi\varphi}(\tau) + \kappa g(\tau)^2\delta(\tau). \tag{43}$$

This is the analogue of eq. (147) in [36]; the treatment above simply extends this framework to the stochastic recursive networks of interest. Upon setting $\gamma = 1$, $g = 0$, and setting all the variances to 0, our result reduces to the non-stochastic, purely feed-forward case considered in the seminal work [52], based on the early MFT analysis in [53]. While we believe this to

be the historical origin of the (use of the phrase) "mean-field approximation" that has recently appeared in the machine learning literature, we emphasize again that this does *not* necessarily correspond to the $N \to \infty$ limit actually used in [17, 29] and related works. We shall show this explicitly in the next section, where correlation functions in the infinite-width limit retain an $\mathcal{O}(1)$ contribution, which for some parameter values represents a substantial correction to the tree-level (MFT) result.

Now, our primary interest is in the propagator $G_{hh}(\tau) = \langle h_i(t_1) h_i(t_2) \rangle$. In particular, in section 3, this solution is taken as a background around which chaotic fluctuations are treated perturbatively (not to be confused with the perturbative quantum field theory treatment in sec. 4). However, we do not actually require an explicit solution to (43); it suffices to obtain expressions for $C_{xx}$, $C_{\phi\phi}$, and $C_{\varphi\varphi}$ in order to express the equation for $G_{hh}(\tau)$ in a more tractable form.

To that end, observe that $h_i(t_1)$, $h_i(t_2)$ are Gaussian random variables with covariance matrix $G_{hh}(\tau)$. That is, dropping the individual neuron indices $i$ and adopting the shorthand $h(t_{1,2}) =: h_{1,2}$, these are drawn from the bivariate normal distribution with

$$(h_1, h_2) \sim \mathcal{N}\left(0, \begin{pmatrix} G_{hh}(0) & G_{hh}(\tau) \\ G_{hh}(\tau) & G_{hh}(0) \end{pmatrix}\right) = \mathcal{N}\left(0, \begin{pmatrix} c_0 & c_\tau \\ c_\tau & c_0 \end{pmatrix}\right), \quad (44)$$

where in the second equality we have introduced the compact notation $c_0 := G_{hh}(0) = \langle h(t)h(t) \rangle$, $c_\tau := G_{hh}(\tau) = \langle h(t_1)h(t_2) \rangle$ (for $t_1 \neq t_2$). Furthermore, $C_{\phi\phi}$ is simply the expectation value of a particular function of these variables, namely $\phi(h_1)\phi(h_2)$, with respect to this distribution:

$$C_{\phi\phi}(t_1, t_2) = \langle \phi(h_1)\phi(h_2) \rangle_h = \int \mathcal{D}h_1 \mathcal{D}h_2 \, \phi(h_1)\phi(h_2), \quad (45)$$

where $\mathcal{D}h_1 \mathcal{D}h_2$ is the bivariate normal measure obtained by diagonalizing (44),

$$\mathcal{D}h_1 \mathcal{D}h_2 = \frac{dh_1 \, dh_2}{2\pi\sqrt{c_0^2(1-\rho^2)}} \exp\left[-\frac{1}{2c_0(1-\rho^2)}\left(h_1^2 + h_2^2 - 2\rho \, h_1 h_2\right)\right], \quad (46)$$

where the Pearson correlation coefficient is defined as $\rho := c_\tau / c_0$. If we then define the new integration variables $h_a$, $h_b$ such that

$$h_1 := \sqrt{c_0} \, h_a, \qquad h_2 := \sqrt{c_0}\left(\rho \, h_a + \sqrt{1-\rho^2} \, h_b\right), \quad (47)$$

then we can express (45) as

$$C_{\phi\phi} = \int \mathcal{D}h_a \mathcal{D}h_b \, \phi\left(\sqrt{c_0} \, h_a\right) \phi\left(\sqrt{c_0}(\rho \, h_a + \sqrt{1-\rho^2} \, h_b)\right), \quad (48)$$

where $\mathcal{D}h_a \mathcal{D}h_b$ is the (factorized) standard Gaussian measure,

$$\mathcal{D}h_a \mathcal{D}h_b = \frac{dh_a \, dh_b}{2\pi} e^{-\frac{1}{2}(h_a^2 + h_b^2)}. \quad (49)$$

By Price's theorem [54] for Gaussian processes,[20] we may express this as

$$C_{\phi\phi} = \frac{\partial}{\partial c_\tau} C_{\Phi\Phi}, \qquad \Phi(x) := \int_0^x dy \, \phi(y). \quad (50)$$

---

[20] Perhaps the easiest way to see this is to work with the original bivariate measure (46), which satisfies $\frac{\partial}{\partial c} \mathcal{D}h_1 \mathcal{D}h_2 = \frac{\partial^2}{\partial h_1 \partial h_2} \mathcal{D}h_1 \mathcal{D}h_2$. Since the fall-off of the Gaussian measure ensures vanishing boundary terms, integration by parts can then be used to move the derivatives to $\frac{\partial}{\partial h_1} \Phi(h_1) \frac{\partial}{\partial h_2} \Phi(h_2)$. Since (50) cannot depend on the choice of integration variables, this proves the theorem for $h_a, h_b$ as well. See also [55] theorem 6.7.

The motivation for this is that it allows us to define the *potential*

$$V(c_\tau, c_0) := -\frac{1}{2}\big(\gamma^2 - \sigma_A^2\big)c_\tau^2 + \Big[\sigma_b^2 + \sigma_B^2 C_{xx}(\tau) + \sigma_u^2 C_{\varphi\varphi}\Big]c_\tau + \sigma_w^2 C_{\Phi\Phi}(c_\tau, c_0), \qquad (51)$$

in terms of which we can express the differential equation (43) for the autocorrelation $c_\tau$ as

$$\partial_\tau^2 c_\tau = -V'(c_\tau, c_0) - \kappa g_\tau^2 \delta(\tau), \qquad (52)$$

where $g_\tau := g(\tau)$, and the prime denotes differentiation with respect to $c_\tau$. This is now a self-consistent[21] kinematic expression for $c_\tau$, where the left-hand side is a time-derivative of the kinetic term, hence the identification of $V$ as the potential energy. In this language, the delta function enforces a discontinuity in the velocity at $\tau = 0$. As alluded above, we do not require an explicit solution to this expression, but the form will prove convenient below.

## 3 The edge of chaos

In the previous section, we obtained a second-order differential equation for the two-point correlator $G_{hh}(\tau)$, (43). With this in hand, we now wish to determine where the edge of chaos lies in phase space. There are at least two ways of proceeding: one is to explicitly solve for $G_{hh}(\tau)$ by making some assumptions about the various correlators on the right-hand side. We will do this in the perturbative expansion in sec. 4, which then allows us to identify the correlation length in the presence of finite-width corrections. However, if one is content with MFT at $N \to \infty$, then there is an alternative way forwards that does not require a solution for $G_{hh}(\tau)$ at all: the basic idea, as pioneered in, e.g., [37], is to examine the response of the system to fluctuations, whose stability is governed by the largest Lyapunov exponent. Following [36], our strategy is to construct a double-copy of the mean-field theory obtained above, and consider the correlator between neurons in different copies as an infinitesimal fluctuation about the MFT correlator between neurons in the same copy. At the most basic level, the idea is to fix $h(0)$ to be the same in both copies – which have initially identical parameters – and examine how $h(t)$ (or rather, $G_{hh}$) differs as a function of time; a similar strategy was used in [17,29]. A formal parallel with the time-independent Schrödinger equation emerges, which relates the largest Lyapunov exponent to the ground-state energy of the system; as we will see, the onset of chaos corresponds to the point at which the ground-state energy becomes negative.

### 3.1 The double-copy system

Denote the MFT correlator $\langle h_i(t_1)h_i(t_2)\rangle = G_{hh}(t_1 - t_2) = G_{hh}(\tau) =: c(\tau)$, and extend this notation to two identical copies of the system in the obvious way, namely $c^{\alpha\beta}(t_1, t_2) := \langle h_i^\alpha(t_1)h_i^\beta(t_2)\rangle$, where $\alpha, \beta$ label the copies (so that $\alpha = \beta$ reduces to the single-copy case considered above). Note that at large $N$,

$$c^{\alpha\beta}(t_1, t_2) = \frac{1}{N}\sum_{i=1}^N h_i^\alpha(t_1)h_i^\beta(t_2), \qquad (53)$$

as a consequence of the self-averaging behavior discussed above. We may then consider the mean-squared distance between two trajectories (i.e., between two identically-prepared copies of the system):

$$d(t_1, t_2) = \frac{1}{N}||h_i^1(t_1) - h_i^2(t_2)||^2 = \frac{1}{N}\sum_{i=1}^N \big(h_i^1(t_1) - h_i^2(t_2)\big)^2$$

$$= c^{11}(\tau) + c^{22}(\tau) - c^{12}(t_1, t_2) - c^{21}(t_1, t_2). \qquad (54)$$

---

[21]Insofar as the value $c_0$ determines the potential via (51).

Note that in general $c^{12} \neq c^{21}$; only at equal times do we have $c^{12}(0) = c^{21}(0)$, so that $d(t,t)$ reduces to the usual mean-squared distance. Our goal in this section is to understand the dynamics of $d(t_1, t_2)$. In particular, we expect that if the system is chaotic, $d(t_1, t_2)$ should have a characteristic divergence governed by the largest Lyapunov exponent.

Since the copies are initially uncoupled, we may immediately write down the partition function analogous to (9)

$$
Z[j^1, j^2] = \prod_{\alpha=1}^{2} \int \mathcal{D}h^\alpha \int \mathcal{D}\tilde{z}^\alpha
$$

$$
\times \exp\left\{ \int dt\, \tilde{z}_i^\alpha(t) \left[ (\partial_t + \gamma) h_i^\alpha(t) - A_{ij} h_j^\alpha(t) - B_{ij} x_j^\alpha(t) - W_{ij} \phi(h_j^\alpha(t)) - U_{ij} \varphi(x_j^\alpha(t)) - b_i \right] \right\}
$$

$$
\times \exp\left\{ \frac{\kappa}{2} \int dt \left( \tilde{z}^1(t) + \tilde{z}^2(t) \right)^2 g(t)^2 \right\},
$$

(55)

where we have suppressed the source terms $j^\alpha(t) h^\alpha(t)$ for brevity, and the term on the last line is the contribution from the common stochasticity, $K_S\left(-g \sum_\alpha \tilde{z}^\alpha\right)$. We now integrate over the (zero-mean) disorder as before, cf. (22) with a sum over copies:

$$
\left\langle e^{-W_{ij} \int dt \sum_\alpha \tilde{z}_i^\alpha \phi(h_j^\alpha)} \right\rangle_W
$$

$$
= \exp \sum_{i,j} \left\{ \frac{\sigma_w^2}{2N} \int dt_1\, dt_2 \Big[ \sum_\alpha \tilde{z}_i^\alpha(t_1) \tilde{z}_i^\alpha(t_2) \phi(h_j^\alpha(t_1)) \phi(h_j^\alpha(t_2)) \right.
$$

$$
\left. + 2\tilde{z}_i^1(t_1) \tilde{z}_i^2(t_2) \phi(h_j^1(t_1)) \phi(h_j^2(t_2)) \Big] \right\}.
$$

(56)

Relative to (22), the novelty lies in the last term, in which we see that an effective interaction between the copies has arisen from marginalizing over the shared disorder.[22] We obtain a formally identical expression for the expectation value with respect to $A, B, U$, and $b$. Thus, in place of (23), we now have, for each $i$,

$$
S_{\text{int},i} = \frac{1}{2N} \int dt_1\, dt_2 \Bigg\{ \sum_\alpha \tilde{z}_i^\alpha(t_1) \tilde{z}_i^\alpha(t_2) \sum_j \Big[ \sigma_b^2 + \sigma_A^2 h_j^\alpha(t_1), h_j^\alpha(t_2) + \sigma_B^2 x_j^\alpha(t_1) x_j^\alpha(t_2) \right.
$$

$$
+ \sigma_w^2 \phi(h_j^\alpha(t_1)) \phi(h_j^\alpha(t_2)) + \sigma_u^2 \varphi(x_j^\alpha(t_1)) \varphi(x_j^\alpha(t_2)) \Big]
$$

$$
+ 2 \sum_j \tilde{z}_i^1(t_1) \tilde{z}_i^2(t_2) \Big[ \sigma_b^2 + \sigma_A^2 h_j^1(t_1) h_j^2(t_2) + \sigma_B^2 x_j^1(t_1) x_j^2(t_2) \right.
$$

$$
\left. + \sigma_w^2 \phi(h_j^1(t_1)) \phi(h_j^2(t_2)) + \sigma_u^2 \varphi(x_j^1(t_1)) \varphi(x_j^2(t_2)) \Big] \Bigg\}.
$$

(57)

We now proceed as above, adding labels to the auxiliary variables $\mathfrak{X} \to \mathfrak{X}^{\alpha\beta}$ in (24) in order

---

[22]Strictly speaking, we should write this as $\tilde{z}^1(t_1) \tilde{z}^2(t_2) \left[ \phi(h^1(t_1)) \phi(h^2(t_2)) + \phi(h^1(t_2)) \phi(h^2(t_1)) \right]$, since while the product $h^1(t) h^2(s)$ is commutative, $h^1(t) h^2(s) \neq h^1(s) h^2(t)$. But this is implicitly covered below when summing over $\alpha, \beta$, since $\mathfrak{W}^{12}(t,s) = \mathfrak{W}^{21}(s,t)$. (That is, one can freely swap *both* the labels and the times, but not either individually).

to accommodate the additional mixed fields with $\alpha \neq \beta$, i.e.,

$$\mathfrak{A}^{\alpha\beta}(t_1, t_2) := \frac{\sigma_A^2}{N} \sum_j h_j^\alpha(t_1) h_j^\beta(t_2), \qquad \mathfrak{B}^{\alpha\beta}(t_1, t_2) := \frac{\sigma_B^2}{N} \sum_j x_j^\alpha(t_1) x_j^\beta(t_2),$$

$$\mathfrak{W}^{\alpha\beta}(t_1, t_2) := \frac{\sigma_w^2}{N} \sum_j \phi(h_j^\alpha(t_1)) \phi(h_j^\beta(t_2)), \qquad \mathfrak{U}^{\alpha\beta}(t_1, t_2) := \frac{\sigma_u^2}{N} \sum_j \varphi(x_j^\alpha(t_1)) \varphi(x_j^\beta(t_2)).$$

$$(58)$$

We then elevate these to fluctuating field variables by introducing delta functions to impose the constraints, thereby obtaining the average partition function (with sources suppressed):[23]

$$\bar{Z} = \prod_{\alpha=1}^{2} \left( \int \mathcal{D}\mathfrak{X}^{\alpha\alpha} \mathcal{D}\tilde{X}^{\alpha\alpha} \right) \int \mathcal{D}\mathfrak{X}^{12} \mathcal{D}\tilde{X}^{12}$$

$$\times \exp\left\{ \int dt_1 dt_2 \sum_{(\alpha,\beta) \in P} \left[ -\frac{N}{\sigma_A^2} \tilde{A}^{\alpha\beta}(t_1, t_2) \mathfrak{A}^{\alpha\beta}(t_1, t_2) - \frac{N}{\sigma_B^2} \tilde{B}^{\alpha\beta}(t_1, t_2) \mathfrak{B}^{\alpha\beta}(t_1, t_2) \right. \right.$$

$$\left. -\frac{N}{\sigma_w^2} \tilde{W}^{\alpha\beta}(t_1, t_2) \mathfrak{W}^{\alpha\beta}(t_1, t_2) - \frac{N}{\sigma_u^2} \tilde{U}^{\alpha\beta}(t_1, t_2) \mathfrak{U}^{\alpha\beta}(t_1, t_2) \right]$$

$$\left. +N \ln \bar{Z}_i[j^\alpha, \tilde{j}^\alpha] \right\}, \tag{59}$$

where in place of (28), we now have

$$\bar{Z}_i[j^\alpha, \tilde{j}^\alpha] := \prod_{\alpha=1}^{2} \left( \int \mathcal{D}h_i^\alpha \int \mathcal{D}\tilde{z}_i^\alpha \right) \exp\left\{ S_0[h_i^\alpha, \tilde{z}_i^\alpha] + S_{\text{source}}[j^\alpha, \tilde{j}^\alpha] \right\}$$

$$\times \exp\left\{ \int dt_1 dt_2 \left( \frac{1}{2} \sum_{\alpha,\beta=1}^{2} \left[ \sigma_b^2 + \mathfrak{A}^{\alpha\beta}(t_1, t_2) + \mathfrak{B}^{\alpha\beta}(t_1, t_2) + \mathfrak{W}^{\alpha\beta}(t_1, t_2) \right. \right. \right.$$

$$\left. + \mathfrak{U}^{\alpha\beta}(t_1, t_2) \right] \tilde{z}_i^\alpha(t_1) \tilde{z}_i^\beta(t_2)$$

$$+ \sum_{(\alpha,\beta) \in P} \left[ \tilde{A}^{\alpha\beta}(t_1, t_2) h_i^\alpha(t_1) h_i^\beta(t_2) + \tilde{B}^{\alpha\beta}(t_1, t_2) x_i^\alpha(t_1) x_i^\beta(t_2) \right.$$

$$\left. \left. \left. + \tilde{W}^{\alpha\beta}(t_1, t_2) \phi(h_i^\alpha(t_1)) \phi(h_i^\beta(t_2)) + \tilde{U}^{\alpha\beta}(t_1, t_2) \varphi(x_i^\alpha(t_1)) \varphi(x_i^\beta(t_2)) \right] \right) \right\}, \tag{60}$$

where $P := \{(1, 1), (2, 2), (1, 2)\}$.[24]

We now perform the saddlepoint approximation as before, and keep the leading-order contribution to obtain the mean-field result. The minimality condition yields the following

---

[23]Here $\mathfrak{X}, \tilde{X}$ are understood to run over all fields, i.e., $\int \mathcal{D}\mathfrak{X}^{\alpha\beta} := \int \mathcal{D}\mathfrak{A}^{\alpha\beta} \mathcal{D}\mathfrak{B}^{\alpha\beta} \mathcal{D}\mathfrak{W}^{\alpha\beta} \mathcal{D}\mathfrak{U}^{\alpha\beta}$.

[24]We exclude $(2, 1)$ because this would double-count equivalent terms from the delta-function constraint; this is in contrast to the first sum in the exponential over all possible pairs, which accounts for the fact that in (57), the cross-term $\tilde{z}^1 \tilde{z}^2$ comes with a factor of 2 relative to $\tilde{z}^\alpha \tilde{z}^\alpha$.

equations of motion for the auxilliary fields:

$$
\begin{aligned}
\mathfrak{A}_0^{\alpha\beta}(t_1, t_2) &= \sigma_A^2 \langle h_i^\alpha(t_1) h_i^\beta(t_2) \rangle_{X_0} =: \sigma_A^2 C_{hh}^{\alpha\beta}(t_1, t_2), \\
\mathfrak{B}_0^{\alpha\beta}(t_1, t_2) &= \sigma_B^2 \langle x_i^\alpha(t_1) x_i^\beta(t_2) \rangle_{X_0} =: \sigma_B^2 C_{xx}^{\alpha\beta}(t_1, t_2), \\
\mathfrak{W}_0^{\alpha\beta}(t_1, t_2) &= \sigma_w^2 \langle \phi(h_i^\alpha(t_1)) \phi(h_i^\beta(t_2)) \rangle_{X_0} =: \sigma_w^2 C_{\phi\phi}^{\alpha\beta}(t_1, t_2), \\
\mathfrak{U}_0^{\alpha\beta}(t_1, t_2) &= \sigma_u^2 \langle \varphi(x_i^\alpha(t_1)) \varphi(x_i^\beta(t_2)) \rangle_{X_0} =: \sigma_u^2 C_{\varphi\varphi}^{\alpha\beta}(t_1, t_2), \\
\tilde{A}_0^{\alpha\beta}(t_1, t_2) &\simeq \sigma_A^2 \langle \tilde{z}_i^\alpha(t_1) \tilde{z}_i^\beta(t_2) \rangle_{X_0} = 0, \\
\tilde{B}_0^{\alpha\beta}(t_1, t_2) &\simeq \sigma_B^2 \langle \tilde{z}_i^\alpha(t_1) \tilde{z}_i^\beta(t_2) \rangle_{X_0} = 0, \\
\tilde{W}_0^{\alpha\beta}(t_1, t_2) &\simeq \sigma_w^2 \langle \tilde{z}_i^\alpha(t_1) \tilde{z}_i^\beta(t_2) \rangle_{X_0} = 0, \\
\tilde{U}_0^{\alpha\beta}(t_1, t_2) &\simeq \sigma_u^2 \langle \tilde{z}_i^\alpha(t_1) \tilde{z}_i^\beta(t_2) \rangle_{X_0} = 0,
\end{aligned}
\tag{61}
$$

which is formally just two copies of (31), plus a set for the cross-correlators with $\alpha \neq \beta$; in the last four expressions, the constant of proportionality is $1/2$ for $\alpha = \beta$, and 1 otherwise. As in the single-copy case considered above, all self-correlations between auxiliary fields $\tilde{z}$ are zero due to the normalization of the partition function. Physically, this is again simply the statement that these auxiliary fields are non-propagating. Thus the only novel contribution is the appearance of cross-correlators between the two copies, denoted by $C^{12}$, which arise as a direct consequence of marginalizing over the shared disorder.

Substituting the minima (61) into the action, we obtain the MFT for two identical copies of the network:

$$
\begin{aligned}
\bar{Z}_{i,0} &= \prod_{\alpha=1}^2 \left( \int \mathcal{D}h_i^\alpha \int \mathcal{D}\tilde{z}_i^\alpha \right) \exp\Bigg\{ \int \mathrm{d}t \left[ \sum_\alpha \tilde{z}_i^\alpha(t)(\partial_t + \gamma) h_i^\alpha(t) + \frac{\kappa}{2} \left( \tilde{z}_i^1(t) + \tilde{z}_i^2(t) \right)^2 g_i(t)^2 \right] \\
&\quad + \frac{1}{2} \sum_{\alpha,\beta} \int \mathrm{d}t_1 \, \mathrm{d}t_2 \, \tilde{z}_i^\alpha(t_1) \Big[ \sigma_b^2 + \sigma_A^2 C_{hh}^{\alpha\beta}(t_1, t_2) + \sigma_B^2 C_{xx}^{\alpha\beta}(t_1, t_2) + \sigma_w^2 C_{\phi\phi}^{\alpha\beta}(t_1, t_2) \\
&\qquad\qquad\qquad\qquad + \sigma_u^2 C_{\varphi\varphi}^{\alpha\beta}(t_1, t_2) \Big] \tilde{z}_i^\beta(t_2) \Bigg\} \\
&= \prod_{\alpha=1}^2 \int \mathcal{D}h_i^\alpha \int \mathcal{D}\tilde{z}_i^\alpha \exp\left\{ -\frac{1}{2} \int \mathrm{d}t_1 \, \mathrm{d}t_2 \, y^\alpha(t_1)^{\mathrm{T}} \Xi^{\alpha\beta}(t_1, t_2) y^\beta(t_2) \right\},
\end{aligned}
\tag{62}
$$

where, in an extension of the single-copy notation (35), we have defined

$$
y_i^\alpha(t) := \begin{pmatrix} h_i^\alpha(t) \\ \tilde{z}_i^\alpha(t) \end{pmatrix}, \qquad
\Xi^{\alpha\beta}(t_1, t_2) = \begin{pmatrix} \Xi_{hh}^{\alpha\beta}(t_1, t_2) & \Xi_{h\tilde{z}}^{\alpha\beta}(t_1, t_2) \\ \Xi_{\tilde{z}h}^{\alpha\beta}(t_1, t_2) & \Xi_{\tilde{z}\tilde{z}}^{\alpha\beta}(t_1, t_2) \end{pmatrix},
\tag{63}
$$

with the operator elements

$$
\begin{aligned}
\Xi_{hh}^{\alpha\beta}(t_1, t_2) &= 0, \\
\Xi_{h\tilde{z}}^{\alpha\beta}(t_1, t_2) &= \delta_{\alpha\beta} \delta(t_1 - t_2)(\partial_{t_2} - \gamma), \\
\Xi_{\tilde{z}h}^{\alpha\beta}(t_1, t_2) &= -\delta_{\alpha\beta} \delta(t_1 - t_2)(\partial_{t_2} + \gamma), \\
\Xi_{\tilde{z}\tilde{z}}^{\alpha\beta}(t_1, t_2) &= -\sigma_b^2 - \sigma_A^2 C_{hh}^{\alpha\beta}(t_1, t_2) - \sigma_B^2 C_{xx}^{\alpha\beta}(t_1, t_2) - \sigma_w^2 C_{\phi\phi}^{\alpha\beta}(t_1, t_2) - \sigma_u^2 C_{\varphi\varphi}^{\alpha\beta}(t_1, t_2) \\
&\quad - \kappa \delta(t_1 - t_2) g(t_2)^2.
\end{aligned}
\tag{64}
$$

We now seek the propagators $G^{\alpha\beta}(t_1, t_2)$ of the double-copy MFT, defined as the matrix elements of the right-inverse of $\Xi^{\alpha\beta}(t_1, t_2)$:

$$\sum_\mu \int ds\, \Xi^{\alpha\mu}(t_1, s) G^{\mu\beta}(s, t_2) = \delta_{\alpha\beta}\delta(t_1 - t_2)\mathbb{1}. \tag{65}$$

This yields the following three equations for the non-vanishing components of $G$:

$$\delta_{\alpha\beta}\delta(t_1 - t_2) = (\partial_1 - \gamma)\, G^{\alpha\beta}_{\tilde{z}h}(t_1, t_2),$$
$$\delta_{\alpha\beta}\delta(t_1 - t_2) = -(\partial_1 + \gamma)\, G^{\alpha\beta}_{h\tilde{z}}\, G(t_1, t_2),$$
$$0 = (\partial_1 + \gamma)\, G^{\alpha\beta}_{hh}(t_1, t_2) + \sum_\mu \int ds \Big[ \sigma_b^2 + \sigma_A^2 C^{\alpha\mu}_{hh}(t_1, s) + \sigma_B^2 C^{\alpha\mu}_{xx}(t_1, s) \tag{66}$$
$$+ \sigma_w^2 C^{\alpha\mu}_{\phi\phi}(t_1, s) + \sigma_u^2 C^{\alpha\mu}_{\varphi\varphi}(t_1, s) + \kappa\delta(t_1 - s)g(s)^2 \Big] G^{\mu\beta}_{\tilde{z}h}(s, t_2).$$

Note that $C^{\alpha\beta}_{hh} \equiv G^{\alpha\beta}_{hh}$, cf. the comment below (39) above. Assuming time-translation invariance[25] then allows us to perform the same trick as in the single-copy case: swapping $\partial_1$ for $-\partial_2$ in the first equation, and multiplying the third by $(\partial_2 + \gamma)$, we obtain

$$(\partial_1 + \gamma)(\partial_2 + \gamma)\, G^{\alpha\beta}_{hh}(t_1, t_2)$$
$$= -\sum_\mu \int ds \Big[ \sigma_b^2 + \sigma_A^2 G^{\alpha\mu}_{hh}(t_1, s) + \sigma_B^2 C^{\alpha\mu}_{xx}(t_1, s) + \sigma_w^2 C^{\alpha\mu}_{\phi\phi}(t_1, s) + \sigma_u^2 C^{\alpha\mu}_{\varphi\varphi}(t_1, s)$$
$$+ \kappa\delta(t_1 - s)g(s)^2 \Big](\partial_2 + \gamma)\, G^{\mu\beta}_{\tilde{z}h}(s, t_2)$$
$$= \sum_\mu \int ds \Big[ \sigma_b^2 + \sigma_A^2 G^{\alpha\mu}_{hh}(t_1, s) + \sigma_B^2 C^{\alpha\mu}_{xx}(t_1, s) + \sigma_w^2 C^{\alpha\mu}_{\phi\phi}(t_1, s) + \sigma_u^2 C^{\alpha\mu}_{\varphi\varphi}(t_1, s) \tag{67}$$
$$+ \kappa\delta(t_1 - s)g(s)^2 \Big]\delta_{\mu\beta}\delta(s - t)$$
$$= \sigma_b^2 + \sigma_A^2 G^{\alpha\beta}_{hh}(t_1, t_2) + \sigma_B^2 C^{\alpha\beta}_{xx}(t_1, t_2) + \sigma_w^2 C^{\alpha\beta}_{\phi\phi}(t_1, t_2) + \sigma_u^2 C^{\alpha\beta}_{\varphi\varphi}(t_1, t_2)$$
$$+ \kappa\delta(t_1 - t_2)g(t_2)^2.$$

Formally, this is the same result we obtained in the single-copy case, cf. (41); the difference lies in the sum over all $\alpha, \beta$ in the action (62), which picks up cross-correlations between the copies. As before, it is convenient to express these equations for the components of $G$ in terms of the temporal difference $\tau := t_1 - t_2$ (where possible), so that the off-diagonal elements are given by

$$(\partial_\tau - \gamma)\, G^{\alpha\beta}_{\tilde{z}h}(\tau) = (-\partial_\tau - \gamma)\, G^{\alpha\beta}_{h\tilde{z}}(\tau) = \delta_{\alpha\beta}\delta(\tau), \tag{68}$$

while the non-zero diagonal element becomes[26]

$$\big[(\partial_1 + \gamma)(\partial_2 + \gamma) - \sigma_A^2\big] G^{\alpha\beta}_{hh}(t_1, t_2) = \sigma_b^2 + \sigma_B^2 C^{\alpha\beta}_{xx}(\tau) + \sigma_w^2 C^{\alpha\beta}_{\phi\phi}(\tau) + \sigma_u^2 C^{\alpha\beta}_{\varphi\varphi}(\tau) + \kappa\delta(\tau)g(\tau)^2. \tag{69}$$

Observe that the $\alpha = \beta$ components of this equation are precisely (43), as expected, and hence the solution for the autocorrelation $G^{\alpha\alpha}_{hh} = c^{\alpha\alpha}(\tau)$ is the same as for the single-copy case $c(\tau)$ above. We may say that in this case, the system resides at a *fixed point* at which the trajectories $h^\alpha(t) = h^\beta(t)$ for all time, and hence $d(t, t) = 0\ \forall\ t$ (since $c^{11} = c^{22} = c^{12}$).

---

[25]Note that we do *not* assume this for the cross-correlator $G^{12}_{hh}$, but the Kronecker delta implies that $G^{12}_{\tilde{z}h} = G^{21}_{h\tilde{z}} = 0$.

[26]We note that this corresponds to eq. (167) of [36]; we have merely derived it via the standard approach.

Our interest is then in the stability of this fixed point. In particular, instability to fluctuations will cause $d(t_1, t_2)$ to grow, indicating chaotic dynamics. Conversely, a stable fixed point implies that the system eventually relaxes back to $d(t_1, t_2) = 0$, so that perfect correlation is restored. Following [36], our strategy will be to take the autocorrelation $c(\tau) = c^{11}(\tau) = c^{22}(\tau)$ – given implicitly by (69) with $\alpha = \beta$, which is equivalent to the single-copy solution (43) – as a fixed background solution, and compute the cross-correlator $c^{12}(t,s) := G_{hh}^{12}(t,s)$ to linear order in the expansion[27]

$$c^{12}(t,s) = c(\tau) + \eta\, k(t,s) + O(\eta^2), \tag{70}$$

where $\eta \ll 1$ is some small expansion parameter.

To proceed, it is convenient to introduce the following notation for the two-point correlators of $\phi(h)$:

$$f_\phi(c^{12}, c_0) := C_{\phi\phi}^{12} \equiv \langle \phi(h_i^1)\phi(h_i^2) \rangle, \qquad f_\phi(c, c_0) := C_{\phi\phi}^{11} = C_{\phi\phi}^{22} \equiv \langle \phi(h_i)\phi(h_i) \rangle, \tag{71}$$

where $c_0$, $c \equiv c_\tau$ are the components of the covariance matrix of the single-copy in (44), and $c_0 \equiv c_0^{11} = c_0^{22}$, $c^{12}$ are the corresponding components for the double-copy:[28]

$$(h^1(t), h^2(s)) \sim \mathcal{N}\left(0, \begin{pmatrix} G_{hh}^{11}(0) & G_{hh}^{12}(t,s) \\ G_{hh}^{12}(t,s) & G_{hh}^{22}(0) \end{pmatrix}\right) = \mathcal{N}\left(0, \begin{pmatrix} c_0 & c^{12} \\ c^{12} & c_0 \end{pmatrix}\right). \tag{72}$$

We can then Taylor expand (69) and identify terms order by order to determine the linear contribution $k(t,s)$. On the left-hand side, we simply substitute (70) for $G_{hh}^{12} = c^{12}$. On the right-hand side, we take the input data to be the same for both copies, i.e., $x^1 = x^2$, so that $C_{xx}^{12} = C_{xx}$ and $C_{\varphi\varphi}^{12} = C_{\varphi\varphi}$; then we only need to expand $C_{\phi\phi}^{12}$:

$$\begin{aligned} f_\phi(c^{12}, c_0) &= f_\phi(c, c_0) + \left(c^{12} - c\right)\frac{\partial}{\partial c^{12}}f_\phi(c^{12}, c_0)\Big|_{c^{12}=c} + \ldots \\ &= f_\phi(c, c_0) + \eta\, k(t,s)f_{\phi'}(c, c_0) + \mathcal{O}(\eta^2), \end{aligned} \tag{73}$$

where the second line follows via Price's theorem (50), with $\phi' = \partial_h \phi(h)$. Upon substituting these expansions into (69), we recover (43) at $\mathcal{O}(1)$, which leaves

$$\left[(\partial_1 + \gamma)(\partial_2 + \gamma) - \sigma_A^2\right]k(t,s) = \sigma_w^2 f_{\phi'}(c, c_0)\, k(t,s) \tag{74}$$

at order $\mathcal{O}(\eta)$.

## 3.2 The largest Lyapunov exponent

We have found that at linear order in fluctuations about the fixed point, the mean-squared distance between copies (54) is

$$\begin{aligned} d(t,t) &= c^{11}(0) + c^{22}(0) - c^{12}(t,t) - c^{21}(t,t) \\ &= 2c_0 - 2(c_0 + \eta k(t,t)) = -2\eta\, k(t,t), \end{aligned} \tag{75}$$

with $k$ given by (74). We now wish to solve (74) in order to determine how the distance behaves as a function of time.

Since (74) is precisely of the same form as that in [36], we may simply rephrase their derivation here. We begin by expressing (74) in terms of the "lightcone coordinates"

$$\tau = t - s, \qquad u = t + s, \tag{76}$$

---

[27]We have changed to $t,s$ rather than $t_1, t_2$ to avoid confusion with the copy labels $\alpha = 1, 2$, so that henceforth $\tau = t - s$.

[28]Note that the lower off-diagonal component is $G^{21}(s,t) = G^{12}(t,s)$.

with the derivative relations

$$\partial_t = \partial_\tau + \partial_u, \qquad \partial_s = -\partial_\tau + \partial_u, \tag{77}$$

whence

$$[(\partial_u + \gamma)^2 - \partial_\tau^2 - \sigma_A^2]k(\tau, u) = \sigma_w^2 f_{\phi'}(c, c_0)k(\tau, u). \tag{78}$$

To solve this equation, we make the separation ansatz

$$k(\tau, u) = e^{\lambda u}\psi(\tau). \tag{79}$$

One can think of this as a Euclidean analogue of the usual plane-wave ansatz with phase factor $e^{-i\lambda u}$; the significance of the constant $\lambda$ will be discussed below. We then have

$$[(\lambda + \gamma)^2 - \partial_\tau^2 - \sigma_A^2]\psi(\tau) = \sigma_w^2 f_{\phi'}(c, c_0)\psi(\tau)$$
$$\implies [-\partial_\tau^2 - V''(c, c_0)]\psi(\tau) = E\psi(\tau), \tag{80}$$

where we have defined the energy

$$E := \gamma^2 - (\lambda + \gamma)^2, \tag{81}$$

and $V'' = \partial_c^2 V$ is the second derivative of the potential defined in (51):[29]

$$V''(c, c_0) = -\gamma^2 + \sigma_A^2 + \sigma_w^2 f_{\phi'}(c, c_0). \tag{82}$$

Note $f_{\phi'}$, and hence the potential term $V''$, depend on $\tau$ via the autocorrelation $c = c(\tau)$.

Formally, (80) is the time-independent Schrödinger equation $H\psi = E\psi$ with Hamiltonian

$$H = -\partial_\tau^2 - V''(c, c_0). \tag{83}$$

Since we seek normalizable (i.e., bound) states, and the energies of bound states are quantized, the solution $\psi(\tau)$ will be characterized by a discrete set of eigenvalues,

$$E_n = \gamma^2 - (\lambda_n + \gamma)^2. \tag{84}$$

As per our ansatz (79), this in turn implies a discretized set of characteristic or *Lyapunov exponents* $\lambda_n$ which control the growth of $k(\tau, u)$. In particular, the growth rate will be governed by the largest Lyapunov exponent, given by the ground state energy $E_0$:

$$\lambda_0 = -\gamma + \sqrt{\gamma^2 - E_0}. \tag{85}$$

The unstable regime is characterized by $\lambda_0 > 0$, so that $d(t_1, t_2)$ grows exponentially with time. Note that if $\gamma < 0$, this will be true for all possible values of the ground state energy $E_0$. Therefore, in order to study networks at the edge of stability, we will henceforth consider the case with $\gamma > 0$ (cf. [36], which considered $\gamma = 1$). We must then determine the point at which the ground state energy becomes negative, $E_0 < 0$. Following [36], our strategy will be to construct a normalizable solution with $E = 0$, and derive a necessary condition on the existence of lower-energy ground states.

To proceed, observe that if we differentiate (52) with respect to $\tau$ for $\tau \neq 0$, and denote $\partial_\tau c_\tau = \dot{c}_\tau$, we obtain

$$\partial_\tau^2 \dot{c}_\tau + \partial_\tau V'(c_\tau, c_0) = [\partial_\tau^2 + V''(c_\tau, c_0)]\dot{c}_\tau = 0, \tag{86}$$

---

[29]Note that $V$ is the potential energy for the autocorrelation $c(\tau)$, while $-V''$ is the potential energy for the fluctuation amplitude $\psi(\tau)$.

where the second step follows from applying the chain rule to $V'$. Comparing this with the Schrödinger equation (80), we see that $\dot{c}(\tau)$ is an eigensolution of $\psi(\tau)$ everywhere except at $\tau = 0$, with $E = 0$. To construct a valid solution for all $\tau$, we must address the discontinuity at the origin caused by the delta function in (52). To do so, define

$$y(\tau) = \begin{cases} \dot{c}(\tau) & \tau > 0 \\ -\dot{c}(\tau) & \tau < 0 \end{cases}, \tag{87}$$

and impose smoothness at $\tau = 0$ (i.e., that $y$ belong to differentiability class $C^1$). Denoting the approach to zero from above and below respectively by $0^{\pm}$, the latter condition amounts to the constraint that $\dot{y}(0^+) - \dot{y}(0^-) = 0$:

$$\begin{aligned} \dot{y}(0^+) - \dot{y}(0^-) &= \ddot{c}(0^+) + \ddot{c}(0^-) = -2V'(c_0, c_0) \\ &= -2\left[-\left(\gamma^2 - \sigma_A^2\right)c_0 + \sigma_w^2 f_\phi(c_0, c_0) + \sigma_b^2 + \sigma_B^2 C_{xx}(\tau) + \sigma_u^2 C_{\varphi\varphi}\right] \overset{!}{=} 0. \end{aligned} \tag{88}$$

Therefore, the existence of such a solution requires

$$V'(c_0, c_0) = -\left(\gamma^2 - \sigma_A^2\right)c_0 + \sigma_w^2 f_\phi(c_0, c_0) + \sigma_b^2 + \sigma_B^2 C_{xx}(\tau) + \sigma_u^2 C_{\varphi\varphi} = 0. \tag{89}$$

This is a necessary but not sufficient condition for the solution to be valid: since $y(\tau)$ has zero total energy by construction, we must also impose that the potential be less than or equal to zero at the extremum given by (89); see [36] for an in-depth discussion on this point. Recall from (83) or footnote 29 that the potential for this solution is $-V''$. Hence,

$$V''(c_0, c_0) \geq 0 \implies \sigma_w^2 f_{\phi'}(c_0, c_0) \geq \gamma^2 - \sigma_A^2. \tag{90}$$

When this inequality is saturated, $y(\tau)$ must be the ground state with $E_0 = 0$. Away from saturation, the energy of the ground state is $E_0 \leq 0$. Intuitively, as the potential becomes more negative, there is more room for lower-energy states. While this does not technically suffice to show the existence of a strictly negative energy ground state, it does provide a necessary condition on the edge of stability for the system, since the corresponding Lyapunov exponent is greater than or equal to zero, $\lambda_0 \geq 0$. In fact, this is consistent with the condition identified in previous literature: recalling the definition of the two-point correlator $f_\phi$ (cf. (71)) in terms of the Gaussian integral (48) with $c_{\tau=0}$ (i.e., $\rho = 1$) we may equivalently express (90) in the form

$$\sigma_w^2 \int \mathcal{D}h_a \, \phi'\left(\sqrt{c_0} h_a\right)^2 \geq \gamma^2 - \sigma_A^2, \tag{91}$$

where the second integral over $h_b$ has evaluated to unity. Note that the quantity on the left-hand side is exactly that denoted $\chi_1$ in [17, 29] and serves as a probe of stability for the fixed-point $\rho = 1$ in their analysis. Comparing this expression with eq. (7) in [29] (which is eq. (5) in [17]), we see that our result precisely recovers the condition for chaos in Gaussian random networks, which corresponds to $\gamma = 1$ and $\sigma_A^2 = 0$, cf. (2). However, as alluded above, and will be discussed in more detail below, mean-field theory is not exact in the infinite-width limit; indeed, it is perhaps surprising that the MFT result (91) agrees with the result from the CLT obtained in the aforementioned works. In general, we must consider the full QFT, in which the MFT correlator $c(\tau)$ represents only the tree-level contribution.

Before turning to perturbative QFT in the next section, we note that in [31] it was reported that in the case of RNNs, the injection of time-series data[30] $x$ destroys the ordered phase, and consequently there is no order-to-chaos phase transition. This arises due to an extra factor that

---

[30]The authors of [31] refer to this as "noise" from the inputs, but we have reserved that term for the true noise encoded in $g(h, x)$, cf. (1).

appears in their analogue of (91) containing possible correlations in $x$. In our case, however, these are contained in the correlator $C_{xx}(\tau)$, which does *not* affect the condition for criticality in the MFT approximation (though it does of course affect the explicit form of $c(\tau)$). Further studies are therefore needed to elucidate the potential role of the data $x$, or the introduction of other forms of noise more generally, in modifying the edge of chaos in different network models.

# 4 Perturbative corrections

As discussed in the introduction, the network becomes a Gaussian process in the infinite-width $(N \to \infty)$ limit. At finite $N$, deviations from Gaussianity require the addition of interaction terms, corresponding to the fact that higher cumulants no longer precisely vanish. In the language of quantum field theory, these correspond to loop corrections to the leading-order or tree-level result above, which have the potential to shift the "classical" edge of stability, which is the chief object of interest here. Indeed, empirical evidence [17,18] suggests that the critical point in real-world networks is noticeably displaced from the CLT prediction, which has practical relevance for initialization. Even away from criticality, the correlation length – which sets the depth scale beyond which trainability sharply falls off – deviates substantially from the large-$N$ result—see for example fig. 5 in [17], fig. 2 in [30], or fig. 6 in [18]. It is therefore of significant practical as well as theoretical interest to quantify the deviations from Gaussianity in networks of finite width.

To that end, in this section we will compute both the leading $\mathcal{O}(1)$ and subleading $\mathcal{O}(T/N)$ corrections to the two-point correlator $c(\tau) = \langle h(t)h(s) \rangle$, which will then allow us to identify the loop-corrected correlation length for small $|\tau|$. As discussed in the introduction and in more detail below, the result holds only at weak 't Hooft coupling,[31] and the perturbative expansion assumes $T/N < 1$; the latter is the regime of practical relevance for modern deep neural networks [2].[32] It would be interesting to explore these connections to $O(N)$ theory in more detail, e.g., to see whether the analysis can be extended beyond the perturbative (weak-coupling) regime we consider here.

Since we will compute the two-point function explicitly, there is no need to employ the double-copy system used in the MFT analysis in the previous section; it is sufficient to examine the correlation length in a single copy of the full QFT for small times. The reason for this can be traced back to (70), which requires that the fluctuation $\eta k(t,s) \sim \eta e^{\lambda T}$ be small relative to the background solution $c(\tau)$. In other words, we do not demand that the two systems will diverge for all time, or in the single-copy theory, that the correlator will grow exponentially without bound. In the language of RG, we are considering infinitesimal perturbations about the critical point by some relevant operator(s), which will trigger a flow to a new, non-trivial fixed point in parameter space. This behavior was observed in [29], see in particular fig. 2 therein, in which networks in the chaotic phase converge to some parameter-dependent fixed point near, but not necessarily at, $c(\tau) = 0$.

To keep the theory as simple as possible while still capturing the essential aspects, we shall consider the partition function for the (single-copy) theory with $A = B = [0]$. Relative to (24)

---

[31]In fact, we shall require both $\sigma_w$ and $\sigma_b$ to be sufficiently small relative to $\gamma$, as will be explained in more detail below.

[32]Note that the authors of [2] referred to $T/N \to \infty$ as the "chaotic limit", which differs from the use of that terminology here, but the important point is that the perturbative expansion cannot be performed if $T/N$ is large. Conversely, $T/N \to 0$ simply recovers the Gaussian theory, including the $\mathcal{O}(1)$ correction to be computed in subsec. 4.3.1. In this sense, the $\mathcal{O}(1)$ contributions are not *bona fide* interactions, but finite-temperature fluctuations in the statistical ensemble, as we explain in sec. 5.

however, it is extremely convenient to modify the auxiliary field to

$$\mathfrak{W}(t_1, t_2) := \sqrt{\frac{\sigma_w^2}{N}} \sum_i \phi(h_i(t_1))\phi(h_i(t_2)), \tag{92}$$

and swap the attachment of the prefactor within the delta function-imposition of the constraint, i.e.,

$$\delta\left(\mathfrak{W}(t_1, t_2) - \sqrt{\frac{\sigma_w^2}{N}} \sum_i \phi(h_i(t_1))\phi(h_i(t_2))\right), \tag{93}$$

and similarly for $\mathfrak{U}$. Additionally, we will henceforth keep the implicit sums over repeated neuron indices, rather than extracting an overall factor of $N$. The previous conventions facilitated the mean-field (i.e., saddlepoint) approximation, but the present conventions will enable us to cleanly organize the perturbative expansion in a manner precisely analogous to the $O(N)$ vector model and Yang-Mills theory. There, the auxiliary field has exactly the same scaling as $\mathfrak{W}$ in (92), with $g_{YM} = \sqrt{\lambda/N}$. We thus see that $\sigma_w^2$ plays the role of the 't Hooft coupling $\lambda$, which must be sufficiently small for the infinite series of loop contributions to converge. Our theory is then

$$\bar{Z} = \int \mathcal{D}\mathfrak{X}\, \mathcal{D}\tilde{X} \exp\left\{-\int dt_1\, dt_2 \left[\tilde{W}(t_1, t_2)\mathfrak{W}(t_1, t_2) + \tilde{U}(t_1, t_2)\mathfrak{U}(t_1, t_2)\right]\right\}$$

$$\times \int \mathcal{D}h\mathcal{D}\tilde{z} \exp\left\{\int dt\left[\tilde{z}_i(t)(\partial_t + \gamma)h_i(t) + \frac{\kappa}{2}g(t)^2\tilde{z}_i(t)\tilde{z}_i(t)\right]\right.$$

$$+ \frac{1}{2}\int dt_1\, dt_2 \left[\sigma_b^2 + \sqrt{\frac{\sigma_w^2}{N}}\mathfrak{W}(t_1, t_2) + \sqrt{\frac{\sigma_u^2}{N}}\mathfrak{U}(t_1, t_2)\right]\tilde{z}_i(t_1)\tilde{z}_i(t_2) \tag{94}$$

$$+ \int dt_1\, dt_2\left[\sqrt{\frac{\sigma_w^2}{N}}\tilde{W}(t_1, t_2)\phi(h_i(t_1))\phi(h_i(t_2)) + \sqrt{\frac{\sigma_u^2}{N}}\tilde{U}(t_1, t_2)\varphi(x_i(t_1))\varphi(x_i(t_2))\right]\right\},$$

where $\mathfrak{X} \in \{\mathfrak{W}, \mathfrak{U}\}$, and $\tilde{X} \in \{\tilde{W}, \tilde{U}\}$.

Now, the perturbative analysis proceeds by expanding around the vacuum expectation value (vev) of the fields (i.e., the "classical" background values), obtained as the solutions to the equations of motion (eom):[33]

$$h_i(t): \qquad (\partial_t - \gamma)\langle\bar{z}_i(t)\rangle = 2\sqrt{\frac{\sigma_w^2}{N}}\int ds\,\langle\tilde{W}(t,s)\delta_{ij}\phi'(h_j(t))\phi(h_j(s))\rangle, \tag{95}$$

---

[33]Note that in the first of these, the boundary term vanishes when performing integration by parts to move the derivative off the delta function; explicitly:

$$\frac{\delta \ln \bar{Z}}{\delta h_i(t)} = \left\langle\int ds\,\tilde{z}_j(s)(\partial_s + \gamma)\frac{\delta h_j(s)}{\delta h_i(t)}\right\rangle$$

$$+ \sqrt{\frac{\sigma_w^2}{N}}\left\langle\int ds_1\, ds_2\,\tilde{W}(s_1, s_2)\left[\delta_{ij}\delta(t - s_1)\phi'(h_j(s_1))\phi(h_j(s_2)) + \delta_{ij}\delta(t - s_2)\phi(h_j(s_1))\phi'(h_j(s_2))\right]\right\rangle$$

$$= \int ds\,\langle\bar{z}_i(s)\rangle(\partial_s + \gamma)\delta(t - s) + 2\sqrt{\frac{\sigma_w^2}{N}}\int ds\,\langle\tilde{W}\delta_{ij}\phi'(h_j(t))\phi(h_j(s))\rangle$$

$$= -\int ds\,\delta(t - s)(\partial_s - \gamma)\langle\bar{z}_i(s)\rangle + 2\sqrt{\frac{\sigma_w^2}{N}}\int ds\,\langle\tilde{W}\delta_{ij}\phi'(h_j(t))\phi(h_j(s))\rangle$$

$$= -(\partial_t - \gamma)\langle\bar{z}_i(t)\rangle + 2\sqrt{\frac{\sigma_w^2}{N}}\int ds\,\langle\tilde{W}\delta_{ij}\phi'(h_j(t))\phi(h_j(s))\rangle,$$

where the third line follows from symmetry of $\tilde{W}(t, s)$ (since the product of the activation functions is symmetric).

$$\tilde{z}_i(t): \qquad (\partial_t + \gamma)\langle h_i(t)\rangle = -\kappa g(t)^2 \langle \tilde{z}_i(t)\rangle$$

$$-\int ds \left\langle \left( \sigma_b^2 + \sqrt{\frac{\sigma_w^2}{N}}\,\mathfrak{W}(t,s) + \sqrt{\frac{\sigma_u^2}{N}}\,\mathfrak{U}(t,s) \right) \tilde{z}_i(s) \right\rangle, \tag{96}$$

$$x_i(t): \qquad \int ds \left\langle \tilde{U}(t,s)\delta_{ij}\varphi'(x_j(t))\varphi(x_j(s)) \right\rangle = 0, \tag{97}$$

$$\mathfrak{X}(t_1,t_2): \qquad \langle \tilde{X}(t_1,t_2)\rangle = \frac{1}{2}\sqrt{\frac{\sigma_x^2}{N}}\langle \tilde{z}_i(t_1)\tilde{z}_i(t_2)\rangle = 0 =: \tilde{X}_0(t_1,t_2), \tag{98}$$

$$\tilde{W}(t_1,t_2): \qquad \langle \mathfrak{W}(t_1,t_2)\rangle = \sqrt{\frac{\sigma_w^2}{N}}\langle \phi(h_i(t_1))\phi(h_i(t_2))\rangle =: \mathfrak{W}_0(t_1,t_2), \tag{99}$$

$$\tilde{U}(t_1,t_2): \qquad \langle \mathfrak{U}(t_1,t_2)\rangle = \sqrt{\frac{\sigma_u^2}{N}}\langle \varphi(x_i(t_1))\varphi(x_i(t_2))\rangle =: \mathfrak{U}_0(t_1,t_2), \tag{100}$$

where the prime denotes the derivative with respect to the argument (e.g., $\phi'(y) = \partial_y\phi(y)$), and in the last three equations we have denoted the solutions by $\mathfrak{X}_0, \tilde{X}_0$, where the identification $\tilde{X}_0 = 0$ follows by (14). Then, since the expectation values of $\mathfrak{X}, \tilde{X}$ with other fields factorizes due to the lack of mixed quadratic terms in the action, we may substitute these into the eom for $h_i$ and $\tilde{z}_i$ to obtain

$$(\partial_t - \gamma)\langle \tilde{z}_i t)\rangle = 0,$$
$$(\partial_t + \gamma)\langle h_i(t)\rangle = -\kappa g(t)^2\langle \tilde{z}_i(t)\rangle - \int ds \left( \sigma_b^2 + \frac{\sigma_w^2}{N}\mathfrak{W}_0(t,s) + \frac{\sigma_u^2}{N}\mathfrak{U}_0(t,s) \right)\langle \tilde{z}_i(s)\rangle. \tag{101}$$

A consistent solution to these equations is

$$h_{i,0}(t) := \langle h_i(t)\rangle = 0, \qquad \tilde{z}_{i,0}(t) := \langle \tilde{z}_i(t)\rangle = 0. \tag{102}$$

Note that $x_i(t)$ is not constrained by its eom (which reduces to $0=0$), and thus we are free to select an arbitrary vev for the inputs; for simplicity (since we shall have in mind a nonlinear activation function with $\varphi(0) = 0$, see below), we shall take $x_{i,0}^\alpha(t) = 0$ as well. We may then expand the fields around these background values by making the following shifts in the action:

$$\mathfrak{X} \to \mathfrak{X}_0 + \mathfrak{X}, \qquad \tilde{X} \to \frac{1}{2}\tilde{X}. \tag{103}$$

Of course, only the $\mathfrak{X}$ fields must[34] be shifted, since the others all have vanishing vev; the factor of $1/2$ is introduced in $\tilde{X}$ for later convenience (see the comment below (142)).

However, the last line of (94) is intractable in its present form, since the fields $h_i, \tilde{z}_i$ are contained within the nonlinear activation functions $\phi, \varphi$. In order to proceed with the perturbative analysis, we shall henceforth take $\phi = \varphi = \tanh$, which permits the series expansion

$$\phi(h_i) = \tanh(h_i) = h_i - \frac{h_i^3}{3} + O(h_i^5), \tag{104}$$

and similarly for $\varphi(x)$. Our theory is then

$$\bar{Z} = \int \mathcal{D}\mathfrak{X}\,\mathcal{D}\tilde{X}\,\mathcal{D}h\,\mathcal{D}\tilde{z}\,e^{S_0 + S_{\text{int}}}, \tag{105}$$

---

[34]This is not technically necessary, but failing to do so complicates the diagrammatic expansion due to the insertion of arbitrary numbers of $\mathfrak{X}$ operators. Shifting the fields to expand around the vevs is a well-known cure for this annoyance; see for example the classic text [40].

where the quadratic part of the action is – again with implicit sums over repeated neuron indices –

$$
\begin{aligned}
S_0 = \int \mathrm{d}t &\left[ \tilde{z}_i(t)(\partial_t + \gamma) h_i(t) + \frac{\kappa}{2} g(t)^2 \tilde{z}_i(t)\tilde{z}_i(t) \right] \\
&+ \frac{1}{2} \int \mathrm{d}t_1 \mathrm{d}t_2 \left[ \sigma_b^2 + \sqrt{\frac{\sigma_w^2}{N}} \, \mathfrak{W}_0(t_1, t_2) + \sqrt{\frac{\sigma_u^2}{N}} \, \mathfrak{U}_0(t_1, t_2) \right] \tilde{z}_i(t_1)\tilde{z}_i(t_2) \\
&- \frac{1}{2} \int \mathrm{d}t_1 \mathrm{d}t_2 \left[ \tilde{W}(t_1, t_2)\mathfrak{W}(t_1, t_2) + \tilde{U}(t_1, t_2)\mathfrak{U}(t_1, t_2) \right],
\end{aligned}
\tag{106}
$$

where we have absorbed the tadpoles arising from the $\tilde{X}\mathfrak{X}_0$ terms by shifting the source fields for $\tilde{X}$. Meanwhile, the interaction part, to fourth-order in $h$ and $x$, is given by

$$
\begin{aligned}
S_{\text{int}} = \frac{1}{2} \int \mathrm{d}t_1 \mathrm{d}t_2 &\left[ \sqrt{\frac{\sigma_w^2}{N}} \, \mathfrak{W}(t_1, t_2) + \sqrt{\frac{\sigma_u^2}{N}} \, \mathfrak{U}(t_1, t_2) \right] \tilde{z}_i(t_1)\tilde{z}_i(t_2) \\
&+ \frac{1}{2} \int \mathrm{d}t_1 \mathrm{d}t_2 \left[ \sqrt{\frac{\sigma_w^2}{N}} \, \tilde{W}(t_1, t_2)\left( h_i(t_1)h_i(t_2) - \frac{2}{3} h_i(t_1)h_i(t_2)^3 \right) \right. \\
&\qquad\qquad \left. + \sqrt{\frac{\sigma_u^2}{N}} \, \tilde{U}(t_1, t_2)\left( x_i(t_1)x_i(t_2) - \frac{2}{3} x_i(t_1)x_i(t_2)^3 \right) \right].
\end{aligned}
\tag{107}
$$

In the next subsection, we will obtain the propagators for the theory given by (105). Unlike the MFT analysis in the previous section however, we will solve for the propagators explicitly, in order to proceed to obtain the finite-width corrections in subsections (4.3.1) and (4.3.2).

## 4.1 Propagators

To find the propagators of the theory (105), we first introduce the two-component fields

$$
y_i(t) := \begin{pmatrix} h_i(t) \\ \tilde{z}_i(t) \end{pmatrix}, \qquad Q(t_1, t_2) := \begin{pmatrix} \mathfrak{W}(t_1, t_2) \\ \tilde{W}(t_1, t_2) \end{pmatrix}, \qquad R(t_1, t_2) := \begin{pmatrix} \mathfrak{U}(t_1, t_2) \\ \tilde{U}(t_1, t_2) \end{pmatrix},
\tag{108}
$$

cf. (35). As will become clear below, it is necessary to formulate the Feynman diagrams in these terms, since $h_i, \tilde{z}_i$ (and by extension, the other pairs in (108)) cannot really be thought of as independent fields per se—the analogue of the kinetic term in $S_0$, for example, implies that the first component of $y_i^\alpha$ can freely propagate into the second. Accordingly, we begin by expressing the Gaussian part of the action in the form

$$
\begin{aligned}
S_0 = &-\frac{1}{2} \int \mathrm{d}t_1 \mathrm{d}t_2 \, y_i(t_1)^\mathsf{T} \Xi_{ij}(t_1, t_2) y_j(t_2) \\
&- \frac{1}{2} \int \prod_{i=1}^{4} \mathrm{d}t_i \left[ Q(t_1, t_2)^\mathsf{T} \Upsilon_w(t_1, t_2, t_3, t_4)Q(t_3, t_4) + R(t_1, t_2)^\mathsf{T} \Upsilon_u(t_1, t_2, t_3, t_4)R(t_3, t_4) \right],
\end{aligned}
\tag{109}
$$

where the operators $\Xi, \Upsilon_m$ with $m \in \{w, u\}$ are defined as

$$
\Xi_{ij}(t_1, t_2) = \delta_{ij} \begin{pmatrix} 0 & \delta(t_1 - t_2)(\partial_{t_2} - \gamma) \\ \delta(t_1 - t_2)(-\partial_{t_2} - \gamma) & -\kappa g(t_1)^2 \delta(t_1 - t_2) - \sigma_b^2 - \sqrt{\frac{\sigma_w^2}{N}} \mathfrak{W}_0(t_1, t_2) - \sqrt{\frac{\sigma_u^2}{N}} \mathfrak{U}_0(t_1, t_2) \end{pmatrix},
$$

$$
\Upsilon_m(t_1, t_2, t_3, t_4) = \frac{1}{2} \delta(t_1 - t_3)\delta(t_2 - t_4) \begin{pmatrix} 0 & 1 \\ 1 & 0 \end{pmatrix}.
\tag{110}
$$

As in the previous MFT analysis, the propagators are obtained as the elements of the Green functions (i.e., the inverse operators) for the operators $\Xi, \Upsilon_m$ appearing in the Gaussian part of the action, cf. (65). For $\Upsilon_m$, we have

$$\int \mathrm{d}s_1 \, \mathrm{d}s_2 \, \Upsilon_m(t_1, t_2, s_1, s_2) G_m(s_1, s_2, t_3, t_4) = \delta(t_1 - t_3)\delta(t_2 - t_4)\mathbb{1}, \qquad (111)$$

from which we immediately obtain

$$G_m(t_1, t_2, t_3, t_4) = 2\delta(t_1 - t_3)\delta(t_2 - t_4)\begin{pmatrix} 0 & 1 \\ 1 & 0 \end{pmatrix}. \qquad (112)$$

Substituting in the definition of the delta function (118) and imposing that the result hold for all momenta $\omega_1, \omega_2$, we find the momentum space propagator

$$\hat{G}_m(\omega_1, \omega_2) = 2\begin{pmatrix} 0 & 1 \\ 1 & 0 \end{pmatrix}. \qquad (113)$$

Turning to $\Xi$, we will henceforth suppress the $\delta_{ij}$ to simplify the notation, it being understood that the $h_i, \tilde{z}_i$ propagators do not mix indices; e.g., $G_{hh}$ is the propagator between a single pair of neurons, and is the same for all $i$:

$$\int \mathrm{d}s \, \Xi(t_1, s)\begin{pmatrix} G_{hh}(s, t_2) & G_{h\tilde{z}}(s, t_2) \\ G_{\tilde{z}h}(s, t_2) & 0 \end{pmatrix} = \delta(t_1 - t_2)\mathbb{1}, \qquad (114)$$

where $G_{\tilde{z}\tilde{z}} = 0$, cf. (14); this yields, for the diagonal elements,

$$\left(\partial_{t_1} - \gamma\right) G_{\tilde{z}h}(t_1 - t_2) = \left(-\partial_{t_1} - \gamma\right) G_{h\tilde{z}}(t_1 - t_2) = \delta(t_1 - t_2), \qquad (115)$$

and for the non-zero off-diagonal element,

$$\begin{aligned}
\left(\partial_{t_1} + \gamma\right)\left(\partial_{t_2} + \gamma\right) G_{hh}(t_1 - t_2) = {} & \kappa g(t_1)^2 \delta(t_1 - t_2) + \sigma_b^2 \\
& + \sqrt{\frac{\sigma_w^2}{N}} \, \mathfrak{W}_0(t_1, t_2) + \sqrt{\frac{\sigma_u^2}{N}} \, \mathfrak{U}_0(t_1, t_2),
\end{aligned} \qquad (116)$$

where we have performed the same $\partial_{t_1} = -\partial_{t_2}$ trick as before, cf. (67), again assuming translation invariance of the correlators. As a quick sanity check on the consistency of our expansion, observe that (after adjusting for the difference in normalization, cf. (92)) we have precisely recovered the MFT results (68), (69), as expected, since the former correspond to the tree-level contribution in perturbation theory.

As usual, it is more convenient to work in momentum (i.e., frequency) space, so we shall proceed by Fourier transforming the above expressions to obtain the momentum space propagators. We shall use the standard high-energy theory convention in which factors of $2\pi$ are attached to the momentum measure, and denote the momentum space functions with hats, i.e.,

$$\hat{f}(\omega) = \int \mathrm{d}t \, e^{-i\omega t} f(t) \qquad \text{and} \qquad f(t) = \int \frac{\mathrm{d}\omega}{2\pi} \, e^{i\omega t} \hat{f}(\omega), \qquad (117)$$

which further implies the convention

$$(2\pi)\delta(\omega) = \int \mathrm{d}t \, e^{-i\omega t} \qquad \text{and} \qquad \delta(t) = \int \frac{\mathrm{d}\omega}{2\pi} \, e^{i\omega t}, \qquad (118)$$

as one can readily verify by substituting $f(t)$ into $\hat{f}(\omega)$ (or vice versa). Beginning with $G_{\tilde{z}h}(t-s)$ in (115), we then have

$$\int \frac{d\omega}{2\pi} (i\omega - \gamma) e^{i\omega(t-s)} \hat{G}_{\tilde{z}h}(\omega) = 1, \tag{119}$$

and similarly for $G_{h\tilde{z}}(t-s)$. We thus identify

$$\hat{G}_{\tilde{z}h}(\omega) = \frac{-i}{\omega + i\gamma}, \qquad \hat{G}_{h\tilde{z}}(\omega) = \frac{i}{\omega - i\gamma}. \tag{120}$$

Note that $\hat{G}_{\tilde{z}h}(\omega) = \hat{G}_{h\tilde{z}}(-\omega)$, as expected, since these correspond to propagation of $y(t)$ in different directions in time. This can be seen by Fourier transforming (120) back to real space, which also clarifies the implicit pole prescription in the identifications thereof:

$$\begin{aligned} G_{\tilde{z}h}(t-s) &= \int \frac{d\omega}{2\pi} e^{i\omega(t-s)} \hat{G}_{\tilde{z}h}(\omega) = \frac{1}{2\pi i} \oint d\omega \frac{e^{i\omega(t-s)}}{\omega + i\gamma} \\ &= -\Theta(s-t) e^{-\gamma(s-t)}, \end{aligned} \tag{121}$$

where in the last step, the only non-vanishing contribution comes from the pole at $\omega = -i\gamma$; recalling from our discussion below (85) that $\gamma > 0$ in the regime of interest, this lies in the lower half-plane, and hence requires $t < s$ in order to close the contour; the negative sign then arises from encircling the pole clockwise. Similarly,

$$G_{h\tilde{z}}(t-s) = -\frac{1}{2\pi i} \oint d\omega \frac{e^{i\omega(t-s)}}{\omega - i\gamma} = -\Theta(t-s) e^{-\gamma(t-s)}, \tag{122}$$

where we have closed the contour in the upper half-plane, with $t > s$, to capture the pole at $\omega = i\gamma$. Thus we see that $G_{\tilde{z}h}(t-s) = G_{h\tilde{z}}(s-t)$, consistent with our observation above. Note that in both cases, the Green function is exponentially decreasing, as usual.

Turning now to the off-diagonal term (116), we must contend with the fact that, from the eom (99), $\mathfrak{W}_0$ will be a function of $G_{hh}$ when we perform the perturbative expansion. For consistency, let us work to fourth order in $h$ as above; then

$$\sqrt{\frac{N}{\sigma_w^2}} \mathfrak{W}_0(t_1, t_2) = \langle \phi(h_i(t_1)) \phi(h_i(t_2)) \rangle = \langle h_i(t_1) h_i(t_2) \rangle - \frac{2}{3} \langle h_i(t_1) h_i(t_2)^3 \rangle. \tag{123}$$

The first term on the right-hand side is simply $NG_{hh}(\tau)$, where $\tau := t_1 - t_2$. For the remaining term, since we are expanding around a free (Gaussian) theory,[35] we can apply Wick's theorem to write this as a sum of products of two-point functions. Let us temporarily drop the neuron indices, and instead employ the shorthand $h_t := h_i(t)$; then the four-point function simplifies to

$$\langle h_1 h_2 h_2 h_2 \rangle = \langle \overline{h_1 h_2} \overline{h_2 h_2} \rangle + \langle \overline{h_1 h_2 h_2 h_2} \rangle + \langle \overline{h_1 h_2 h_2 h_2} \rangle = 3 G_{hh}(0) G_{hh}(\tau), \tag{124}$$

where $G_{hh}(0) = G_{hh}(t, t)$ may be solved for self-consistently, cf. (135). Therefore, to this order,

$$\sqrt{\frac{\sigma_w^2}{N}} \mathfrak{W}_0(t_1, t_2) = \sigma_w^2 [1 - 2G_{hh}(0)] G_{hh}(\tau) = \sigma_{w,\text{eff}}^2 G_{hh}(\tau), \tag{125}$$

where for compactness we have defined the *effective variance*

$$\sigma_{w,\text{eff}}^2 := \sigma_w^2 (1 - 2G_{hh}(0)), \tag{126}$$

---

[35]That is, the full expectation values are computed with respect to the interacting theory (105), but when evaluating them, we expand in terms of the coupling (i.e., $\sqrt{N/\sigma_w^2}$), so that the leading contribution is (123), in which the expectation values are taken with respect to the Gaussian theory.

for reasons which will become clear in subsec. 4.3.1. Note that if we had worked to next (sixth) order in $h$, we would obtain an additional term of the form

$$\langle h_1 h_1 h_1 h_2 h_2 h_2 \rangle = 9\, G_{hh}(0)^2 G_{hh}(\tau) + 6\, G_{hh}(\tau)^3 \,, \tag{127}$$

where the combined combinatoric factor for the $n$-point correlator is $(n-1)!!$ (to see this, choose any point at random; there are $(n-1)$ ways to contract it, which leaves $(n-3)$ choices for the second pair, and so on). This would lead to a cubic differential equation for $G_{hh}(\tau)$, in place of the much simpler linear equation (128). To avoid this – and the associated explosion of possible Feynman diagrams – we have limited ourselves to only the first two orders in the Taylor expansion; we comment on this truncation in more detail below. Hence, substituting (125) into (116), we have

$$\left[ (\partial_\tau + \gamma)(-\partial_\tau + \gamma) - \sigma_{w,\mathrm{eff}}^2 \right] G_{hh}(\tau) = \kappa \delta(\tau) + \sigma_b^2 + \sqrt{\frac{\sigma_u^2}{N}}\, \mathfrak{U}_0(\tau) \,, \tag{128}$$

where for simplicity we will henceforth assume that $g(\tau)$ is constant (which we can absorb into $\kappa$, and hence set $g = 1$), and have written $\mathfrak{U}_0$ as though it were time-translation invariant.[36] Fourier transforming with respect to $\tau$, this then becomes

$$\int \frac{d\omega}{2\pi} \left[ (i\omega + \gamma)(-i\omega + \gamma) - \sigma_{w,\mathrm{eff}}^2 \right] \hat{G}_{hh}(\omega) e^{i\omega\tau} = $$
$$\int \frac{d\omega}{2\pi} \left( \kappa + 2\pi \sigma_b^2 \delta(\omega) + \sqrt{\frac{\sigma_u^2}{N}}\, \tilde{\mathfrak{U}}_0(\omega) \right) e^{i\omega\tau} \,, \tag{129}$$

which implies the identification

$$\hat{G}_{hh}(\omega) = \frac{\kappa + 2\pi \sigma_b^2 \delta(\omega) + \sqrt{\frac{\sigma_u^2}{N}}\, \tilde{\mathfrak{U}}_0(\omega)}{\omega^2 + \gamma^2 - \sigma_{w,\mathrm{eff}}^2} \,. \tag{130}$$

We can then Fourier transform this back to real space, where the sign of $\tau = t - s$ determines the contour (i.e., which pole we encircle):

$$G_{hh}(\tau) = \int \frac{d\omega}{2\pi} \frac{e^{i\omega\tau}}{\omega^2 + \gamma^2 - \sigma_{w,\mathrm{eff}}^2} \left( \kappa + 2\pi \sigma_b^2 \delta(\omega) + \sqrt{\frac{\sigma_u^2}{N}}\, \tilde{\mathfrak{U}}_0(\omega) \right) \,. \tag{131}$$

Let us evaluate the three terms in this expression one by one. The first is proportional to the inverse Fourier transform $\mathcal{F}^{-1}$ of $(\omega^2 + \gamma^2 - \sigma_{w,\mathrm{eff}}^2)^{-1}$, and is readily evaluated using the same contour methods as above:

$$\begin{aligned}
\mathcal{F}^{-1}\left( \frac{\kappa}{\omega^2 + \gamma^2 - \sigma_{w,\mathrm{eff}}^2} \right) &= \kappa \int \frac{d\omega}{2\pi} \frac{e^{i\omega\tau}}{\omega^2 + \gamma^2 - \sigma_{w,\mathrm{eff}}^2} \\
&= \kappa \int \frac{d\omega}{2\pi} \frac{e^{i\omega\tau}}{\left( \omega + i\sqrt{\gamma^2 - \sigma_{w,\mathrm{eff}}^2} \right)\left( \omega - i\sqrt{\gamma^2 - \sigma_{w,\mathrm{eff}}^2} \right)} \\
&= i\kappa \left[ \Theta(\tau) \frac{e^{-\sqrt{\gamma^2 - \sigma_{w,\mathrm{eff}}^2}\,\tau}}{2i\sqrt{\gamma^2 - \sigma_{w,\mathrm{eff}}^2}} - \Theta(-\tau) \frac{e^{\sqrt{\gamma^2 - \sigma_{w,\mathrm{eff}}^2}\,\tau}}{-2i\sqrt{\gamma^2 - \sigma_{w,\mathrm{eff}}^2}} \right] \\
&= \frac{\kappa\, e^{-\sqrt{\gamma^2 - \sigma_{w,\mathrm{eff}}^2}\,|\tau|}}{2\sqrt{\gamma^2 - \sigma_{w,\mathrm{eff}}^2}} \,.
\end{aligned} \tag{132}$$

---

[36]This amounts to assuming a sufficient degree of temporal uniformity in the injected data. In fact, we shall shortly assume that $\mathfrak{U}_0$ is time-independent, in order to perform the inverse Fourier transform. In principle, this is not required for the theory, but one must otherwise provide the form of the correlator $\mathfrak{U}_0$ as an input to the following analysis to obtain a closed-form result.

The second term is trivial on account of the delta function:

$$\mathcal{F}^{-1}\left(\frac{2\pi\sigma_b^2\delta(\omega)}{\omega^2+\gamma^2-\sigma_{w,\text{eff}}^2}\right)=\frac{\sigma_b^2}{\gamma^2-\sigma_{w,\text{eff}}^2}\,. \tag{133}$$

Lastly, by the convolution theorem, the third and final term can be expressed as

$$\mathcal{F}^{-1}\left(\frac{\sqrt{\frac{\sigma_u^2}{N}}\,\tilde{\mathfrak{U}}_0(\omega)}{\omega^2+\gamma^2-\sigma_{w,\text{eff}}^2}\right)=\sqrt{\frac{\sigma_u^2}{N}}\,\mathcal{F}^{-1}\left(\frac{1}{\omega^2+\gamma^2-\sigma_{w,\text{eff}}^2}\right)*\mathcal{F}^{-1}\left(\tilde{\mathfrak{U}}_0(\omega)\right)$$

$$=\frac{1}{2}\sqrt{\frac{\sigma_u^2}{N}}\int\mathrm{d}t\,\frac{e^{-\sqrt{\gamma^2-\sigma_{w,\text{eff}}^2}|t|}}{\sqrt{\gamma^2-\sigma_{w,\text{eff}}^2}}\,\mathfrak{U}_0(\tau-t)\,. \tag{134}$$

This is as far as we can evaluate this term without further information about the data $\mathfrak{U}_0$. To obtain a closed-form result, we shall take $\mathfrak{U}_0$ to be time-independent; cf. footnote 36. Summing these contributions, the result for the propagator is

$$G_{hh}(\tau)=\frac{\kappa}{2}\frac{e^{-\sqrt{\gamma^2-\sigma_{w,\text{eff}}^2}|\tau|}}{\sqrt{\gamma^2-\sigma_{w,\text{eff}}^2}}+\frac{\sigma_{b,\text{eff}}^2}{\gamma^2-\sigma_{w,\text{eff}}^2}\,, \tag{135}$$

where $\sigma_{w,\text{eff}}^2$ is defined in (126), and for compactness we have also defined an effective variance for the bias,

$$\sigma_{b,\text{eff}}^2:=\sigma_b^2+\sqrt{\frac{\sigma_u^2}{N}}\,\mathfrak{U}_0\,. \tag{136}$$

The form of the two-point correlator allows us to immediately identify the correlation length at tree-level:

$$\xi_0:=\frac{1}{\sqrt{\gamma^2-\sigma_{w,\text{eff}}^2}}\,, \tag{137}$$

that is,

$$G_{hh}(\tau)=\frac{\kappa}{2}\xi_0\,e^{-|\tau|/\xi_0}+\xi_0^2\sigma_{b,\text{eff}}^2. \tag{138}$$

As discussed in the introduction, the critical point is defined by $\xi_0\to\infty$, which occurs when

$$\sigma_{w,\text{eff}}^2=\sigma_w^2(1-2c_0)=\gamma^2\,, \tag{139}$$

which is precisely the point at which the MFT condition (91) is saturated to subleading order in $\phi'(h)=\text{sech}^2(h)\approx1-h^2$, with $\sigma_A^2=0$. Of course, this is to be expected, since MFT is nothing but the leading-order saddlepoint in the perturbative expansion. However, a critical reader might object that the MFT result (91) appears more accurate than the tree-level result (139), since the latter only includes the first two terms in the Taylor expansion of the nonlinearity. Here it is important to keep in mind that there are two unrelated expansions at play: the Taylor expansion of $\phi(h)=\tanh(h)$, and the perturbative QFT expansion in $T/N$. One advantage of the MFT approach to criticality in the previous section is that we were formally able to keep all orders in $h$, although the final expression (91) can only be evaluated numerically. The main disadvantage is that MFT ignores all fluctuations, and we have no *a priori* reason to expect that the prediction for the critical point is correct, even at infinite width. In contrast, the statistical field theory framework we have employed allows us to go beyond the MFT regime, but requires us to express the nonlinearity as a polynomial function of $h$; as mentioned when defining the effective variance (126) above, we have modelled the system up to quartic order in $h$ to obtain

more tractable equations. Since $\tanh(h) \in [-1, 1]$, we expect the next (sextic) order term to be less important than the subleading $T/N$ corrections for most values of $h$; in fact, this should hold for general non-pathological activation functions with $\phi(0) = 0$, since $\langle h \rangle = 0$ and the two-point function of $h$ is exponentially decreasing; see A.1 for further discussion. It would be interesting to explore the relative importance of these disparate corrections empirically.

Additionally, note that in this section we are working at weak 't Hooft coupling $\sigma_{w,\text{eff}}^2 < \gamma^2$, so that $\xi_0 \in \mathbb{R}$. Conversely, we derived (91) (i.e., $\sigma_{w,\text{eff}}^2 > \gamma^2$) as the condition on the potential for which the system is chaotic. In that regime, we can no longer define a meaningful correlation length, as the propagator $G_{hh}(\tau)$ picks up a complex part, with phase given by $e^{-i|\tau/\xi_0|}$. In fact, the transition from order to chaos can be seen in the complex frequency plane as a movement of the poles in the Green function (130) from the imaginary to the real axis, respectively, through a degenerate pole at the origin corresponding to the critical point. When computing the loop corrections below, we shall see that the weak coupling condition automatically guarantees convergence of the infinite series of Feynman diagrams, which is the basis for its importance in perturbative QFT.

Since the effective variance $\sigma_{w,\text{eff}}^2$ depends on $G_{hh}(0)$, cf. (125), it remains to solve (138) for this pseudo-initial condition in order to evaluate $G_{hh}(\tau)$. (Concretely, one evaluates $G_{hh}(\tau)$ at $\tau = 0$, which via (126) allows one to solve for $\xi_0$). While this can be done analytically, the resulting expression is exceedingly lengthy and not particularly enlightening, so we refrain from including it here. Of course, if one considers the linear model with $\phi(h) = h$, then $\sigma_{w,\text{eff}}^2 = \sigma_w^2$, and one can evaluate (138) directly.

As an aside, we note that we may express the first term in (135) as

$$G_{hh}(\tau) = \frac{\kappa}{2\sqrt{\gamma^2 - \sigma_{w,\text{eff}}^2}} \left[ \Theta(\tau) e^{-t\sqrt{\gamma^2 - \sigma_{w,\text{eff}}^2}} + \Theta(-\tau) e^{t\sqrt{\gamma^2 - \sigma_{w,\text{eff}}^2}} \right], \tag{140}$$

which is proportional to the sum of $G_{h\tilde{z}}$, $G_{\tilde{z}h}$ under the rescaling $\gamma \to \sqrt{\gamma^2 - \sigma_{w,\text{eff}}^2}$. In this sense, the leading term of $G_{hh}$ plays the role of the Feynman propagator in quantum field theory, while $G_{\tilde{z}h}$ and $G_{h\tilde{z}}$ are very loosely (given the cross-component nature of the propagation) analogous to the retarded and advanced propagators, respectively.

## 4.2 Feynman rules

In order to compute the perturbative corrections to the tree-level propagator (135), we must deduce the Feynman rules for the theory. To that end, the interaction part of the action (107) may be written in terms of the two-component fields (108) as

$$\begin{aligned}
S_{\text{int}} = \int \prod_{i=1}^4 dt_i \sum_{a,b} \Big\{ &\big[ q_{ab}(t_1, t_2, t_3, t_4) y_a(t_1) y_a(t_2) \\
&+ q'_{ab}(t_1, t_2, t_3, t_4) y_a(t_1) y_a(t_2)^3 \big] Q_b(t_3, t_4) \\
&+ \big[ r_{ab}(t_1, t_2, t_3, t_4) y_a(t_1) y_a(t_2) + r_b(t_1, t_2, t_3, t_4) x(t_1) x(t_2) \\
&+ r'_b(t_1, t_2, t_3, t_4) x(t_1) x(t_2)^3 \big] R_b(t_3, t_4) \Big\},
\end{aligned} \tag{141}$$

where in this subsection we have suppressed the neuron indices $i$ to make room for the subscripts $a, b \in \{1, 2\}$, which are component labels for the fields (108) (so for example, $y_1 = h$,

$y_2 = \tilde{z}$). We have also defined the coupling coefficients

$$q_{ab} := \frac{1}{2}\sqrt{\frac{\sigma_w^2}{N}}\,\delta(t_1 - t_3)\delta(t_2 - t_4)[\delta_{a2}\delta_{b1} + \delta_{a1}\delta_{b2}],$$

$$q'_{ab} := -\frac{1}{3}\sqrt{\frac{\sigma_w^2}{N}}\,\delta(t_1 - t_3)\delta(t_2 - t_4)\delta_{a1}\delta_{b2},$$

$\hspace{11cm}$ (142)

for the $Q$ vertices, and

$$r_{ab} := \frac{1}{2}\sqrt{\frac{\sigma_u^2}{N}}\,\delta(t_1 - t_3)\delta(t_2 - t_4)\delta_{a2}\delta_{b1},$$

$$r_a := \frac{1}{2}\sqrt{\frac{\sigma_u^2}{N}}\,\delta(t_1 - t_3)\delta(t_2 - t_4)\delta_{a2},$$

$$r'_a := -\frac{1}{3}\sqrt{\frac{\sigma_u^2}{N}}\,\delta(t_1 - t_3)\delta(t_2 - t_4)\delta_{a2},$$

$\hspace{11cm}$ (143)

for the $R$ vertices. Note that the reason for rescaling $\tilde{X}$ by $1/2$ in (103) was to make $q_{12} = q_{21}$; otherwise, the sum of Kronecker delta functions would be replaced by $\delta_{a2}\delta_{b1} + 2\delta_{a1}\delta_{b2}$.

Let us make a few observations before writing down the Feynman rules for this theory. First, while we have included them in this expression for completeness, the data-dependent ($x$) terms are in fact completely inconsequential, since the lack of any $xx$, $xh$, or $x\tilde{z}$ propagators means that $x$ never appears in any Feynman diagram. This further implies that all the internal $G_m$ propagators are $G_w$ (never $G_u$), cf. (112). Second, observe that each interaction vertex involves only a single factor of $Q$ or $R$, whose expectation value vanishes by the eom (99), (100). Therefore, we infer that only diagrams involving an even number of vertices will contribute. Additionally, the component index must flip between $G_m$ and $G_{hh}$, $G_{h\tilde{z}}$, or $G_{\tilde{z}h}$ at the vertex; e.g., $Q_{a=1} = \mathfrak{W}$ couples to $y_{a=2}y_{a=2} = \tilde{z}\tilde{z}$, and $Q_{a=2} = \tilde{W}$ couples to $y_{a=1}y_{a=1} = hh$. Furthermore, note that the component index flips (e.g., from $y_{a=1}$ to $y_{a=2}$) along all propagators except $G_{hh}$.

At a glance, it appears that the only $N$-dependence is in the coupling constants, such that a diagram with $2m$ vertices will contribute at $O(1/N^m)$. However, in analogy with the $O(N)$ vector model, we must sum over any internal neuron indices, which implies that internal loops have the potential to give rise to additional factors of $N$ that disrupts this simple counting. In fact, it turns out that the leading correction to the tree-level (MFT) result is due to an infinite number of *cactus diagrams* that all contribute at $\mathcal{O}(1)$; we shall begin with these in the next subsection.

Therefore, working to second-order in the expansion of tanh (but keeping terms only up to quartic order in $h, x$; see above), we have the following Feynman rules for the theory:

- $h(t)$ propagating to $h(s)$:  $G_{hh}(t-s) =$   $t \longrightarrow s$

- $h(t)$ propagating to $\tilde{z}(s)$:  $G_{h\tilde{z}}(t-s) =$   $t \longrightarrow s$

- $\tilde{z}(t)$ propagating to $h(s)$:  $G_{\tilde{z}h}(t-s) =$   $t \longrightarrow s$

- $\mathfrak{W}(t_1, t_2)$ propagating to $\tilde{W}(t_3, t_4)$:   $G_w(t_1, t_2, t_3, t_4) =$   $\begin{array}{c} t_1 \\ t_2 \end{array} \Longrightarrow \begin{array}{c} t_3 \\ t_4 \end{array}$

- 3-pt vertex $\tilde{W}(t_3, t_4)h(t_1)h(t_2)$:   $q_{12}(t_1, t_2, t_3, t_4) =$   $\begin{array}{c} t_1 \\ \\ \\ t_2 \end{array} \succ \begin{array}{c} t_3 \\ t_4 \end{array}$

- 3-pt vertex $\mathfrak{W}(t_3, t_4)\tilde{z}(t_1)\tilde{z}(t_2)$:   $q_{21}(t_1, t_2, t_3, t_4) =$

- 5-pt vertex $\tilde{W}(t_3, t_4)h(t_1)h(t_2)^3$:   $q'_{12}(t_1, t_2, t_3, t_4) =$

- Consider all possible arrangements at the vertices; e.g., include a copy of the diagram with $q_{21}(t_1, t_2, t_3, t_4) \to q_{21}(t_2, t_1, t_3, t_4)$ only if this produces a distinct diagram.

- Combinatorics: consider all possible ways to contract internal legs; e.g., in the 5-pt vertex $q'_{12}$, we can form loops by connecting pairs of incident $h$ legs in 3 different ways, cf. (124).

- For each internal loop with a free neuron index, we have an additional factor of $N$.

- Add the contributions of all diagrams that contribute at the desired order in $1/N$.

Some comments are in order. First, note that in all cases, we have used solid lines for the "physical" fields $h$, $\mathfrak{W} \sim \phi(h)\phi(h)$, and shaded lines for the auxiliary fields $\tilde{z}$, $\tilde{W}$. Second, the double faded line indicating $\mathfrak{W}$ propagating to $\tilde{W}$ does not represent true propagation per se, since $G_w$ is proportional to a product of delta functions. Consequently, the arrows on the double lines are not meant to indicate physical propagation, but are merely visual aids to help the reader trace the flow of fields through the Feynman diagrams, and may be chosen accordingly. Third, note that the order of the arguments in the couplings $q, q'$ is important; e.g., $q_{12}(t_1, t_2, t_3, t_4) \sim \delta(t_1 - t_3)\delta(t_2 - t_4)$ connects $\tilde{W}(t_3, t_4)$ to $h(t_1 = t_3)h(t_2 = t_4)$ as drawn, while $q_{12}(t_2, t_1, t_3, t_4) \sim \delta(t_2 - t_3)\delta(t_1 - t_4)$ swaps the attachment of the $h$ legs, so that $\tilde{W}(t_3, t_4)$ connects to $h(t_1 = t_4)h(t_2 = t_3)$. Lastly, as remarked in the introduction, the double-line notation here closely resembles that introduced by 't Hooft [38] to model quantum chromodynamics (QCD), and we will see deeper connections to the $O(N)$ vector model in the next subsection.

## 4.3   Linear models

To organize the presentation, we shall first consider perturbative corrections for linear models, i.e., $\phi(h) \approx h$. That is, we consider only the leading term in the Taylor series (123). In this case we have only the 3-pt vertices $q_{ab}$ in (141), which greatly simplifies the diagrammatic expansion. In the next two subsections, we will compute both the leading $\mathcal{O}(1)$ and subleading $\mathcal{O}(T/N)$ contributions to the two-point function (138). We will then generalize the analysis to include the 5-pt vertices $q'_{12}$ corresponding to the subleading $h^4$ term in (123) in subsec. 4.4.

### 4.3.1   Infinite $\mathcal{O}(1)$ cacti

As in the $O(N)$ model [38–44], the leading correction to the tree-level propagator $G_{hh}(\tau)$ comes from an infinite series of so-called *cactus diagrams*, each of which contributes at $\mathcal{O}(1)$. Intuitively, these quantify fluctuations from typicality in the ensemble of networks, and vanish in the limit where the 't Hooft coupling $\sigma_w^2 \to 0$, as we discuss in more detail in sec. 5.

Crucially, the contribution from these diagrams remains even as $N \to \infty$, and thus represents an important effect in networks of arbitrary size.[37] The first three diagrams in the infinite series, to linear order in $\phi(h)$, are illustrated in fig. 1. Note that in each case, we have $N$ choices for the neuron that runs in each loop, which precisely cancels the factor of $1/N$ from each pair of vertices.

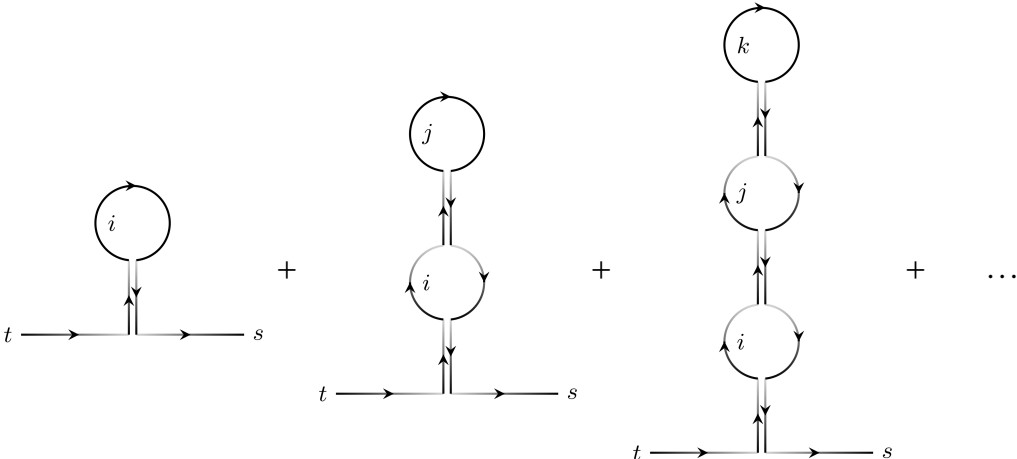

Figure 1: The leading correction to the MFT result is due to an infinite series of $\mathcal{O}(1)$ cactus diagrams; here, we have illustrated those arising to first order in the expansion of tanh, i.e., $\phi(h) \approx h$. Each pair of vertices contributes a combined factor of $1/N$, which is cancelled by a complementary factor of $N$ from the sum over an internal neuron index, labelled by $i, j, k$.

To illustrate the computation of such Feynman diagrams for those to whom they may be unfamiliar, let us proceed with maximum pedanticness on the first (one-loop) cactus. At that point the pattern should become clear, which will enable us to swiftly evaluate all remaining diagrams at this order. Label the legs at the lower vertex with times $u_1, \ldots, u_4$, and those at the upper vertex with times $v_1, \ldots, v_4$ (we have suppressed these in the figure to avoid clutter, but it is a trivial exercise for the reader to figure out where we put them). Note that we have the freedom to attach the legs at the lower vertex in two possible ways: $\mathfrak{W}(u_3, u_4)\tilde{z}(u_1)\tilde{z}(u_2)$, or $\mathfrak{W}(u_3, u_4)\tilde{z}(u_2)\tilde{z}(u_1)$, which leads to a distinct Feynman diagram with an equivalent contribution, so we have a combinatoric factor of 2. (For exactly the same reason, one obtains this factor of 2 for each internal $\mathfrak{W}\tilde{z}\tilde{z}$ vertex, which one can visualize as the freedom to twist the "stem" of the cactus 180 degrees around its vertical axis at each "bulb". Note that we do not have the freedom to rotate the top-most bulb, since this would produce an identical diagram). Introducing an overall factor of $N$ from the sum over the $i$ neurons running in the loop, we have

---

[37]This is the reason that MFT is generically *not* what one obtains from the CLT, contrary to some indications in the literature. In the Ising model for example, MFT amounts to replacing each degree of freedom, together with its nearest-neighbor interactions, with an effective degree of freedom – the so-called mean field – in which these interactions have been averaged over. This ignores long range interactions which become important near criticality, and is known to give incorrect results in low dimensions, while the CLT is agnostic about both.



$$
\begin{aligned}
&= 2N \int \prod_{i=1}^{4} du_i \, dv_i \, q_{21}(u_1, u_2, u_3, u_4) q_{12}(v_1, v_2; v_3, v_4) \\
&\quad \times G_{h\bar{z}}(t-u_1) G_{hh}(v_1-v_2) G_{\bar{z}h}(u_2-s) G_w(u_3, u_4, v_3, v_4) \\
&= 2N \int \prod_{i=1}^{4} du_i \, dv_i \, \frac{1}{2}\sqrt{\frac{\sigma_w^2}{N}} \delta(u_1-u_3)\delta(u_2-u_4) \\
&\quad \times \frac{1}{2}\sqrt{\frac{\sigma_w^2}{N}} \delta(v_1-v_3)\delta(v_2-v_4) G_{h\bar{z}}(t-u_1) G_{hh}(v_1-v_2) G_{\bar{z}h}(u_2-s) \\
&\quad \times 2\delta(u_3-v_3)\delta(u_4-v_4) \\
&= \sigma_w^2 \int du_1 \, du_2 \, G_{h\bar{z}}(t-u_1) G_{hh}(u_1-u_2) G_{\bar{z}h}(u_2-s) \\
&= \sigma_w^2 \int d\tau' \, d\tau'' \, G_{h\bar{z}}(\tau') G_{\bar{z}h}(\tau'') G_{hh}(\tau-\tau'-\tau'') \\
&= \sigma_w^2 (G_{h\bar{z}} * G_{hh} * G_{\bar{z}h})(\tau),
\end{aligned}
\tag{144}
$$

where in the penultimate step, we have made the change of variables

$$
\tau := t-s, \qquad \tau' := t-u_1, \quad \tau'' := u_2-s, \tag{145}
$$

whereupon we see that the result can be expressed in terms of convolutions $(f * g)(\tau) := \int dt \, f(\tau) g(t-\tau)$. In the last step, we have used commutativity to reorder the propagators in the order they appear as we traverse the diagram from $t$ to $s$, purely for illustrative purposes.

The general pattern exhibited by these cactus diagrams should now be clear.[38] To simplify notation, let us introduce $c := G_{hh}(\tau)$ as in our MFT treatment above, as well as $f := G_{h\bar{z}}(\tau)$ and $\bar{f} := G_{\bar{z}h}(\tau)$. An $n$-loop cactus diagram is then

$$
\text{cactus}_n(\tau) = \sigma_w^{2n} \, c * (f * \bar{f})^{*n}(\tau), \tag{146}
$$

where the exponent $*n$ denotes that the base is convolved $n$ times. However, since convolutions correspond to products in momentum space, it is more convenient to instead work with the Fourier transform

$$
\text{cactus}_n(\omega) = \sigma_w^{2n} \, c(\omega)[f(\omega)\bar{f}(\omega)]^n, \tag{147}
$$

where whether $c, f, \bar{f}$ are position or momentum space propagators will be indicated by the argument ($\tau$ or $\omega$, respectively).

We can now sum the infinite series to obtain the correction from all $\mathcal{O}(1)$ cactus diagrams in fig. 1. Including the bare propagator $c(\omega)$ as $n=0$, we have

$$
\begin{aligned}
\sum_{n=0}^{\infty} \text{cactus}_n(\omega) &= c(\omega) \sum_{n=0}^{\infty} \left[\sigma_w^2 f(\omega)\bar{f}(\omega)\right]^n \\
&= c(\omega) \sum_{n=0}^{\infty} \left(\frac{\sigma_w^2}{\omega^2+\gamma^2}\right)^n = c(\omega)\frac{\omega^2+\gamma^2}{\omega^2+\gamma^2-\sigma_w^2},
\end{aligned}
\tag{148}
$$

---

[38]As we commented when writing down the Feynman rules above, the effect of the internal $G_w$ propagator is simply to connect the legs on the vertices at either end via delta functions; in the QFT literature, cactus diagrams are often drawn without such internal propagators for this reason, but we shall continue to include them for clarity.

where we have substituted in the momentum space propagators (120), and summed the geometric series subject to the convergence condition

$$\sigma_w^2 < \omega^2 + \gamma^2 \,. \tag{149}$$

Since $\omega^2 > 0$ and these diagrams arise at leading order in the Taylor expansion of $\phi(h)$, at which $\sigma_{w,\text{eff}}^2 = \sigma_w^2$, this condition is automatically satisfied at weak coupling. Indeed, we observe that divergent behavior begins at precisely the tree-level criticality condition $\sigma_w^2 = \gamma^2$.

### 4.3.2 Infinite $\mathcal{O}(T/N)$ mushrooms

The subleading correction to the Gaussian propagator is the order at which we observe manifest finite-width effects. Here we will also find that the more relevant expansion parameter is $T/N$ rather than $1/N$ itself, in agreement with [2]. We shall first detail the computation of a generic, one-particle irreducible (1PI) $\mathcal{O}(T/N)$ diagram, and then use this to compute all contributions (including non-1PI diagrams) to the propagator at $\mathcal{O}(T/N)$ below.

As before, we have an infinite series of diagrams with cubic interactions; examples with one, two, and three loops are shown in fig. 2. Observe that in these diagrams, the propagator

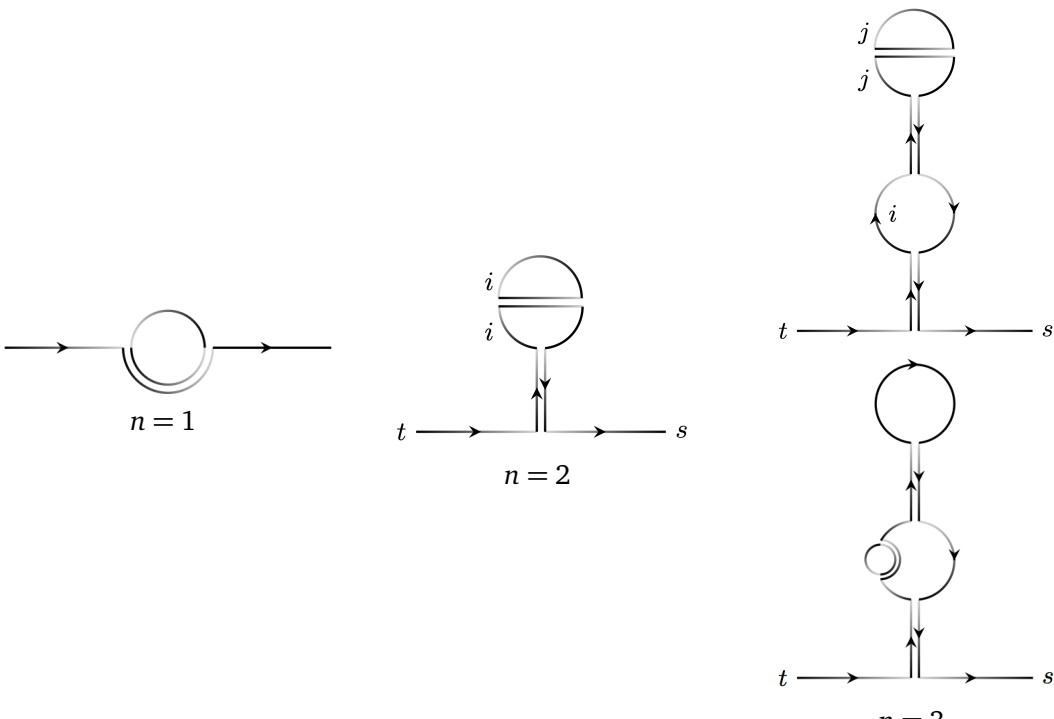

Figure 2: The subleading correction to the MFT result is due to an infinite series of $\mathcal{O}(T/N)$ mushroom diagrams. For linear models, the 1PI diagrams take the form shown here. Note that we also have a second, virtually identical series with the opposite orientation for the uppermost horizontal double-bar, or (in the case of the bottom right diagram) the nested loop on the opposite side. Relative to the $\mathcal{O}(1)$ cacti in fig. 1, the only difference is the so-called cap (hence the name) either at the top or on the side, which requires us to regulate the integrals, thereby introducing a factor of $T$. The cap also makes these diagrams $1/N$ suppressed, due to the extra pair of vertices with no additional sum over neurons. Note that the second diagram contains 2 loops, both diagrams in the third column contain 3, and the first diagram contains no free neuron indices.

that forms a closed/disconnected loop – which we call the mushroom "cap" – with one of the double lines in $G_w$ will evaluate to $G_{h\tilde{z}}(0) = G_{\tilde{z}h}(0) = -1/2$ (cf. (121), (122)) due to the identification of its endpoints via $G_w$, which leaves a free integral over an internal time variable that must be regulated, for a factor of $T$.[39] Additionally, while in the cactus diagrams the factor of $1/2^2$ from each pair of vertices precisely cancelled the factor of $2^2$ from the $G_w$ propagator (2 from the propagator itself, and 2 from the freedom to twist the stem), in the present case we do not have the freedom to twist the $G_w$ propagator in the cap, since the result would no longer be $\mathcal{O}(T/N)$. Instead, we have the option to change the orientation of the cap so that the propagator, traversed clockwise, may be either $f$ or $\bar{f}$. The momentum space contribution from a single $n$-loop 1PI mushroom diagram is then

$$\text{mushroom}_n(\omega) = -\frac{T}{N}\frac{\sigma_w^{2n}}{4}c(\omega)[f(\omega) + \bar{f}(\omega)][f(\omega)\bar{f}(\omega)]^{n-1}. \tag{150}$$

By (120),

$$f(\omega) + \bar{f}(\omega) = \frac{-i}{\omega + i\gamma} + \frac{i}{\omega - i\gamma} = \frac{-2\gamma}{\omega^2 + \gamma^2} = -2\gamma f(\omega)\bar{f}(\omega), \tag{151}$$

whence the previous expression simplifies to

$$\text{mushroom}_n(\omega) = \frac{T}{N}\frac{\gamma}{2}\sigma_w^{2n}c(\omega)[f(\omega)\bar{f}(\omega)]^n. \tag{152}$$

While it is possible to sum the infinite series to obtain the contribution from all 1PI mushroom diagrams, this does not yet include all contributions at $\mathcal{O}(T/N)$, since we must also consider non-1PI diagrams—in particular, those involving an arbitrary $n$-loop cactus, followed by an $n=1$ mushroom (or the reverse). To accomplish this task, let us introduce the following elegant recursion relation, as a warm-up for the nonlinear analysis in subsec. 4.4: let  represent the $hh$ propagator including all $\mathcal{O}(1)$ and $\mathcal{O}(T/N)$ corrections, and denote its momentum space expression by $X(\omega)$. The dots at either side of the circle are to indicate that the latter subsumes half of the external legs, that is, that the external legs can be either $c$, or $f$, $\bar{f}$. The diagrammatic recursion relation for $X(\omega)$ is then

$$ \tag{153} $$

where $X^{(0)}$ and $X^{(1)}$ respectively denote the $\mathcal{O}(1)$ and $\mathcal{O}(T/N)$ contributions to $X$ in the expansion

$$X = X^{(0)} + \frac{\gamma T}{N}X^{(1)}, \tag{154}$$

where we have pulled out the factor of $\gamma$ simply to make $T/N$ dimensionless, cf. (169). Note that $X$ includes the bare propagator, and that we can generate any $\mathcal{O}(1)$ or $\mathcal{O}(T/N)$ diagram

---

[39]That is, $\int_{-T/2}^{T/2} d\tau f(0) = -T/2$. We are then considering the perturbative regime in which $T$ and $N$ both go to infinity with $T/N \ll 1$ fixed, which is the regime of practical relevance for modern deep neural networks; see [2] for further discussion.

by recursively substituting for $X$, $X^{(0)}$ on the right-hand side. For example, to generate an $n=1$ cactus, we place the first diagram on the right-hand side (the bare propagator) in the space for $X$ in the second diagram; we then take this diagram and substitute it in again to generate an $n=2$ cactus, and so on. On the second line, the circles containing $X^{(0)}$ indicate that only $\mathcal{O}(1)$ diagrams may be recursively inserted there, since the presence of the explicit $n=1$ mushroom means that any further mushroom insertions would make the resulting diagram $\mathcal{O}(T^2/N^2)$.

Now, given the analysis of the 1PI diagrams above, the computation of this expression is quite straightforward. Consider the second diagram on the right-hand side: this takes the form of (147) with $n=1$ and $c(\omega) \to X(\omega)$. Similarly, the sum of the two diagrams on the second line is given by (152) with $n=1$ and $c(\omega) \to X^{(0)}(\omega)$. Thus, (153) reads

$$X(\omega) = c(\omega) + \sigma_w^2 f(\omega)\bar{f}(\omega)X(\omega) + \frac{\gamma T}{N}\frac{\sigma_w^2}{2}f(\omega)\bar{f}(\omega)X^{(0)}(\omega), \tag{155}$$

which we may solve order by order in $\gamma T/N$. Expanding $X$ as in (154) and solving for the $\mathcal{O}(1)$ contribution, we find

$$X^{(0)}(\omega) = \frac{c(\omega)}{1 - \sigma_w^2 f(\omega)\bar{f}(\omega)} = c(\omega)\frac{\omega^2 + \gamma^2}{\omega^2 + \gamma^2 - \sigma_w^2}, \tag{156}$$

which is precisely the $\mathcal{O}(1)$ correction obtained in (148), as expected. We then use this to solve for the $\mathcal{O}(T/N)$ contribution:

$$X^{(1)}(\omega) = \frac{\sigma_w^2}{2\left[1 - \sigma_w^2 f(\omega)\bar{f}(\omega)\right]}X^{(0)}(\omega) = \frac{\sigma_w^2}{2}c(\omega)\frac{\omega^2 + \gamma^2}{(\omega^2 + \gamma^2 - \sigma_w^2)^2}. \tag{157}$$

We emphasize that this expression includes all perturbative $\mathcal{O}(T/N)$ contributions to linear models (that is, to leading order in the Taylor expansion (123)), including diagrams involving an infinite number of loops.

### 4.3.3 The loop-corrected correlation length

Adding the results (156) (equivalently (148)) and (157), we have thus found that the loop-corrected two-point function in momentum space, to leading order in $T/N$, is

$$X(\omega) = c(\omega)\frac{\omega^2 + \gamma^2}{\omega^2 + \gamma^2 - \sigma_w^2}\left[1 + \frac{T}{N}\frac{\gamma\sigma_w^2}{2(\omega^2 + \gamma^2 - \sigma_w^2)}\right], \tag{158}$$

where the tree-level propagator $c(\omega)$ is given by (130) with $\sigma_{w,\text{eff}}^2 = \sigma_w^2$ for linear models. Performing the inverse Fourier transform, the result in position space reads

$$\begin{aligned}
X(\tau) = \frac{\kappa}{4}\xi_0\, e^{-|\tau|/\xi_0}&\left\{1 + \gamma^2\xi_0^2 + \xi_0\sigma_w^2|\tau|\right.\\
&\left.+ \frac{T}{N}\frac{\gamma\sigma_w^2}{8}\xi_0\left[\xi_0\left(1 + 3\gamma^2\xi_0^2 + \sigma_w^2\tau^2\right) + \left(1 + 3\gamma^2\xi_0^2\right)|\tau|\right]\right\}\\
&+ \frac{\gamma^2}{2}\xi_0^4\left(2 + \gamma\frac{T}{N}\xi_0^2\sigma_w^2\right)\sigma_{b,\text{eff}}^2,
\end{aligned} \tag{159}$$

we have again taken $\mathfrak{U}_0$ to be time-independent, with $\sigma_{b,\text{eff}}^2$ as in (136).

With the loop-corrected propagator in hand, we would now like to identify the correlation length in the presence of both $\mathcal{O}(1)$ fluctuations and $\mathcal{O}(T/N)$ finite-width effects. At a glance, this is obstructed by the mixed exponential and polynomial dependence on $\tau$. However, as explained at the beginning of this section, we do not require an expression for the correlation

length valid at all times; rather, it suffices to identify an effective correlation length $\xi$ that governs the evolution of the correlator for small $|\tau|$ (i.e., small perturbations about the critical point, as in [17] and related works, as well as our MFT treatment in sec. 3).[40] Thus, we can identify a meaningful loop-corrected correlation length by first expanding the exponential in (159), collecting terms order by order in $|\tau|$, and re-exponentiating to obtain a purely exponential term of the form $e^{-|\tau|/\xi}$. To linear order in $|\tau|$, we find

$$X(\tau) \approx \frac{\kappa}{2} \xi \, e^{-|\tau|/\xi} + \frac{\gamma^2}{2} \xi_0^4 \left( 2 + \gamma \frac{T}{N} \xi_0^2 \sigma_w^2 \right) \sigma_{b,\text{eff}}^2, \tag{160}$$

where we have identified the loop-corrected correlation length

$$\xi := \frac{\xi_0}{2} \left[ 1 + \gamma^2 \xi_0^2 + \frac{\gamma T}{8N} \left( 1 + 3\gamma^2 \xi_0^2 \right) \xi_0^2 \sigma_w^2 \right]. \tag{161}$$

Note that we have not included any contributions from the effective bias variance $\sigma_{b,\text{eff}}^2$ in the new correlation length, in order to clearly separate the corrections to the exponential vs. constant term in the tree-level propagator (138). (Another way to see this distinction is that the exponential pieces vanish as $\tau \to \infty$, and therefore including terms proportional to $\sigma_{b,\text{eff}}^2$ in the loop-corrected correlation length would not recover the $\tau$-independent contribution to (159)).

Let us make a few comments about this result. First, as expected, we observe corrections to the correlation length at $\mathcal{O}(T/N)$ due to finite-width effects. Intuitively, non-Gaussianities accumulate with depth as a result of effective intralayer interactions that can be seen to appear in a manner precisely analogous to RG [2, 19]. Conversely, increasing the width suppresses these higher cumulants and brings one closer to the Gaussian result predicted from the CLT. Observe however that taking $T/N \to 0$ does *not* recover the tree-level result (137) (equivalently, the MFT result in sec. 3), due to the presence of $\mathcal{O}(1)$ contributions. In other words, this $\mathcal{O}(1)$ effect is independent of the network size, and thus represents an important contribution even in the infinite-width limit. As will be discussed in more detail in sec. 5, the $\mathcal{O}(1)$ corrections quantify fluctuations from typicality in the ensemble of random networks under consideration; in the limit $\sigma_w^2 \to 0$, the weight initialization becomes deterministic, so the MFT result (which ignores fluctuations about the mean) becomes exact. However, perhaps surprisingly, we do not observe any shift in the critical point itself: since $\sigma_w \in \mathbb{R}$, the only divergence in $\xi$ is precisely the tree-level divergence $\gamma^2 = \sigma_w^2$. We shall comment further on this in the next subsection as well as in sec. 5 below.

## 4.4 Nonlinear models

In the previous subsection, we obtained the loop-corrected propagator to subleading order in perturbation theory for linear models, for which $\phi(h) = h$. In this subsection, we would like to generalize this analysis to nonlinear models by including the second, $h^4$ term in (123), i.e., the 5-pt interaction $q'_{12}$ in (141). This allows an infinite variety of new Feynman diagrams that contribute at all orders in $T^m/N^n$ with $0 \leq m \leq n$.[41] In this work, we shall limit ourselves to diagrams with $m = n = 1$, as those with $1 < m < n$ are subleading in the limit $N \to \infty$ with $T/N$ held fixed. We shall start in subsubsec. 4.4.1 by considering all additional $\mathcal{O}(1)$ cacti that become possible once 5-pt interactions are included, and then do the same for mushroom diagrams in subsubsec. 4.4.2.

---

[40]A sufficient but not necessary condition for this is $|\tau| \ll \xi_0$, which certainly holds near the critical point.

[41]This is an oversimplification, as there exists one particular 1PI diagram that contains a contribution at $\mathcal{O}(T^2/N)$, which can be accommodated with a constraint on $\sigma_{b,\text{eff}}^2$, as discussed below (167).

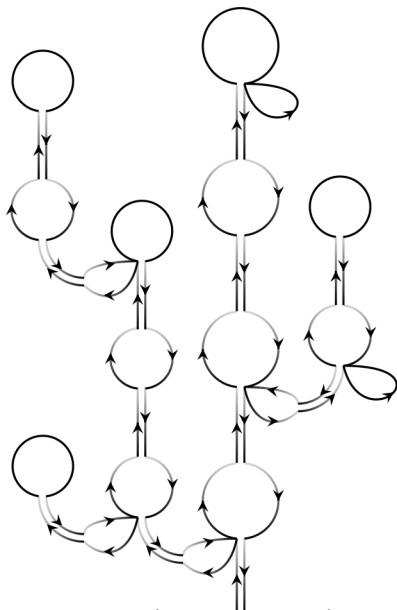

Figure 3: A generic $O(1)$ cactus, including both 3-pt ($\tilde{W}h^2$) and 5-pt ($\tilde{W}h^4$) vertices. Here we have a combined factor of $1/N^{12}$ from the vertices, which is precisely cancelled by the factor of $N^{12}$ from neurons running in the internal loops. In this particular diagram, we have included two *petals* on the right-hand side of the cactus (the internal $c_0$ propagators that begin and end at the same vertices), and four *branches* (self-similar cacti whose would-be external legs form a loop at the vertex where the child branch joins the parent).

### 4.4.1 Branching cacti

Observe that the inclusion of 5-pt interactions allows us to add up to one "petal" (a closed $hh$ propagator) or "branch" (a self-similar cactus with $n > 1$ loops) to any number of $\tilde{W}(t,s)h(t)h(s)$ vertices in fig. 1. A single specimen from the resulting infinitely-branching forest of cacti is shown in fig. 3.

We thus have a recursive sequence of cactus diagrams, which we can evaluate using a similar recursive approach to that in the previous subsection. Let  represent the loop-corrected propagator including all possible cacti, allowing both 3-pt and 5-pt interactions, and denote its momentum space expression by $Y^{(0)}(\omega)$. That is, we expand

$$Y = Y^{(0)} + \frac{\gamma T}{N} Y^{(1)}, \tag{162}$$

cf. (154), where here we have used $Y$ in place of $X$ to denote the inclusion of the subleading term in the expansion of the nonlinearity (123). Then the recursion relation for all $\mathcal{O}(1)$ diagrams is simply

$$\text{(163)}$$

Implicitly, we also include a copy of the last diagram with the branch attached to the other side of the stem.

Now, the second diagram on the right-hand side is of course simply (147) with $n = 1$ and $c(\omega) \to Y^{(0)}(\omega)$. For the third, let us first consider the case in which we replace the $Y^{(0)}$ in the branch with the bare propagator, resulting in a closed $hh$ propagator or petal. Since the temporal endpoints are the same, this simply contributes a factor of $-c_0 := -c(\tau = 0)$, where the negative sign stems from the coupling $q'_{12}$, cf. (142). The factor of $1/3$ in the latter is precisely cancelled by the 3 ways to contract the two pairs of $h$-legs. Importantly, note that the neuron index in the petal is the same as that in the adjacent loop; this can be seen from the indices in (107). Therefore, while petalous vertices gain an extra loop, they contribute at the same order as their apetalous counterparts in the perturbative expansion. Therefore, this diagram – plus the copy with the petal attached to the opposite side of the stem – is given by (147) with $\sigma_w^2 \to -2\sigma_w^2 c_0$ and $c(\omega) \to Y^{(0)}(\omega)$.

This lesson generalizes to the case in which the branch is a recursive cactus of arbitrary complexity: the delta functions within it will collapse all times to those at the vertex at which the branch joins the trunk, so that the entire branch is evaluated at $\tau = 0$. Hence, the expression for the third diagram on the right-hand side of (163) – plus the copy with the branch attached to the opposite side of the stem – is formally identical to the petalous cactus just described, but with $\sigma_w^2 \to -2\sigma_w^2 Y_0^{(0)}$, where $Y_0^{(0)} := Y^{(0)}(\tau = 0)$ includes the bare propagator $c_0$. The recursion relation (163) then reads

$$Y^{(0)}(\omega) = c(\omega) + \hat{\sigma}_w^2 f(\omega)\bar{f}(\omega)Y^{(0)}(\omega), \tag{164}$$

where we have defined the $\mathcal{O}(1)$ loop-corrected variance

$$\hat{\sigma}_w^2 := \sigma_w^2(1 - 2Y_0^{(0)}). \tag{165}$$

Solving this for $Y^{(0)}$ and recalling from (151) that $f\bar{f} = (\omega^2 + \gamma^2)^{-1}$, we obtain the $\mathcal{O}(1)$-corrected propagator, including both 3-pt ($h^2$) and 5-pt ($h^4$) interactions:

$$Y^{(0)}(\omega) = c(\omega)\frac{\omega^2 + \gamma^2}{\omega^2 + \gamma^2 - \hat{\sigma}_w^2}. \tag{166}$$

Comparing this with (148), we see that the effect of the 5-pt vertex is to modify the effective variance we defined in (126) to (165).

### 4.4.2 Branching mushrooms

As in the $\mathcal{O}(1)$ cactus diagrams above, the inclusion of 5-pt vertices again allows us to add up to one petal or branch to any number of $\tilde{W}hh$ vertices in fig. 2, so that we obtain an infinitely-branching forest of mushroom diagrams similar to the generic cactus in fig. 3, but which contain exactly one cap. Note that the cap need not be placed at the top of a stem, but can alternatively occur on any internal loop, as illustrated in fig. 2.

However, there is an additional type of mushroom cap that becomes possible once 5-pt interactions are included, which is given in (167). We have a factor of $-\frac{1}{3}\sqrt{\frac{\sigma_w^2}{N}}$ from the 5-pt vertex, cf. $q'_{12}$ in (141), and a factor of $\frac{1}{2}\sqrt{\frac{\sigma_w^2}{N}}$ from the 3-pt vertex as before. The two internal $h\tilde{z}$ propagators evaluate to $f(\tau = 0)^2 = (-1/2)^2$. We have a combinatoric factor of $3 \cdot 2$ from the 5-pt vertex (3 choices for which $h$ to contract with $\tilde{z}$, and 2 ways to connect the remaining $h$'s to the two external lines, which accounts for the possibility of rotating the entire diagram 180 degrees around the vertical axis), and an additional factor of 2 for the choice of whether to connect the external lines to the bottom left or bottom right of the internal $G_w$ propagator. Finally, we have a factor of $T$ from the disconnected cap, and a convolution of the

two external $c(\tau)$ propagators. Note that there is no free neuron index, so that the diagram is formally $\mathcal{O}(T/N)$. We therefore have

$$\text{[diagram]} = -\frac{T}{N}\frac{\sigma_w^2}{2}c(\omega)^2. \tag{167}$$

This diagram presents an important subtlety: recall that the bare propagator (130) contains a term proportional to $\sigma_{b,\text{eff}}^2\delta(\omega)$, where the effective bias variance is given by (136). Therefore, the product $c(\omega)^2$ will contain a factor of $\delta(\omega)^2 \sim \delta(0)\delta(\omega)$. This is an IR divergence that arises from integrating the constant term over the entire depth of the network, i.e.,

$$2\pi\delta(0) = \lim_{T\to\infty}\int_{-T/2}^{T/2}\mathrm{d}\tau\, e^{-i\omega\tau}\Big|_{\omega=0} = \lim_{T\to\infty}\int_{-T/2}^{T/2}\mathrm{d}\tau = \lim_{T\to\infty}T. \tag{168}$$

As before, we must regulate this divergence by imposing a finite cutoff $T$, so that the above diagram contains a contribution that scales like $T^2/N$. At a glance, this appears pathological, but instead highlights the fact that the weak coupling regime is governed by both $\sigma_w^2$ and $\sigma_b^2$; in particular, in addition to the convergence condition $\sigma_w^2 < \gamma^2$ encountered above, we also require $\sigma_b^2 < \gamma^2$. To see this, observe that by dimensional analysis, the various components of the action (106) have the following mass (inverse time) dimensions:

$$[h] = [\tilde{z}] = [\phi] = 0, \qquad [\gamma] = [\sigma_w] = [\sigma_b] = 1, \tag{169}$$

which further implies $[\mathfrak{X}] = [\tilde{X}] = 1$, cf. (92). Hence, expressing the constant (non-exponential) term of $c(\omega)$ as $T\xi_0^2\sigma_{b,\text{eff}}^2$, the $\delta(\omega)^2$ term in (167) under discussion reads

$$-\frac{T}{N}\frac{\sigma_w^2}{2}\Big(T\xi_0\sigma_{b,\text{eff}}^2\Big)^2 = -\frac{1}{2}\Big(\frac{\gamma T}{N}\Big)\Big(\frac{\sigma_w}{\gamma}\Big)^2(\gamma\xi_0)^4\Big(\frac{\sigma_{b,\text{eff}}}{\gamma}\Big)^4(\gamma T), \tag{170}$$

where on the right-hand side each factor in parentheses is dimensionless. Now, we see that the (dimensionless) perturbative expansion parameter is $\gamma T/N \ll 1$. As discussed above, convergence of the infinite series of diagrams at each order demands weak 't Hooft coupling, $\sigma_w/\gamma < 1$. The value of $\xi_0$ itself is unconstrained. Therefore, consistency of the perturbative expansion requires that $\sigma_{b,\text{eff}}/\gamma \ll 1$ to compensate for the fact that networks of practical interest have $\gamma T \gg 1$. Note that this constraint is not required for the other terms in (167), which scale like $T/N$; for this reason, we will include the $T^2/N$ term at this order in perturbation theory under the assumption that $\sigma_{b,\text{eff}}^2$ is sufficiently small, which can be seen as completing the prescription of the weak coupling regime in the 2d phase space parametrized by $(\sigma_w^2, \sigma_b^2)$.[42]

While the above exhausts the types of mushroom caps themselves which are possible at this order in the expansion of the nonlinearity (123), it does not exhaust the possible types of 1PI diagrams at $\mathcal{O}(T/N)$, since we also have the freedom to introduce internal pairs of $hh$ propagators connecting two different $\tilde{W}hh$ vertices. In fact, we can replace any of these internal propagators with a recursive $\mathcal{O}(1)$ diagram, i.e., $c(\omega) \to Y(\omega)$. The 1PI diagrams which contribute at order $T^m/N^n$ with $m = n$ are of the type shown in fig. 4. The key observation is that in these four diagrams, these extra internal propagators connect two vertices with the same internal neuron index, so they do not change the order at which the diagram contributes in the perturbative expansion. Conversely, had we connected vertices in two different loops, the internal neuron indices would be identified (since this cannot change along $c_\tau$), which

---

[42]Note that $\sigma_{b,\text{eff}}^2$ includes the injection of constant data, which will have the same physical effect as a constant bias variance when integrated over infinite depth.

thereby removes a factor of $N$, so that the resulting diagram would be $1/N$ suppressed relative to the same diagram without that internal connection. By adding an arbitrary number of these interloop connections to the generic mushroom/cactus diagram in fig. 3, one readily sees the possibility to construct diagrams at any order in $T^m/N^n$ with $0 \leq m \leq n$. Even this does not exhaust the list of possible diagrams, since one can grow these interloop connections into branches, and form nested loops of arbitrary complexity. However, all diagrams with connections between different vertices, as well as all intervertex branches, will have $m < n$ due to the aforementioned identification of internal neuron indices, and are thus subleading in the limit $N \to \infty$ with $T/N$ held fixed. For this reason (and to keep our analysis itself within the metaphorical ordered phase), we shall limit ourselves here to diagrams with $0 \leq m = n$.

Unfortunately, these last four diagrams are considerably more complicated than any we have considered above, since they involve a mixture of products and convolutions in both position and momentum space. Let us begin with the first, in comparison to the momentum space expression (150): we have the same factor of $-T/2$ from the top-most $f(0) = \bar{f}(0)$ together with the remaining (regulated) integral over $\tau$ in the cap, and a net factor of $1/N$ ($1/N^2$ from the vertices, and $N$ from the internal neuron index). The two $q_{ab}$ vertices will contribute a collective factor of $(\sigma_w/2)^2$, while the two $q'_{ab}$ vertices will contribute $(-\sigma_w/3)^2$, cf. (141). The $G_w$ propagators themselves carry a net factor of $2^2$, and the freedom to twist the $G_w$ in the stem contributes an additional factor of 2. Note that we do not have the freedom to twist the horizontal $G_w$ propagator, but can instead change the orientation of the cap, so that diagrams with either $f$ or $\bar{f}$ running along the underside top-most bulb are allowed. We also have a combinatoric factor of 3! from the possible ways of connecting the three internal $hh$ propagators, and – as mentioned above – can insert a self-similar $Y^{(0)}$ to any of them. Finally, the external legs contribute $f\bar{f}$ as before, so that

$$
\text{[diagram]} \quad = \frac{4}{3}\frac{T}{N}\gamma \sigma_w^4 [f(\omega)\bar{f}(\omega)]^2 Y^{(0)}(\omega)^{*3}, \tag{171}
$$

where we have used (151).

For the next diagram in fig. 4, we again have a collective factor of $\sigma_w^4/(2^2 3^2 N^2)$ from the vertices, a factor of $2^3$ from the two $G_w$ propagators plus the freedom to twist the one in the stem, an $f\bar{f}$ from the external legs, and $f + \bar{f} = -2\gamma f\bar{f}$ (in momentum space) from the possible orientations of the cap, and $Y^{(0)}$ in place of each of the internal $c$ propagators. However, while the top-most propagator again evaluates to $f(0) = \bar{f}(0) = -1/2$, we are not quite free to integrate over the remaining vertex position (time), since this will be encountered by the intervertex $(Y^{(0)} * Y^{(0)})(\omega)$ propagator, which will thus be evaluated at $\tau = 0$ (similar to

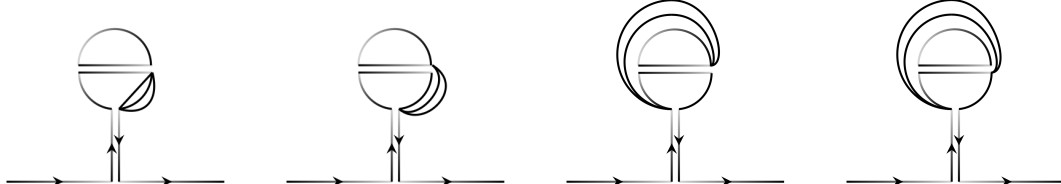

Figure 4: Additional $\mathcal{O}(T/N)$ mushroom diagrams that become possible once 5-pt interactions – corresponding to the second term in the Taylor series (123) – are included. Note that since the extra internal $c_\tau$ (generically, $Y^{(0)}(\tau)$) propagators connect two vertices through which the same neuron index already flows, they do not change the order of these diagrams relative to the 3-pt mushrooms in fig. 2.

how the delta functions collapse the temporal indices on the branches to $X_0$, $Y_0$ in the diagrams above). Since $Y^{(0)} * Y^{(0)}$ will generically be some sum of exponentials plus a constant piece, this last is the only divergent contribution that requires us to regulate the integral, and hence carries the sought-after factor of $T$. The other, exponential terms will carry no such factors,[43] and hence contribute at order $1/N$ in the expansion; we therefore drop them, and keep only the constant $\times T$ part of this convolution, which we denote $T b_0^2$ (so-named because the constant part of the propagator is always the bias-dependent term). Additionally, the other difference relative to the previous diagram is that in place of 3! from the intervertex combinatorics, we have $2 \cdot 3^2$: 3 for which of the legs from one vertex to connect with (3 additional choices for) a leg from the other vertex, and 2 ways to connect the remaining two pairs. Thus, including the overall factor of $N$ from the internal neuron index, we have

$$= 4\frac{T}{N}\gamma\sigma_w^4 b_0^2 [f(\omega)\bar{f}(\omega)]^2 Y^{(0)}(\omega). \tag{172}$$

The third diagram in fig. 4 offers a moment's respite: it is precisely the same as (172). We thus move to the fourth and final diagram, which has $-T/2$ from the cap and a net factor of $\sigma_w^4/(2^2 3^3 N)$ from the vertices and neuron index as in the first diagram, and the combinatoric factor of $2^3 \cdot 2 \cdot 3^2$ as in the second/third. As for the propagators themselves, observe that by labelling the vertices on either side of the stem $u_1, u_3$, and the vertex on the underside of the horizontal $G_w$ propagator $u_2$, the two possible orientations of the diagram read[44]

$$\int du_1 du_2 du_3 f(t-u_1)f(u_1-u_2)Y^{(0)}(u_1-u_2)^2 Y^{(0)}(u_2-u_3)\bar{f}(u_3-s)$$
$$= \left[ f * \left( f \cdot Y^{(0)2} \right) * Y^{(0)} * \bar{f} \right](\tau), \tag{173}$$

which one can see by traversing the diagram from left to right, and

$$\int du_1 du_2 du_3 f(t-u_1)Y^{(0)}(u_1-u_2)Y^{(0)}(u_2-u_3)^2 \bar{f}(u_2-u_3)\bar{f}(u_3-s)$$
$$= \left[ f * Y^{(0)} * \left( Y^{(0)2} \cdot \bar{f} \right) * \bar{f} \right](\tau), \tag{174}$$

which one can see by traversing the diagram from right to left, with $t \leftrightarrow s$ and $u_1 \leftrightarrow u_3$. Therefore,

$$= -2\frac{T}{N}\sigma_w^4 \left\{ f * \bar{f} * Y^{(0)} * \left[ (f+\bar{f}) \cdot Y^{(0)2} \right] \right\}(\tau)$$

$$= 4\frac{T}{N}\gamma\sigma_w^4 \left\{ f * \bar{f} * Y^{(0)} * \left[ (f*\bar{f}) \cdot Y^{(0)2} \right] \right\}(\tau)$$
$$\xrightarrow{\mathcal{F}} 4\frac{T}{N}\gamma\sigma_w^4 f(\omega)\bar{f}(\omega) Y^{(0)}(\omega) \left\{ Y^{(0)}(\omega) * Y^{(0)}(\omega) * [f(\omega)\bar{f}(\omega)] \right\}, \tag{175}$$

where in the last step we have Fourier transformed to momentum space. Prior to this, since in the first line $f + \bar{f}$ is evaluated in position space, (121) and (122) imply

$$f(\tau) + \bar{f}(\tau) = -e^{-\gamma|\tau|} = -2\gamma(f * \bar{f})(\tau), \tag{176}$$

---

[43]More precisely, the regulators in all convergent integrals are exponentially suppressed; see sec. 5.

[44]Recall from (112) that the temporal indices on either end of the $G_w$ propagators are identified by virtue of the delta functions.

where we have used the identity (B.2).

With the form of these 1PI diagrams in hand, we can now proceed to consider all possible contributions to $c(\omega)$ from both 1PI and non-1PI diagrams. To do so, let us extend the diagrammatic notation introduced in subsubsec. 4.3.2 to include 5-pt interactions, so that  represents the loop-corrected propagator up to $\mathcal{O}(T/N)$ and subleading order in the expansion of the nonlinearity (123), and denote its momentum space expression by $Y(\omega)$, cf. (162). We then have the recursion relation

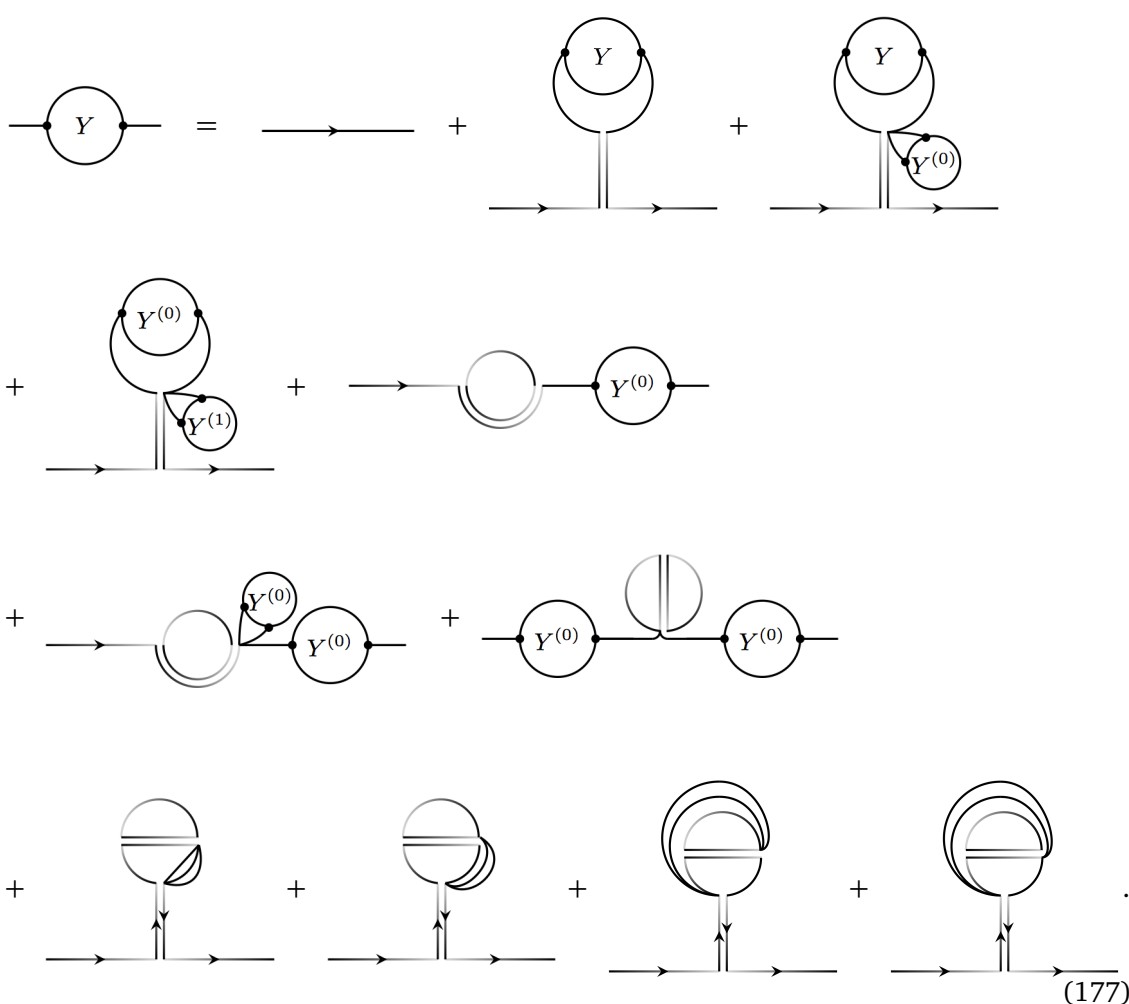

$$\tag{177}$$

Let us consider the various contributions to this expression in turn. From our analysis in the previous subsubsection, we know that the first two cactus-like diagrams are collectively given by (147) with $c(\omega) \to Y(\omega)$ and $\sigma_w^2 \to \hat{\sigma}_w^2 = \sigma_w^2(1 - 2Y_0^{(0)})$. Similarly, for the third cactus diagram, we have $c(\omega) \to Y^{(0)}(\omega)$ (since allowing $Y^{(1)}$ would make this diagram $\mathcal{O}(T^2/N^2)$) and $\sigma_w^2 \to -2\sigma_w^2 \frac{\gamma T}{N} Y_0^{(1)}$; note that we are including both possible choices for the side at which to attach the branches. The next two diagrams on the second row are collectively given by (152) with $n=1$, $c(\omega) \to Y^{(0)}(\omega)$, and $\sigma_w^2 \to \hat{\sigma}_w^2$. Similarly, the first diagram on the third row is given by (167) with $c(\omega) \to Y^{(0)}(\omega)$; note that there is no possibility to attach branches, since the only explicit 5-pt vertex is already exhausted. Again, we are including both possible orientations for these three diagrams. Lastly, we have the four diagrams in fig. 4 with $c(\omega) \to Y(\omega)$ and no further branching possibilities, which we have already given in (171), (172), and (175). Collecting results, we thus obtain the following recursive expression for

$Y(\omega)$:

$$
\begin{aligned}
Y = {} & c + \hat\sigma_w^2 f\bar f\, Y - 2\frac{\gamma T}{N}\sigma_w^2 Y_0^{(1)} f\bar f\, Y^{(0)} + \frac{\gamma T}{N}\frac{\hat\sigma^2}{2} f\bar f\, Y^{(0)} - \frac{\gamma T}{N}\frac{\sigma_w^2}{2\gamma} Y^{(0)2} \\
& + 4\frac{\gamma T}{N}\sigma_w^4 f\bar f\left\{\frac{1}{3} f\bar f\, Y^{(0)*3} + 2 b_0^2 f\bar f\, Y^{(0)} + Y^{(0)}\left[Y^{(0)} * Y^{(0)} * \left(f\bar f\right)\right]\right\},
\end{aligned}
\tag{178}
$$

which is readily solved, at least formally, to yield

$$
\begin{aligned}
Y(\omega) = {} & \frac{f\bar f}{1 - \hat\sigma_w^2 f\bar f}\left\{\frac{c}{f\bar f} + \frac{\gamma T}{N} Y^{(0)}\left(\frac{\hat\sigma^2}{2} - 2\sigma_w^2 Y_0^{(1)} - \frac{\sigma_w^2}{2\gamma f\bar f} Y^{(0)}\right)\right. \\
& \left. + 4\frac{\gamma T}{N}\sigma_w^4 f\bar f\left[\frac{1}{3} Y^{(0)*3} + 2 b_0^2 Y^{(0)} + \frac{1}{f\bar f} Y^{(0)}\left(Y^{(0)} * (f\bar f)\right)\right]\right\},
\end{aligned}
\tag{179}
$$

where for compactness we have suppressed the $\omega$ dependence, it being understood that these expressions are written in momentum space. The expression (179) is the analogue of (155) in the presence of 5-pt ($h^4$) interactions arising from the subleading term in the Taylor expansion of the nonlinearity, cf. (123), and can likewise be solved order by order in $\gamma T/N$. At $\mathcal{O}(1)$, (179) reduces to

$$
Y^{(0)}(\omega) = \frac{1}{1 - \hat\sigma_w^2 f\bar f} c(\omega) = \frac{\omega^2 + \gamma^2}{\omega^2 + \gamma^2 - \hat\sigma_w^2} c(\omega),
\tag{180}
$$

which is precisely the $\mathcal{O}(1)$-corrected propagator obtained in (166), as expected.

At $\mathcal{O}(T/N)$, (179) becomes

$$
\begin{aligned}
Y^{(1)}(\omega) = {} & \frac{f\bar f}{1 - \hat\sigma_w^2 f\bar f}\left\{\frac{1}{2} Y^{(0)}\left(\mathring\sigma_w^2 - \frac{\sigma_w^2}{\gamma f\bar f} Y^{(0)}\right)\right. \\
& \left. + 4\sigma_w^4 f\bar f\left[\frac{1}{3} Y^{(0)*3} + 2 b_0^2 Y^{(0)} + \frac{1}{f\bar f} Y^{(0)}\left(Y^{(0)} * Y^{(0)} * (f\bar f)\right)\right]\right\},
\end{aligned}
\tag{181}
$$

with $Y^{(0)}$ replaced by (180), and where for compactness we have absorbed the pseudo-initial conditions into

$$
\mathring\sigma_w^2 := \hat\sigma_w^2 - 4\sigma_w^2 Y_0^{(1)} = \sigma_w^2(1 - 2Y_0^{(0)} - 4Y_0^{(1)}),
\tag{182}
$$

where $\hat\sigma_w^2$ was defined in (165). It then remains to compute $Y^{(0)} * Y^{(0)} * (f\bar f)$, $b_0^2$, and $Y^{(0)*3}$, which we do in appendix B; the results are given in (B.14), (B.15), and (B.17), respectively. The explicit expression for $Y^{(1)}$ that we obtain upon substituting these into (181) is however exceedingly lengthy, so we refrain from displaying it here. Nonetheless, we now have all the ingredients to obtain the loop-corrected propagator to linear order in $T/N$, cf. (162), which enables us to define an effective correlation length in the next subsubsection.

### 4.4.3 The loop-corrected correlation length

Having computed the loop-corrected propagator (162) with (180) and (181), including both the leading and subleading terms in the expansion (123) of the nonlinearity $\phi(h)$, we now wish to repeat the analysis in subsubsec. 4.3.3, in which we identified a loop-corrected correlation length by expanding in small $|\tau|$. That is, upon inverse Fourier transforming to

$$
Y(\tau) = Y^{(0)}(\tau) + \frac{\gamma T}{N} Y^{(1)}(\tau),
\tag{183}
$$

we again obtain an expression involving multiple different exponentials. Following the logic below (159), we therefore approximate the loop-corrected propagator for small $|\tau|$ as

$$
Y(\tau) \approx A e^{-|\tau|/\Xi} + B,
\tag{184}
$$

where the constants $A$, $B$ are given by

$$A := Y^{(0)}(0) + \frac{\gamma T}{N} Y^{(1)}(0) - B, \qquad B := \lim_{\tau \to \infty} \left[ Y^{(0)}(\tau) + \frac{\gamma T}{N} Y^{(1)}(\tau) \right], \qquad (185)$$

and the loop-corrected correlation length at small $|\tau|$ is

$$\Xi := \frac{A}{\partial_{|\tau|} Y^{(0)}(0) + \frac{\gamma T}{N} \partial_{|\tau|} Y^{(1)}(0)} = \frac{A}{\partial_{|\tau|} Y^{(0)}(0)}, \qquad (186)$$

where in the present case, one finds that $\partial_{|\tau|} Y^{(1)}(0) = 0$. Note that taking the limit $\tau \to \infty$ in $B$ simply isolates the constant part of the corresponding expression, since all exponentials are decaying. As above, the explicit expressions for the components of $Y$ on the right-hand sides of (185) and (186) are rather lengthy, but are given in appendix C.

As in the linear case, the correlation length receives corrections at both $\mathcal{O}(1)$ and $\mathcal{O}(T/N)$; see the discussion below (161). To the extent of our knowledge, this is the first instance in which these effects have been explicitly computed in the literature, though finite-width corrections have also been considered in a closely related context in [2, 7]. Again however, we do not observe any shift in the location of the critical point itself: by examining the explicit expression for $\Xi$ in appendix C, one sees that the only potential divergences occur at $\sigma_{w,\text{eff}}^2 = \sigma_w^2 (1 - 2c_0) = \gamma^2$ and $\hat{\sigma}_w^2 = \sigma_w^2 (1 - 2Y_0^{(0)}) = \gamma^2$ (cf. (126) and (165), respectively). Since $Y_0^{(0)}$ is $c_0$ plus a (positive) $\mathcal{O}(1)$ contribution, $\sigma_{w,\text{eff}}^2 > \hat{\sigma}_w^2$. Therefore, when $\hat{\sigma}_w^2 = \gamma^2$, $\sigma_{w,\text{eff}}^2 > \gamma^2$, and we are no longer in the weak coupling regime. In other words, in the phase space parametrized by $(\sigma_w^2, \sigma_b^2)$, the line $\hat{\sigma}_w^2 = \gamma^2$ lies further to the right than the line $\sigma_{w,\text{eff}}^2 = \gamma^2$, past which the expression for the two-point function becomes complex.[45] Note that, as we discussed in subsec. 4.3.1, this does not imply that the true critical point does not exhibit a shift (and indeed, empirical observations demonstrate that there are sizable corrections to the result from the CLT; see, e.g., [17, 18]), but simply that the critical point happens to lie at the edge of the strong coupling regime at which the present perturbative treatment breaks down. It would be very interesting to understand this more thoroughly, and to explore whether this approach can be further improved to go beyond this limitation.

## 5 Discussion

In this work, we have explicitly constructed the quantum (statistical) field theory corresponding to the general class of networks described by (1), which includes both RNNs and deep MLPs. Relative to previous works on the nascent NN-QFT correspondence that take a phenomenological perspective, we have pursued a more fundamental or bottom-up approach, using well-known methods in statistical field theory to derive the partition function for an ensemble of random networks. This has led us to several intriguing parallels with well-studied $O(N)$ vector models and the perturbative expansion in Yang-Mills/QCD, and it would be interesting to study these formal similarities in more detail. For convenience, we have summarized the elements of the *NN-QFT dictionary* in appendix A.

At a physical level, the interpretation of the $\mathcal{O}(T/N)$ corrections is fairly clear. As expected on general grounds, and explored quantitatively in [2, 5, 7], the Gaussian ($N \to \infty$) limit does not suffice to describe real-world networks at finite width, whose distributions (i.e., actions) include effective interaction terms corresponding to higher cumulants. These finite-width effects are of both theoretical and practical relevance, and the depth-to-width ratio $T/N$ was

---

[45]As for the denominator, any zeros would require $\hat{\sigma}_w^2 = \sigma_{w,\text{eff}}^2$, in which case the equation is invalid, since this would imply the existence of degenerate poles in the momentum space expression.

identified as an important emergent scale in [2]. Note however that, as mentioned in sec. 4.4, we can in fact have corrections at any order in $T^m/N^n$ with $0 \leq m \leq n$. While those with $m = n$ – such as the $\mathcal{O}(T/N)$ term that we have computed here – will dominate in the $N \to \infty$ limit, terms with $m \neq n$ can become comparably important in networks of finite size; for example, if $T \sim 10$ and $N \sim 100$, then $\mathcal{O}(T^3/N^3) \sim \mathcal{O}(T/N^2)$. Physically, the effect is the same: non-Gaussianities accumulate with increasing depth, and are suppressed with increasing width. However, this suggests that the precise interplay between the two may be more subtle.[46]

Additionally, we have identified a dominant $\mathcal{O}(1)$ correction that persists even at infinite width. It is unclear whether this is implicitly included in the results from the central limit theorem (CLT) in [17,29] and related work, but regardless represents an important and novel contribution in the field-theoretic approach. While we have not proven this,[47] we expect – based on consistency with the CLT, i.e., the vanishing of interactions in the $N \to \infty$ limit – that the $\mathcal{O}(1)$ contribution does not alter the Gaussian nature of the distribution, and instead merely changes the variance from (126) to (165). Therefore, at a physical level, the $\mathcal{O}(1)$ contribution does not represent *bona fide* interactions per se, but rather quantifies fluctuations from typicality in the ensemble of networks.

To see this, note that since we are working in Euclidean signature, our action is proportional to $\beta \sim 1/\hbar$, where $\beta$ is the inverse temperature.[48] As in the $O(N)$ model, the 't Hooft coupling $\sigma_w \sim \hbar$ in order that the action be dimensionless, and thus higher-order terms in the perturbative expansion carry more factors of $\hbar$. In QFT in Lorentzian signature, this reflects the fact that loops represent quantum effects, with the order of the loop diagram in some sense quantifying the distance from classicality at which that effect arises. For QFT at finite temperature, i.e., statistical field theory, $\sigma_w \sim \beta^{-1}$, and the loops can be understood as thermal (statistical) fluctuations around the zero-temperature ($\sigma_w \to 0$) background. The 't Hooft coupling thus controls the size of the quantum/statistical fluctuations, so that the classical/MFT result is recovered in the limit when the weight initialization becomes deterministic. We emphasize again that this $\mathcal{O}(1)$ effect is present even when $T/N \to 0$, as can be seen from (161) or (186). That is, MFT is *not* obtained by simply taking $N \to \infty$, but rather by ignoring *all* fluctuations, including both genuine interactions (which disappear in the Gaussian limit, and are hence quantified by powers of $1/N$) and thermal fluctuations (which are extrinsic, i.e., independent of system size, and hence $\mathcal{O}(1)$).

As mentioned in the introduction, the location of the critical point predicted by the CLT (i.e., at $N \to \infty$) differs from empirical observations; see for example fig. 6 of [18], in which the theoretical result – derived via the CLT, following [17, 29] – lies at $(\sigma_b^2, \sigma_w^2) \approx (0.05, 1.76)$, whereas the empirical result for networks with $N = 784$ appears to lie at $(\sigma_b^2, \sigma_w^2) \approx (0.05, 1.55)$. However, as we remarked beneath (161) and (186), we are unable to observe a shift in the location of the critical point at either $\mathcal{O}(1)$ or $\mathcal{O}(T/N)$. In the linear case, (161) does not exhibit any new divergences, while in the nonlinear case, the new divergences in (186) lie to the right of the tree-level divergence, i.e., further into the strong coupling regime; see the discussion at the end of sec. 4.4. In principle, we see no reason that these finite-width corrections could not have shifted the edge of chaos to the left, into the weak coupling regime. In practice however, this may represent a limitation of the present perturbative approach.

There are many ways in which the first-principles approach pursued here could be further

---

[46]Note that since $T$ is dimensional, the depth of the network is measured in units of $\gamma$, cf. (169); e.g., an MLP has $\gamma T$ layers.

[47]An in principle straightforward albeit computationally prohibitive way to do this would be to compute all (or a convincing number of) higher-point correlators and show that they reduce to sums of products of two-point correlators as per Wick's theorem.

[48]Throughout this paper, we have followed the theoretical physics convention of working in civilized units, in which fundamental constants such as $\hbar$, $c$, $G_N$, etc. are unity.

explored and improved. For example, we have only computed the (quantum-corrected) two-point function $\langle h(t)h(s)\rangle$, whereas in principle there is no obstruction to computing higher-point functions or more complicated observables of interest. This leads to the question of whether such a theory is renormalizable: we have shown that the infinite series of corrections to the two-point function converge at weak coupling only to linear order in $T/N$, and while we see no obvious obstruction to convergence at higher orders, we have not proven that this continues to hold. Should it fail, then we have an effective theory valid only for sufficiently low-point correlators, to finite order in perturbation theory. By analogy, general relativity provides an extremely useful effective theory of gravity in the weak coupling regime, but is non-renormalizable, and hence breaks down at sufficiently high energies; the relevant question is then whether the neural networks of interest happen to lie at the metaphorical black hole singularity.

A related question is whether nonperturbative effects might extend this analysis beyond the weak coupling regime, and in particular, whether these shift the edge of chaos or otherwise contribute to the correlation length. The potential presence of nonperturbative effects was examined recently in [56], which demonstrated that the exact output distributions for linear and ReLU networks with zero bias exhibit heavy tails. Of course, for networks with sufficiently small $N$ (such at those with $N = 1, 2, 5$ illustrated in fig. 1 therein), we expect that perturbation theory will cease to apply, since the network is no longer Gaussian in the first place. However, while Gaussianity is recovered in the $N \to \infty$ limit, one can see the appearance of heavy tails for larger values of $N$ (e.g., $N = 100$ with $T \sim 1$ in fig. 1 of [56]) that cannot be accounted for by an $\mathcal{O}(1)$ effect of the type we compute above, i.e., a simple shift in the variance (note that in the aforementioned figure, the finite-$N$ curves intersect the Gaussian in two locations). It would be interesting to study this in more detail, e.g., to quantify the minimum network size at which the perturbative approach breaks down, or to investigate whether other nonperturbative effects persist at large $N$.

Another interesting direction would be to study the renormalization group (RG) in this framework. Empirical evidence [29] suggests that the critical point $\sigma_{w,\text{eff}}^2 = \gamma^2$ is semi-stable: it is attractive in the ordered phase, and repulsive in the chaotic phase. As discussed at the beginning of sec. 4, infinitesimal fluctuations away from criticality in the latter will drive the system to a new, infra-red (IR) fixed point at some smaller value of the coupling. Investigating the RG flow of the theory, e.g., using the methods of [8], may lead to further insights about the phase behavior of the system, and an understanding of the associated beta function would provide a new and mathematically rigorous perspective on the notion of RG scale in the context of hierarchical learning (see for example [18] and references therein). Additionally, since networks tuned to lie at the edge of chaos exhibit marked advantages in training performance, the ability to automatically induce a flow to the critical point would have great practical utility, and potentially lead to significant improvements over current initialization schemes.

Of course, the utility of this approach must ultimately be borne out in experiments. For example, it would be interesting to explore the extent to which the $\mathcal{O}(1)$ correction we have highlighted might explain the large discrepancy between the experimentally observed correlation length and that predicted from the CLT, cf. [17, 18]. For nonlinear activation functions $\phi(h)$, one would like to quantify the relative importance of higher-order terms in the Taylor expansion vs. higher-order terms in perturbation theory. More generally, it is necessary to quantify the regime of $T$ and $N$ (or rather, $\gamma T/N$), not to mention the 't Hooft coupling $\sigma_{w,\text{eff}}^2$ and $\sigma_{b,\text{eff}}^2$,[49] for which the results give good agreement with experiment. Additionally, note

---

[49]Recall from the discussion below (167) that $\sigma_{b,\text{eff}}^2$ must be sufficiently small to control the constant bias term, and therefore the weak coupling regime may be thought of as a subregion of the 2d phase space parametrized by $(\sigma_w^2, \sigma_b^2)$. That we must limit ourselves to small bias is however not surprising, since large $\sigma_b^2$ results in a strong correlation between neurons, and hence places one deep into the ordered phase, cf. fig. 1 in [17].

that we implicitly ignored boundary effects except where it was necessary to regulate the divergent integrals over $\tau$ when computing mushroom diagrams. Strictly speaking, we should keep $T$ finite everywhere, but in convergent integrals this cutoff would be exponentially suppressed, and hence is hierarchically smaller than the $\mathcal{O}(T/N)$ effects we have isolated. A related point is our assumption of time-translation invariance; while we expect this to hold to good approximation in the bulk (hidden) layers, it may break-down for observables computed at the boundaries $t = 0$, $T$, cf. the relation to [2] discussed in sec. 1. While we have focused on what we believe to be the dominant bulk (hidden-layer) physics here, a more careful treatment of boundary terms may be required depending on the application. We hope to explore some of these empirical questions in follow-up work.

In closing, we offer this work as a first-principles contribution to the rapidly emerging NN-QFT correspondence [2, 5–9]. While there is still much to learn, we believe this and similar approaches from theoretical physics can shed light on the underlying mathematical principles of deep neural networks, and have the potential to provide practical guidance for the development of more sophisticated machine learning techniques.

## Acknowledgements

It is a pleasure to thank James Giammona, Jim Halverson, Anindita Maiti, Dan Roberts, Keegan Stoner, and Sho Yaida for comments on a draft of this manuscript, as well as Johanna Erdmenger, Boris Hanin, and Soon H. Lim for discussions. K.T.G. acknowledges financial support from the Deutsche Forschungsgemeinschaft (DFG, German Research Foundation) under Germany's Excellence Strategy through the Würzburg-Dresden Cluster of Excellence on Complexity and Topology in Quantum Matter ct.qmat (EXC 2147, project id 390858490), as well as the Hallwachs-Röntgen Postdoc Program of ct.qmat. K.T.G. has also received funding from the European Union's Horizon 2020 research and innovation programme under the Marie Skłodowska-Curie grant agreement No 101024967.

## A   The NN-QFT dictionary

As explained in the introduction, we have generally used the phrase "NN-QFT correspondence" broadly to refer to the general philosophy of using techniques and ideas from QFT to understand deep neural networks. However, we have gone beyond previous works in directly constructing a *bona fide* field theory, allowing us to precisely identify various elements on either side of the correspondence:[50]

| NN | QFT | comments |
|---|---|---|
| width | rank of internal symmetry group, $N$ | dimensionless |
| depth | dimensionless time, $\gamma T$ | $\gamma$ required to make dimensionless, cf. (169) |
| layer of activations | $N$-component field $h$ | cf. footnote 12 |
| $\sigma_w^2$ | 't Hooft coupling | cf. (92), weak coupling $\sigma_w^2 < \gamma^2$ |
| $\sigma_b^2$ | auxiliary coupling | cf. (94), (170); weak coupling $\sigma_b^4 T < \gamma^3$ (nonlinear models) |
| depth/width | perturbative expansion parameter, $\gamma T/N$ | dimensionless |
| statistical fluctuations | $\mathcal{O}(1)$ effect | leading perturbative correction, cf. sec. 5 |
| finite-width effects | $\mathcal{O}(\gamma T/N)$ effect | subleading perturbative correction, cf. sec. 5 |
| trainable depth scale | correlation length $\xi$ | cf. (137), $\xi \to \infty$ at criticality |

[50]Here, we use the term "layer" to refer to either a single layer of an MLP, or a single timestep of an RNN. "Width" then refers to the number of neurons $N$ per layer; see the comment below (3). Let us also reiterate that the basic tools used in our construction are not new, and can be traced at least back to the work by Sompolinsky [36, 37]; nonetheless, the correspondence has not been explicitly stated in these terms.

Let us add a few comments about this *NN-QFT dictionary*. First, as remarked in the main text, the perturbative analysis applies in the regime $T, N \to \infty$ with $T/N \ll 1$ fixed. This agrees with previous results [2] that the behavior of the network is controlled by the ratio of depth to width, rather than either of these parameters independently: if $N \to \infty$ with $T$ fixed, the network becomes a Gaussian process (i.e., we turn off interactions), but is effectively shallow; conversely, if $T \to \infty$ with $N$ fixed, we lose all control of the perturbative expansion (i.e., effective interactions become too strong). As emphasized in [2], networks of practical interest typically have small depth to width ratios, and therefore we expect real-world networks to fall within this perturbative regime. (Note that $T$ also serves as an IR regulator on otherwise divergent integrals, cf. footnote 39).

However, our analysis has uncovered an ambiguity in the "infinite-width limit" as presented in the literature, namely that previous works have often conflated MFT and the CLT. As mentioned in the introduction, and in more detail in footnote 37, these are generally inequivalent. In the present case for example, the MFT (i.e., tree-level) result is *not* recovered in the infinite-width limit, since the $\mathcal{O}(1)$ correction is independent of $N$ (i.e., intensive, as opposed to extensive). See sec. 5 for further discussion of the physical interpretation of the $\mathcal{O}(1)$ and $\mathcal{O}(T/N)$ contributions.

As an additional constraint on the validity of our analysis, we found that $\sigma_w^2$ (and, in the case of non-linear models, also $\sigma_b^2$) must be sufficiently small in order for the infinite series of Feynman diagrams appearing at each order in the perturbative expansion to converge, cf. (149) and (170). This is consistent with our observation that $\sigma_w^2$ plays the role of the 't Hooft coupling, since the latter controls the weak/strong regime in Yang-Mills theory: perturbative quantum field theory of the kind we explore here is only tractable in the weak coupling regime, i.e., $\sigma_w^2 < \gamma^2$. At a practical level, this implies that our perturbative analysis can only be applied to the left of the critical point in $\sigma_w^2$, and breaks down precisely when $\sigma_w^2 = \gamma^2$. For non-linear models, there is an additional constraint that $\sigma_b^2$ be sufficiently small to avoid an IR divergence, but this is less restrictive, since networks of practical interest typically have $\sigma_b^2 \ll \gamma^2$.

Finally, we emphasize that the continuum limit is taken in the layer index, not the neural index, as explained in footnote 12. There is no meaningful notion of distance within a given layer. However, there *is* a meaningful notion of distance between different layers, since the networks under study do not admit skip connections; i.e., layer 1 is "closer" to layer 2 than it is to layer 3, and any effects of layer 1 on layer 3 are mediated via layer 2. At a technical level, this is the reason that $T$ has units of length in our analysis, while $N$ remains dimensionless, hence the appearance of $\gamma$ in the dimensionless expansion parameter $\gamma T/N$ (in contrast, both $L \equiv T$ and $N$ are dimensionless in [2], no continuum limit was taken, and there is no field theory). Physically, this gives rise to the correspondence between a single layer of $N$ neurons in the network, and an $N$-component field in the QFT. Hence we have (0+1)-dimensional field theory, consistent with the dimensional analysis in (169). The interaction between components as the field evolves in time (i.e., in subsequent layers) is ultimately responsible for the $N$ free neurons running in the internal loops of the Feynman diagrams discussed in sec. 4.

## A.1  Simplifying assumptions

Throughout this work, we have endeavored to be as explicit and as general as possible; however, we have made a number of technical assumptions along the way, most of which can be grouped into two broad categories:

1. Conditions which are required for our approach to hold on mathematical grounds (e.g., for convergence), and

2. Assumptions which are not strictly necessary, but which facilitate obtaining closed form expressions.

The first category includes conditions which are necessary for our approach, based on perturbative quantum field theory, to hold in general. Perhaps the most central is that $T$ and $N$ are both large, with $\gamma T/N \ll 1$ held fixed, since this is the dimensionless parameter that controls the perturbative expansion. Fortunately, as we have emphasized in the main text as well as when discussing the NN-QFT dictionary above, this is the regime of most modern deep neural networks used in practice; see also [2]. The second most important condition is that of weak coupling: as mentioned in the introduction, $\sigma_w^2$ plays the role of the 't Hooft coupling, and hence we expect perturbative QFT of the kind we explore here to only be valid in the weak-coupling regime. This expectation was borne out in (149), i.e., $\sigma_{w,\text{eff}}^2 < \gamma^2$ in general. Furthermore, we discovered that for non-linear models, the weak-coupling regime must be further specified by a constraint on $\sigma_b^2$, so that it describes a subset of the 2d phase space parametrized by $(\sigma_w^2, \sigma_b^2)$. This can be traced back to (167), which naïvely appears to contribute at $\mathcal{O}(T^2/N)$. However, the ensuing dimensional analysis reveals that this diagram is effectively $\mathcal{O}(T/N)$ provided that $\sigma_{b,\text{eff}}^4 T/\gamma^3 \ll 1$.

In (128), we have restricted to the case where the stochastic function $g(\tau)$ is constant, and absorbed this into $\kappa$. While this is not strictly necessary for the single-copy theory, it would be unusual for the stochasticity to depend on the current state of the system. In any case however, it is necessary for the double-copy analysis in sec. 3 that the stochasticity be common to both copies, which generically cannot hold if it depends on the states, cf. (55). If this were not the case, then the systems would deviate due to the different stochasticities even in the ostensibly ordered phase.

Turning now to the second category above, let us briefly summarize the various simplifying assumptions here, in roughly chronological order:

- In (15), we have chosen to work with (Gaussian) random networks largely for analytical tractability, since the Gaussian form of the initializations allows many integrals to be performed analytically. We note however that this class of networks is a standard ansatz in the literature, e.g., [17,29], and in any case the distributions will be Gaussian at large $N$ due to the CLT.

- To simplify (22) and related integrals, we have further taken the means of the initializations in (15) to be zero, again consistent with the cited literature.

- In (33), we chose the stochastic increments to also be i.i.d. Gaussian with zero mean. In principle however, one could consider other forms of stochasticity; see [51] and references therein.

- In solving for the Green functions, we assumed that the system exhibits time translation symmetry, cf. (40). While this seems like a strong assumption, we expect it to hold to good approximation in the bulk of the network (i.e., away from the boundaries), where our analysis is concerned; see sec. 1, in particular footnote 2.

- In the perturbative analysis in sec. 4, we have set $A = B = 0$ in order to focus on the core elements of the theory. As seen previously, these can in principle be treated analogously to $W$ and $U$.

- We have set the vevs in (102) to zero, as this is the simplest solution to the equations of motion. However, other, more complicated vacua may exist; we hope to explore this in future work. Similarly, we have chosen the data to also have zero vev, though this is an external parameter that one can fix based on the dataset in question.

- In the course of solving for the propagators, we have taken $\mathfrak{U}_0$ to be constant, cf. footnote 36. This has the effect of modifying $\sigma_b^2$ to $\sigma_{b,\text{eff}}^2$. An equally simple alternative would be to set $\tilde{\mathfrak{U}}_0$ constant, which would modify $\kappa$, cf. (130).

Another important restriction on the generality of our perturbative treatment is that we have restricted to tanh as our activation function. As discussed in [2] (see in particular sec. 2.2), in order to achieve criticality, we desire an activation function which is smooth, and which passes through the origin, i.e., $\phi(0) = 0$. While there are other activation functions which posses this property (e.g., sin, SWISH, GELU), none are as widely used as tanh; insofar as the latter captures the key properties we wish to explore, we have limited ourselves to $\phi(h) = \tanh(h)$ for concreteness. This however is not strictly necessary, and it would be interesting to consider other activation functions in this context. An alternative route would be to simply work with a generic Taylor expansion[51]

$$\phi(h) = h + h^2 \phi^{(2)}(0) + h^3 \phi^{(3)}(0) + \dots, \tag{A.1}$$

where we have taken $\phi^{(0)}(0) = 0$ for reasons just explained, and rescaled so that the coefficient of the linear term is unity. The case considered herein then corresponds to taking $\phi^{(2)}(0) = 0$ and $\phi^{(3)}(0) = -1/3$, and dropping all higher powers of $h$. As discussed below (139), these are negligible for activation functions of this type.

## B  Convolution identities

In this appendix, we evaluate the convolutions that appear in the perturbative (finite-width) corrections to the correlation function in subsec. 4.4.2. Specifically, to evaluate (181), we require $Y^{(0)} * Y^{(0)} * (f\bar{f})(\omega)$, $b_0^2$, and $Y^{(0)*3}$.

Let us introduce the convenient shorthand notation

$$\mathcal{L}_x(\omega) := \frac{1}{\omega^2 + 1/x^2} \qquad \Longrightarrow \qquad \mathcal{F}^{-1}(\mathcal{L}_x) = \frac{x}{2} e^{-|\tau|/x}, \tag{B.1}$$

where $\mathcal{F}^{-1}$ is the inverse Fourier transform; for compactness, we will often suppress the argument of $\mathcal{L}_x$, it being understood that this function acts on momentum space. We now begin by collecting a few useful identities that will enable us to straightforwardly evaluate the rather complicated expressions for the various convolutions in the main text above. First, consider the convolution

$$(\mathcal{L}_x * \mathcal{L}_y)(\omega) = \mathcal{F}[\mathcal{F}^{-1}(\mathcal{L}_x)\mathcal{F}^{-1}(\mathcal{L}_y)] = \frac{xy}{4}\mathcal{F}\left[e^{-|\tau|/\left(\frac{xy}{x+y}\right)}\right] = \frac{x+y}{2}\mathcal{L}_{\frac{xy}{x+y}}(\omega), \tag{B.2}$$

which follows from the convolution theorem and (B.1). Next, consider the convolution of $\mathcal{L}_z$ with the product

$$\mathcal{L}_x \mathcal{L}_y(\omega) = \frac{x^2 y^2}{x^2 - y^2}\left(\mathcal{L}_x - \mathcal{L}_y\right)(\omega), \tag{B.3}$$

which is a trivial application of (B.2):

$$\mathcal{L}_z * (\mathcal{L}_x \mathcal{L}_y)(\omega) = \frac{1}{2}\frac{x^2 y^2}{x^2 - y^2}\left[(x+z)\mathcal{L}_{\frac{xz}{x+z}} - (y+z)\mathcal{L}_{\frac{yz}{y+z}}\right](\omega). \tag{B.4}$$

Lastly, observe that convolving with $2\pi\delta(\omega)$ is the identity operation; that is, if $g(\omega)$ is an arbitrary smooth distribution on momentum space, then

$$g(\omega) * 2\pi\delta(\omega) = 2\pi \int \frac{d\omega'}{2\pi} g(\omega')\delta(\omega - \omega') = g(\omega), \tag{B.5}$$

where the $(2\pi)^{-1}$ in the measure is due to our Fourier conventions in (117). Note in particular that this formula applies for $g(\omega) = \delta(\omega)$.

---

[51]We thank Harold Erbin for this suggestion.

**Expressing $Y^{(0)}(\omega)$**

In the notation (B.1), the bare (tree-level/MFT) propagator $c(\omega)$ given in (130) may be written

$$c(\omega) = \kappa \mathcal{L}_{\xi_0} + 2\pi\delta(\omega)\xi_0^2\sigma_{b,\text{eff}}^2, \qquad \xi_0 := \frac{1}{\sqrt{\gamma^2 - \sigma_{w,\text{eff}}^2}}, \tag{B.6}$$

where as in the main text we have taken $\sigma_{b,\text{eff}}^2$ to be time-independent. The $\mathcal{O}(1)$-corrected propagator given in (180) is then

$$\begin{aligned}
Y^{(0)}(\omega) &= \left(1 + \frac{\hat{\sigma}_w^2 f\bar{f}}{1 - \hat{\sigma}_w^2 f\bar{f}}\right)c(\omega) = \left(1 + \hat{\sigma}_w^2\mathcal{L}_{\xi_1}\right)c(\omega) \\
&= \kappa\mathcal{L}_{\xi_0} + \kappa\hat{\sigma}_w^2\mathcal{L}_{\xi_0}\mathcal{L}_{\xi_1} + 2\pi\delta(\omega)\xi_0^2(1 + \hat{\sigma}_w^2\mathcal{L}_{\xi_1})\sigma_{b,\text{eff}}^2,
\end{aligned} \tag{B.7}$$

where we have defined

$$\xi_1 := \frac{1}{\sqrt{\gamma^2 - \hat{\sigma}_w^2}}. \tag{B.8}$$

Observing that $\delta(\omega)\mathcal{L}_{\xi_1} = \xi_1^2$ and applying (B.3), we may express this as

$$\begin{aligned}
Y^{(0)}(\omega) &= \kappa\left(1 + \frac{\hat{\sigma}_w^2}{\sigma_{w,\text{eff}}^2 - \hat{\sigma}_w^2}\right)\mathcal{L}_{\xi_0} - \frac{\kappa\hat{\sigma}_w^2}{\sigma_{w,\text{eff}}^2 - \hat{\sigma}_w^2}\mathcal{L}_{\xi_1} + 2\pi\delta(\omega)\xi_0^2(1 + \xi_1^2\hat{\sigma}_w^2)\sigma_{b,\text{eff}}^2 \\
&=: a_0\mathcal{L}_{\xi_0} + a_1\mathcal{L}_{\xi_1} + 2\pi a_\delta\delta(\omega),
\end{aligned} \tag{B.9}$$

where we have used the fact that

$$\frac{\xi_0^2\xi_1^2}{\xi_0^2 - \xi_1^2} = \frac{1}{\sigma_{w,\text{eff}}^2 - \hat{\sigma}_w^2}. \tag{B.10}$$

On the second line of (B.9), we have defined the coefficients $a_x$ for later convenience.

**Evaluating $(Y^{(0)} * Y^{(0)})(\omega)$**

With the above expressions in hand, consider the convolution

$$\begin{aligned}
(Y^{(0)} * Y^{(0)})(\omega) &= a_0^2(\mathcal{L}_{\xi_0} * \mathcal{L}_{\xi_0})(\omega) + a_1^2(\mathcal{L}_{\xi_1} * \mathcal{L}_{\xi_1})(\omega) + 2a_0a_1(\mathcal{L}_{\xi_0} * \mathcal{L}_{\xi_1})(\omega) \\
&\quad + 4\pi a_0a_\delta(\mathcal{L}_{\xi_0} * \delta)(\omega) + 4\pi a_1a_\delta(\mathcal{L}_{\xi_1} * \delta)(\omega) + (2\pi a_\delta)^2(\delta * \delta)(\omega),
\end{aligned} \tag{B.11}$$

with the coefficients $a_x$ defined in (B.9). Applying (B.2) and (B.5), this becomes

$$\begin{aligned}
(Y^{(0)} * Y^{(0)})(\omega) &= a_0^2\xi_0\,\mathcal{L}_{\xi_0/2} + a_1^2\xi_1\,\mathcal{L}_{\xi_1/2} + a_0a_1(\xi_0 + \xi_1)\mathcal{L}_{\frac{\xi_0\xi_1}{\xi_0+\xi_1}} \\
&\quad + 2a_0a_\delta\,\mathcal{L}_{\xi_0} + 2a_1a_\delta\,\mathcal{L}_{\xi_1} + 2\pi a_\delta^2\,\delta(\omega).
\end{aligned} \tag{B.12}$$

We can now use this expression to evaluate the three desired convolutions mentioned at the beginning of this appendix.

**Evaluating $(Y^{(0)} * Y^{(0)} * \mathcal{L}_{1/\gamma})(\omega)$**

In the notation (B.1), $f(\omega)\bar{f}(\omega) = \mathcal{L}_{1/\gamma}$, cf. (151). Hence, to compute $Y^{(0)} * Y^{(0)} * (f\bar{f})$ in momentum space, we simply convolve each term in (B.12) with $\mathcal{L}_{1/\gamma}$. By (B.2), this will result in convolutions of the form

$$(\mathcal{L}_x * \mathcal{L}_{1/\gamma})(\omega) = \frac{1}{2}(x + 1/\gamma)\mathcal{L}_{\frac{x}{\gamma x+1}}. \tag{B.13}$$

We then immediately obtain

$$
\begin{aligned}
(Y^{(0)} * Y^{(0)} * \mathcal{L}_{1/\gamma})(\omega) &= \frac{\xi_0}{4}(\xi_0 + 2/\gamma)a_0^2 \mathcal{L}_{\frac{\xi_0}{\gamma\xi_0+2}} + \frac{\xi_1}{4}(\xi_1 + 2/\gamma)a_1^2 \mathcal{L}_{\frac{\xi_1}{\gamma\xi_1+2}} \\
&+ (\xi_0 + 1/\gamma)a_0 a_\delta \mathcal{L}_{\frac{\xi_0}{\gamma\xi_0+1}} + (\xi_1 + 1/\gamma)a_1 a_\delta \mathcal{L}_{\frac{\xi_1}{\gamma\xi_1+1}} \\
&+ \frac{1}{2\gamma}(\gamma\xi_0\xi_1 + \xi_0 + \xi_1)a_0 a_1 \mathcal{L}_{\frac{\xi_0\xi_1}{\gamma\xi_0\xi_1+\xi_0+\xi_1}} + a_\delta^2 \mathcal{L}_{1/\gamma}.
\end{aligned}
\tag{B.14}
$$

### Evaluating $b_0^2$

Recall from the discussion above (172) that $b_0^2$ refers to the non-exponential part of $(Y^{(0)} * Y^{(0)})(\omega)$, evaluated at $\tau = 0$. That is, we keep only the term in $\mathcal{F}^{-1}(Y^{(0)} * Y^{(0)})(\tau)$ that requires regulating by imposing a cutoff as in footnote 39. From the form of (B.12) and the inverse Fourier transform (B.1), we see that the only term that thus contributes is proportional to $\delta(\omega)$, hence:

$$
b_0^2 = \mathcal{F}^{-1}\left[2\pi a_\delta^2 \delta(\omega)\right]\big|_{\tau=0} = \xi_0^4 \sigma_{b,\text{eff}}^4 (1 + \hat{\sigma}_w^2 \xi_1^2)^2,
\tag{B.15}
$$

where we have substituted in the coefficient $a_\delta$ defined in (B.9).

### Evaluating $Y^{(0)*3}(\omega)$

Finally, to evaluate $(Y^{(0)} * Y^{(0)} * Y^{(0)})(\omega)$, we must convolve (B.12) with (B.9):

$$
\begin{aligned}
Y^{(0)*3}(\omega) &= \xi_0 a_0^3 \mathcal{L}_{\xi_0/2} * \mathcal{L}_{\xi_0} + \xi_1 a_1^3 \mathcal{L}_{\xi_1/2} * \mathcal{L}_{\xi_1} + \xi_0 a_0^2 a_1 \mathcal{L}_{\xi_0/2} * \mathcal{L}_{\xi_1} + \xi_1 a_0 a_1^2 \mathcal{L}_{\xi_1/2} * \mathcal{L}_{\xi_0} \\
&+ 2a_0^2 a_\delta \mathcal{L}_{\xi_0} * \mathcal{L}_{\xi_0} + 4a_0 a_1 a_\delta \mathcal{L}_{\xi_0} * \mathcal{L}_{\xi_1} + 2a_1^2 a_\delta \mathcal{L}_{\xi_1} * \mathcal{L}_{\xi_1} \\
&+ (\xi_0 + \xi_1)a_0^2 a_1 \mathcal{L}_{\xi_0} * \mathcal{L}_{\frac{\xi_0\xi_1}{\xi_0+\xi_1}} + (\xi_0 + \xi_1)a_0 a_1^2 \mathcal{L}_{\xi_1} * \mathcal{L}_{\frac{\xi_0\xi_1}{\xi_0+\xi_1}} \\
&+ 6\pi a_0 a_\delta^2 \mathcal{L}_{\xi_0} * \delta + 6\pi a_1 a_\delta^2 \mathcal{L}_{\xi_1} * \delta + 2\pi \xi_0 a_0^2 a_\delta \mathcal{L}_{\xi_0/2} * \delta + 2\pi \xi_1 a_1^2 a_\delta \mathcal{L}_{\xi_1/2} * \delta \\
&+ 2\pi(\xi_0 + \xi_1)a_0 a_1 a_\delta \mathcal{L}_{\frac{\xi_0\xi_1}{\xi_0+\xi_1}} * \delta + (2\pi)^2 a_\delta^3 \delta * \delta.
\end{aligned}
\tag{B.16}
$$

Applying (B.2) and (B.5) as before and collecting terms, this becomes

$$
\begin{aligned}
Y^{(0)*3}(\omega) &= \frac{3}{4}\xi_0^2 a_0^3 \mathcal{L}_{\xi_0/3} + \frac{3}{4}\xi_1^2 a_1^3 \mathcal{L}_{\xi_1/3} + 3\xi_0 a_0^2 a_\delta \mathcal{L}_{\xi_0/2} + 3\xi_1 a_1^2 a_\delta \mathcal{L}_{\xi_1/2} \\
&+ 3a_0 a_\delta^2 \mathcal{L}_{\xi_0} + 3a_1 a_\delta^2 \mathcal{L}_{\xi_1} + 3(\xi_0 + \xi_1)a_0 a_1 a_\delta \mathcal{L}_{\frac{\xi_0\xi_1}{\xi_0+\xi_1}} \\
&+ \frac{3}{4}\xi_0(\xi_0 + 2\xi_1)a_0^2 a_1 \mathcal{L}_{\frac{\xi_0\xi_1}{\xi_0+2\xi_1}} + \frac{3}{4}\xi_1(\xi_1 + 2\xi_0)a_0 a_1^2 \mathcal{L}_{\frac{\xi_0\xi_1}{\xi_1+2\xi_0}} + 2\pi a_\delta^3 \delta(\omega).
\end{aligned}
\tag{B.17}
$$

The convolutions (B.14), (B.15), and (B.17) can then be substituted into (181) to obtain an explicit expression for $Y^{(1)}(\omega)$.

## C   Explicit expressions for the loop-corrected propagator

In this appendix, we collect the explicit expressions for the various components of the loop-corrected propagator $Y(\tau)$ and associated correlation length $\Xi$ in the nonlinear case, in the small-$|\tau|$ approximation (184):

$$
Y(\tau) \approx A e^{-|\tau|/\Xi} + B,
\tag{C.1}
$$

where the constants $A, B$ were defined as

$$A := Y^{(0)}(0) + \frac{\gamma T}{N} Y^{(1)}(0) - B, \qquad B := \lim_{\tau \to \infty} \left[ Y^{(0)}(\tau) + \frac{\gamma T}{N} Y^{(1)}(\tau) \right], \qquad \text{(C.2)}$$

and the loop-corrected correlation length is

$$\Xi := \frac{A}{\partial_{|\tau|} Y^{(0)}(0)}, \qquad \text{(C.3)}$$

where $Y^{(0)}(\tau)$, $Y^{(1)}(\tau)$ are the inverse Fourier transforms of (180) and (181), respectively. The former is straightforward and relatively compact:

$$Y^{(0)}(\tau) = \frac{1}{2} \left( a_0 \xi_0 \, e^{-|\tau|/\xi_0} + a_1 \xi_1 \, e^{-|\tau|/\xi_1} \right) + a_\delta, \qquad \text{(C.4)}$$

where the coefficients $a_0$, $a_1$, and $a_\delta$ defined in (B.9) are

$$a_0 := \kappa \left( 1 + \frac{\hat{\sigma}_w^2}{\sigma_{w,\text{eff}}^2 - \hat{\sigma}_w^2} \right), \qquad a_1 := \kappa - a_0, \qquad a_\delta := \xi_0^2 (1 + \xi_1^2 \hat{\sigma}_w^2) \sigma_{b,\text{eff}}^2, \qquad \text{(C.5)}$$

and $\xi_0$, $\xi_1$ were defined in (137) and (B.8), respectively,

$$\xi_0 := \frac{1}{\sqrt{\gamma^2 - \sigma_{w,\text{eff}}^2}}, \qquad \xi_1 := \frac{1}{\sqrt{\gamma^2 - \hat{\sigma}_w^2}}, \qquad \text{(C.6)}$$

with $\hat{\sigma}_w^2 = \sigma_w^2 (1 - 2Y_0^{(0)}) < \sigma_w^2 (1 - 2c_0) = \sigma_{w,\text{eff}}^2$, cf. (165), (126). From (C.4), one readily obtains the various $Y^{(0)}$-dependent terms appearing on the right-hand side of (C.2) and (C.3). We note however that this is technically still only an implicit equation for $Y^{(0)}(\tau)$, since $\xi_0$, $\xi_1$ contain the pseudo-initial conditions $c_0$, $Y_0^{(0)}$, which must be obtained self-consistently by solving (C.4) for $\tau = 0$.

Unfortunately, $Y^{(1)}(\tau)$ is substantially more complicated, but can nonetheless be obtained by using the shorthand (B.1) and associated identities introduced in appendix B to write (181) as an expression linear in $\mathcal{L}_x(\omega)$, each instance of which can then be trivially inverse Fourier transformed to $\mathcal{F}^{-1}(\mathcal{L}_x) = \frac{x}{2} e^{-|\tau|/x}$. After a great deal of algebra, we find that the constant portion appearing in $B$ is

$$\begin{aligned}
\lim_{\tau \to \infty} Y^{(1)}(\tau) = \frac{1}{6} \xi_1^2 \Bigg( & 6\sigma_{b,\text{eff}}^2 \xi_0^2 \xi_1^2 \sigma_{w,\text{eff}}^2 \left( \gamma \left( \frac{a_0^2 \xi_0^3}{\gamma \xi_0 + 2} + \frac{2 a_0 a_1 \xi_0^2 \xi_1^2}{\gamma \xi_0 \xi_1 + \xi_0 + \xi_1} + \frac{a_1^2 \xi_1^3}{\gamma \xi_1 + 2} \right) \right. \\
& + 4 a_\delta \gamma \left( \frac{a_0 \xi_0^2}{\gamma \xi_0 + 1} + \frac{a_1 \xi_1^2}{\gamma \xi_1 + 1} \right) + 4 a_\delta^2 \Bigg) - 6 a_\delta \gamma \sigma_{w,\text{eff}}^2 \left( a_0 \xi_0^2 + a_1 \xi_1^2 \right) \\
& + \frac{56 a_\delta^3 \sigma_{w,\text{eff}}^2}{\gamma^2} - 3 a_\delta^2 \gamma T \sigma_{w,\text{eff}}^2 + 3 a_\delta \mu^2 \Bigg).
\end{aligned} \qquad \text{(C.7)}$$

Note that the extra factor of $T$ on the right-hand side appears multiplied by $\sigma_{b,\text{eff}}^2$ (via $a_\delta$), the smallness of which keeps the perturbative expansion under control; see the discussion below (167).

As for the value at $\tau = 0$ appearing in the coefficient $A$, let us express the five terms in the inverse Fourier transform of (181) separately. Noting that

$$\frac{f \bar{f}}{1 - \hat{\sigma}_w^2 f \bar{f}} = \mathcal{L}_{\xi_1}, \qquad f \bar{f} = \mathcal{L}_{1/\gamma}, \qquad \text{(C.8)}$$

cf. (151), we find:

$$\frac{\hat{\sigma}_w^2}{2}\mathcal{F}^{-1}\big[\mathcal{L}_{\xi_1}Y^{(0)}(\omega)\big]\Big|_{\tau=0} = \frac{\hat{\sigma}_w^2}{8}\xi_1^2\bigg(\frac{2a_0\xi_0^2}{\xi_0+\xi_1}+a_1\xi_1+4a_\delta\bigg), \tag{C.9a}$$

$$-\frac{\sigma_w^2}{2\gamma}\mathcal{F}^{-1}\big[\mathcal{L}_{1/\gamma}\mathcal{L}_{\xi_1}Y^{(0)}(\omega)^2\big]\Big|_{\tau=0} =$$
$$-\frac{\sigma_w^2}{32\gamma(\xi_0+\xi_1)^2}\big(4a_0^2\xi_0^3\big(\xi_0^2\big(\xi_1^2\hat{\sigma}_w^2+1\big)+2\xi_0\big(\xi_1^3\hat{\sigma}_w^2+\xi_1\big)+\xi_1^2\big)$$
$$+32a_\delta(\xi_0+\xi_1)^2\big(\xi_1^2\hat{\sigma}_w^2+1\big)\big(a_0\xi_0^2+a_1\xi_1^2\big)+8a_0a_1\xi_0^2\xi_1^2\big(2\xi_0\xi_1^2\hat{\sigma}_w^2+2\xi_0+\xi_1^3\hat{\sigma}_w^2+2\xi_1\big)$$
$$+a_1^2\xi_1^3(\xi_0+\xi_1)^2\big(3\xi_1^2\hat{\sigma}_w^2+4\big)+16a_\delta^2 T(\xi_0+\xi_1)^2\big(\xi_1^2\hat{\sigma}_w^2+1\big)\big), \tag{C.9b}$$

$$\frac{4}{3}\sigma_w^4\mathcal{F}^{-1}\big[\mathcal{L}_{1/\gamma}\mathcal{L}_{\xi_1}Y^{(0)*3}(\omega)\big]\Big|_{\tau=0} = \frac{\sigma_w^2}{48}\bigg(\frac{8a_0^3\xi_1^2\xi_0^7}{\big(\gamma^2\xi_0^2-9\big)\big(\xi_0^2-9\xi_1^2\big)}+\frac{48a_0^2a_\delta\xi_1^2\xi_0^6}{\big(\gamma^2\xi_0^2-4\big)\big(\xi_0^2-4\xi_1^2\big)}$$

$$-\frac{6a_0^2a_1\xi_1^4\xi_0^6}{(\xi_0+\xi_1)\big(\big(\gamma^2\xi_1^2-1\big)\xi_0^2-4\xi_1\xi_0-4\xi_1^2\big)}-\frac{24a_0a_1^2\xi_1^6\xi_0^5}{(\xi_0+\xi_1)(3\xi_0+\xi_1)\big(\big(\gamma^2\xi_1^2-4\big)\xi_0^2-4\xi_1\xi_0-\xi_1^2\big)}$$

$$-\frac{96a_0a_1a_\delta\xi_1^4\xi_0^5}{(2\xi_0+\xi_1)\big(\big(\gamma^2\xi_1^2-1\big)\xi_0^2-2\xi_1\xi_0-\xi_1^2\big)}-\frac{96a_0a_\delta^2\xi_0}{\big(\frac{1}{\xi_0^2}-\gamma^2\big)\big(\frac{1}{\xi_1^2}-\frac{1}{\xi_0^2}\big)}+\frac{64a_\delta^3\xi_1^2}{\gamma^2}$$

$$+\frac{24}{\gamma\big(\gamma^2-\frac{1}{\xi_1^2}\big)^2}\bigg(\frac{\xi_1^2\big(\gamma^2\xi_1^2-1\big)a_1^3}{\gamma^2\xi_1^2-9}$$

$$+\frac{\xi_1\big(\gamma^2\xi_1^2-1\big)\big(a_0(2\xi_0+\xi_1)\big(\gamma^2\xi_1^2-4\big)\xi_0^2+4a_\delta\big(\big(\gamma^2\xi_1^2-4\big)\xi_0^2-4\xi_1\xi_0-\xi_1^2\big)\big)a_1^2}{\big(\gamma^2\xi_1^2-4\big)\big(\big(\gamma^2\xi_1^2-4\big)\xi_0^2-4\xi_1\xi_0-\xi_1^2\big)}$$

$$+\frac{1}{\big(\gamma^6\xi_0^6-14\gamma^4\xi_0^4+49\gamma^2\xi_0^2-36\big)\xi_1^2}\big(a_0\xi_0^2\big(4\big(\gamma^4\xi_0^4-13\gamma^2\xi_0^2+36\big)a_\delta^2$$

$$+4a_0\xi_0\big(\gamma^4\xi_0^4-10\gamma^2\xi_0^2+9\big)a_\delta+a_0^2\xi_0^2\big(\gamma^4\xi_0^4-5\gamma^2\xi_0^2+4\big)\big)\big(\gamma^2\xi_1^2-1\big)\big)$$

$$+\frac{a_1}{\big(\big(\gamma^2\xi_1^2-1\big)\xi_0^2-4\xi_1\xi_0-4\xi_1^2\big)\big(\big(\gamma^2\xi_1^2-1\big)\xi_0^2-2\xi_1\xi_0-\xi_1^2\big)}\big(a_0^2(\xi_0+2\xi_1)\big(\gamma^2\xi_1^2-1\big)$$

$$\times\big(\big(\gamma^2\xi_1^2-1\big)\xi_0^2-2\xi_1\xi_0-\xi_1^2\big)\xi_0^3$$

$$+4a_0a_\delta(\xi_0+\xi_1)\big(\gamma^2\xi_1^2-1\big)\big(\big(\gamma^2\xi_1^2-1\big)\xi_0^2-4\xi_1\xi_0-4\xi_1^2\big)\xi_0^2+4a_\delta^2\big(\big(\gamma^2\xi_1^2-1\big)^2\xi_0^4$$

$$+\big(6\xi_1-6\gamma^2\xi_1^3\big)\xi_0^3+\big(13\xi_1^2-5\gamma^2\xi_1^4\big)\xi_0^2+12\xi_1^3\xi_0+4\xi_1^4\big)\big)\big)$$

$$-\frac{\xi_1^4}{(\xi_0^2-\xi_1^2)(2\xi_0+\xi_1)(3\xi_0+\xi_1)\big(\xi_0^2-9\xi_1^2\big)\big(\xi_0^2-4\xi_1^2\big)\big(\gamma^2\xi_1^2-1\big)^2}\big(24a_0^3\xi_1\big(\gamma^2\xi_1^2-1\big)$$

$$\times\big(6\xi_0^6+5\xi_1\xi_0^5-29\xi_1^2\xi_0^4-25\xi_1^3\xi_0^3+19\xi_1^4\xi_0^2+20\xi_1^5\xi_0+4\xi_1^6\big)\xi_0^4$$

$$-6a_0^2\big(\gamma^2\xi_1^2-1\big)\big(6\xi_0^5-\xi_1\xi_0^4-58\xi_1^2\xi_0^3+8\xi_1^3\xi_0^2+36\xi_1^4\xi_0+9\xi_1^5\big)$$

$$\times\big(a_1(\xi_0-2\xi_1)(\xi_0+2\xi_1)^2-16a_\delta\xi_1(\xi_0+\xi_1)\big)\xi_0^3$$

$$+24a_0\big(\gamma^2\xi_1^2-1\big)\big(\xi_0^4-13\xi_1^2\xi_0^2+36\xi_1^4\big)\big(4\xi_1\big(6\xi_0^2+5\xi_1\xi_0+\xi_1^2\big)a_\delta^2$$

$$-4a_1(\xi_0+\xi_1)^2\big(3\xi_0^2-2\xi_1\xi_0-\xi_1^2\big)a_\delta-a_1^2(\xi_0-\xi_1)\xi_1^2(2\xi_0+\xi_1)^2\big)\xi_0^2$$

$$+a_1\xi_1\big(6\xi_0^8+5\xi_1\xi_0^7-83\xi_1^2\xi_0^6-70\xi_1^3\xi_0^5+280\xi_1^4\xi_0^4+245\xi_1^5\xi_0^3-167\xi_1^6\xi_0^2-180\xi_1^7\xi_0-36\xi_1^8\big)$$

$$\times\big(96a_\delta^2-32a_1\xi_1\big(\gamma^2\xi_1^2-1\big)a_\delta-3a_1^2\xi_1^2\big(\gamma^2\xi_1^2-1\big)\big)+\frac{48a_1a_\delta^2\xi_1^5}{\gamma^2\xi_1^2-1}+\frac{16a_1^2a_\delta\xi_1^6}{4-\gamma^2\xi_1^2}+\frac{a_1^3\xi_1^7}{9-\gamma^2\xi_1^2}\bigg), \tag{C.9c}$$

$$8\sigma_w^2b_0^2\mathcal{F}^{-1}\big[\mathcal{L}_{1/\gamma}\mathcal{L}_{\xi_1}Y^{(0)}(\omega)\big]\Big|_{\tau=0} =$$
$$\frac{2}{\gamma^2}a_\delta^2\xi_1^2\sigma_w^2\bigg(\frac{2a_0\gamma\xi_0^2(\gamma\xi_0\xi_1+\xi_0+\xi_1)}{(\gamma\xi_0+1)(\gamma\xi_1+1)(\xi_0+\xi_1)}+\frac{a_1\gamma\xi_1^2(\gamma\xi_1+2)}{(\gamma\xi_1+1)^2}+4a_\delta\bigg), \tag{C.9d}$$

$$4\sigma_w^2 \mathcal{F}^{-1}\left[\mathcal{L}_{\xi_1} Y^{(0)}\big(Y^{(0)} * Y^{(0)} * \mathcal{L}_{1/\gamma}\big)(\omega)\right]\Big|_{\tau=0} =$$

$$\frac{\sigma_w^2}{4}\Bigg( \frac{2a_0^2\kappa\xi_1^2\big(\big(\gamma^2\xi_1^2 - \hat{\sigma}_w^2\xi_1^2 - 1\big)\xi_0^2 + 4\gamma\xi_1^2\xi_0 + 4\xi_1^2\big)\xi_0^6}{\gamma\big(\gamma^2\xi_0^2 + 4\gamma\xi_0 + 3\big)\big(\big(\gamma^2\xi_1^2 - 1\big)\xi_0^2 + 4\gamma\xi_1^2\xi_0 + 4\xi_1^2\big)^2}$$

$$+ \frac{4a_0a_1\kappa\xi_1^4\big(\big(\xi_1\gamma^2 + 2\gamma - \xi_1\hat{\sigma}_w^2\big)\xi_0^2 + 2(\gamma\xi_1 + 1)\xi_0 + \xi_1\big)\xi_0^4}{\gamma(\gamma\xi_0 + 1)^2(\gamma\xi_1 + 1)(\gamma\xi_1\xi_0 + \xi_0 + 2\xi_1)(\xi_1 + \xi_0(\gamma\xi_1 + 2))^2}$$

$$+ \frac{8a_0a_\delta\kappa\xi_1^2\big(\big(\gamma^2\xi_1^2 - \hat{\sigma}_w^2\xi_1^2 - 1\big)\xi_0^2 + 2\gamma\xi_1^2\xi_0 + \xi_1^2\big)\xi_0^4}{\gamma^2(\gamma\xi_0 + 2)\big(\big(\gamma^2\xi_1^2 - 1\big)\xi_0^2 + 2\gamma\xi_1^2\xi_0 + \xi_1^2\big)^2}$$

$$+ 2\kappa\xi_1^4\Bigg(-\frac{4\xi_1\big(\big(\hat{\sigma}_w^2 - \gamma^2\big)\xi_1^4 + \xi_1^2 + \xi_0^2\big(\big(\hat{\sigma}_w^2\xi_1^4 + \xi_1^2\big)\gamma^2 - 2\xi_1^2\hat{\sigma}_w^2 - 1\big)\big)a_\delta^2}{\big(\xi_0^2 - \xi_1^2\big)^2\big(\gamma^2\xi_1^2 - 1\big)^2}$$

$$- \frac{1}{\gamma(\gamma\xi_0 + 1)^2(\xi_0 - \xi_1)^2(\xi_0 + \xi_1)^2(\xi_1 + \xi_0(\gamma\xi_1 + 2))^2}$$

$$\times \big(2a_0a_1\xi_0^2(\gamma\xi_1\xi_0 + \xi_0 + \xi_1)\big(\big(\big(\hat{\sigma}_w^2\xi_1^3 + \xi_1\big)\gamma^2 + 2\big(\xi_1^2\hat{\sigma}_w^2 + 1\big)\gamma - \xi_1\hat{\sigma}_w^2\big)\xi_0^4$$

$$+ 2(\gamma\xi_1 + 1)\big(\xi_1^2\hat{\sigma}_w^2 + 1\big)\xi_0^3 + \big(-\big(\gamma^2 - 2\hat{\sigma}_w^2\big)\xi_1^3 - 2\gamma\xi_1^2 + \xi_1\big)\xi_0^2 - 2\xi_1^2(\gamma\xi_1 + 1)\xi_0 - \xi_1^3\big)\big)$$

$$- \frac{1}{\gamma\big(\xi_0^2 - \xi_1^2\big)^2\big(\big(\gamma^2\xi_1^2 - 1\big)\xi_0^2 + 4\gamma\xi_1^2\xi_0 + 4\xi_1^2\big)^2}\big(a_0^2\xi_0^3(\gamma\xi_0 + 2)\xi_1\big(\big(\big(\hat{\sigma}_w^2\xi_1^4 + \xi_1^2\big)\gamma^2 - 2\xi_1^2\hat{\sigma}_w^2 - 1\big)\xi_0^4$$

$$+ 4\gamma\xi_1^2\big(\xi_1^2\hat{\sigma}_w^2 + 1\big)\xi_0^3 + \big(5\xi_1^2 - \xi_1^4\big(\gamma^2 - 5\hat{\sigma}_w^2\big)\big)\xi_0^2 - 4\gamma\xi_1^4\xi_0 - 4\xi_1^4\big)$$

$$- \frac{1}{\gamma\big(\xi_0^2 - \xi_1^2\big)^2\big(\gamma^2\xi_1^2 + 4\gamma\xi_1 + 3\big)^2}\big(a_1^2\xi_1^2(\gamma\xi_1 + 2)\big(\xi_0^2\big(\big(\hat{\sigma}_w^2\xi_1^4 + \xi_1^2\big)\gamma^2$$

$$+ 4\big(\hat{\sigma}_w^2\xi_1^3 + \xi_1\big)\gamma + 2\xi_1^2\hat{\sigma}_w^2 + 3\big) - \xi_1^2\big(\gamma^2\xi_1^2 - \hat{\sigma}_w^2\xi_1^2 + 4\gamma\xi_1 + 3\big)\big)\big)$$

$$- \frac{4a_\delta}{\gamma^3(\gamma\xi_1 + 2)^2(\gamma\xi_1\xi_0 + \xi_0 + \xi_1)^2\big(\xi_0^2 - \xi_1^2\big)^2(\xi_1 + \xi_0(\gamma\xi_1 - 1))^2}$$

$$\times \big(a_0\gamma^2\xi_0^2(\gamma\xi_0 + 1)\xi_1\big(\big(\big(\hat{\sigma}_w^2\xi_1^4 + \xi_1^2\big)\gamma^2 - 2\xi_1^2\hat{\sigma}_w^2 - 1\big)\xi_0^4$$

$$+ 2\gamma\xi_1^2\big(\xi_1^2\hat{\sigma}_w^2 + 1\big)\xi_0^3 + \big(2\xi_1^2 - \xi_1^4\big(\gamma^2 - 2\hat{\sigma}_w^2\big)\big)\xi_0^2 - 2\gamma\xi_1^4\xi_0 - \xi_1^4\big)(\gamma\xi_1 + 2)^2$$

$$+ a_1(\gamma\xi_1 + 1)\big(\big(\gamma^2\xi_1^2 - 1\big)\xi_0^2 + 2\gamma\xi_1^2\xi_0 + \xi_1^2\big)^2\big(\big(\xi_0^2\big(\hat{\sigma}_w^2\xi_1^3 + \xi_1\big) - \xi_1^3\big)\gamma^2 + 2\big(\xi_0^2\big(\xi_1^2\hat{\sigma}_w^2 + 1\big) - \xi_1^2\big)\gamma$$

$$+ \xi_1\big(\xi_1^2 - \xi_0^2\big)\hat{\sigma}_w^2\big)\big)\xi_0^2 + \frac{4\sigma_{b,\text{eff}}^2}{\gamma^2}$$

$$+ \frac{4\sigma_{b,\text{eff}}^2}{\gamma^2}\Bigg(\xi_1^2\Bigg(4a_\delta^2 + 4\gamma\Bigg(\frac{a_0\xi_0^2}{\gamma\xi_0 + 1} + \frac{a_1\xi_1^2}{\gamma\xi_1 + 1}\Bigg)a_\delta + \gamma\Bigg(\frac{a_0^2\xi_0^3}{\gamma\xi_0 + 2} + \frac{2a_0a_1\xi_1^2\xi_0^2}{\gamma\xi_1\xi_0 + \xi_0 + \xi_1} + \frac{a_1^2\xi_1^3}{\gamma\xi_1 + 2}\Bigg)\Bigg)\Bigg)$$

$$\times \big(\xi_1^2\hat{\sigma}_w^2 + 1\big)\xi_0^2\Bigg)$$

$$+ \frac{8a_1a_\delta\kappa\xi_1^4\big(\xi_1\gamma^2 + 2\gamma - \xi_1\hat{\sigma}_w^2\big)\xi_0^2}{\gamma^3(\gamma\xi_1 + 2)^2\big((\gamma\xi_1\xi_0 + \xi_0)^2 - \xi_1^2\big)} + \frac{8a_\delta^2\kappa\xi_1^2\big(\gamma^2\xi_1^2 - \hat{\sigma}_w^2\xi_1^2 - 1\big)\xi_0^2}{\gamma\big(\gamma^2\xi_0^2 - 1\big)\big(\gamma^2\xi_1^2 - 1\big)^2}$$

$$+ \frac{2a_1^2\kappa\xi_1^6\big(\gamma^2\xi_1^2 - \hat{\sigma}_w^2\xi_1^2 + 4\gamma\xi_1 + 3\big)\xi_0^2}{\gamma\big(\gamma^2\xi_1^2 + 4\gamma\xi_1 + 3\big)^2\big(\xi_0^2(\gamma\xi_1 + 2)^2 - \xi_1^2\big)}$$

$$+ \frac{\kappa\xi_1^3}{\sigma_{w,\text{eff}}^2 - \hat{\sigma}_w^2}\Bigg(\frac{4a_\delta^2}{\frac{1}{\xi_1^2} - \gamma^2} + \frac{4}{\gamma^2}\xi_1\Bigg(-\frac{a_0\gamma(\gamma\xi_0 + 1)\xi_1\xi_0^2}{\big(\gamma^2\xi_1^2 - 1\big)\xi_0^2 + 2\gamma\xi_1^2\xi_0 + \xi_1^2} - \frac{\gamma\xi_1a_1 + a_1}{\gamma\xi_1 + 2}\Bigg)a_\delta$$

$$- \frac{1}{\gamma}\xi_1\Bigg(\frac{a_0^2(\gamma\xi_0 + 2)\xi_1\xi_0^3}{\big(\gamma^2\xi_1^2 - 1\big)\xi_0^2 + 4\gamma\xi_1^2\xi_0 + 4\xi_1^2} + \frac{2a_0a_1(\gamma\xi_1\xi_0 + \xi_0 + \xi_1)\xi_0^2}{(\gamma\xi_0 + 1)(\xi_1 + \xi_0(\gamma\xi_1 + 2))} + \frac{a_1^2\xi_1^2(\gamma\xi_1 + 2)}{\gamma^2\xi_1^2 + 4\gamma\xi_1 + 3}\Bigg)\Bigg)\hat{\sigma}_w^2$$

$$+ \frac{2\kappa\xi_0}{\big(\sigma_{w,\text{eff}}^2 - \hat{\sigma}_w^2\big)^2}\Bigg(\frac{\xi_0(\gamma\xi_0 + 2)a_0^2}{\gamma\big(\gamma^2\xi_0^2 + 4\gamma\xi_0 + 3\big)\xi_1^2} + \frac{2}{\gamma^2\xi_0}\Bigg(\frac{2a_\delta(\gamma\xi_0 + 1)}{(\gamma\xi_0 + 2)\xi_1^2} + \frac{a_1\gamma(\gamma\xi_1\xi_0 + \xi_0 + \xi_1)}{(\gamma\xi_1 + 1)(\gamma\xi_1\xi_0 + \xi_0 + 2\xi_1)}\Bigg)a_0$$

$$+ \frac{4a_\delta^2}{\big(\gamma^2\xi_0^2 - 1\big)\xi_1^2} + \frac{a_1^2\xi_1(\gamma\xi_1 + 2)}{\gamma\big(\xi_0^2(\gamma\xi_1 + 2)^2 - \xi_1^2\big)} + \frac{4a_1a_\delta(\gamma\xi_1 + 1)}{\gamma\big((\gamma\xi_1\xi_0 + \xi_0)^2 - \xi_1^2\big)}\Bigg)\big(\xi_0^2\big(\xi_1^2\hat{\sigma}_w^2 + 1\big) - \xi_1^2\big)\Bigg). \quad \text{(C.9e)}$$

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
