# Peer review of "The edge of chaos: quantum field theory and deep neural networks"

_SciPost Physics, doi:SciPost Phys. 12, 081 (2022)_

## Round 1 · Referee Report · Bryan Zaldivar · 2021-11-29

Report
The present manuscript contributes to the development of the emergent field of NN-QFT correspondence. The authors present a quite detailed analysis including the use of Feynman diagrams to compute the perturbative corrections to the Gaussian limit, and the corresponding study of criticality. The work is sound, well written, and well structured. That being said, before recommending it for publication, I suggest the authors to address the following comments.
+++++++++++++++++++++++++++++++++++
1. In the introduction, the authors state that for neural networks of infinite width, the representations do not evolve during gradient-based learning. The statement is supported by ref.[2]. However no detailed explanation is given in the manuscript, and I also didn't find a detailed explanation of it in [2]. I suggest the authors dedicate a brief discussion on why this is correct. The confusion comes from the following two points:
A. By theorem, neural networks with large enough width are proven to be able to approximate any function (given very mild assumptions). So the author's statement (or the equivalent one in [2]), taken as such, may be interpreted as contradicting this theorem, unless I have misinterpreted their statement (if this is the case, please rephrase the statement or elaborate more on it).
B. The authors seem to base their work on the observation that the output of infinite-width neural nets behave as Gaussian Processes -GP- (by the way, if this is the case, the authors should cite the original work by Neal, 1996). Actually this result is obtained for a particular realization of Bayesian neural networks, whose prior over parameters are Gaussians with zero mean. It is true that GP's are restrictive in the sense that not always they can describe well enough real-word data. However, one could consider other priors for the weights giving more expressive models, or even abandon weight space altogether and consider inference in function space instead, where the neural net outputs become Implicit Processes, which are obviously much more expressive than GP's. So again, the point about infinite networks having an inherent lack of expressiveness does not seem to be clear.
++++++++++++++++++++++++++++++++++++++++++++
2. About the study of criticality. The authors mention in the introduction that, at the end of the day, the location of the critical point does not change after considering corrections coming from the finiteness of the network. However I was not able to find a discussion about the reason for this unexpected result in the manuscript. I suggest the authors do an exercise, independent of the perturbation theory formalism, where one takes a small N network (even with 1 hidden layer) and computes its critical point. If this were possible and the critical point turns out to be different from the network with infinite N, it would signal a limitation with the perturbation theory formalism. The other possibility commented by the authors, about the correction appearing only at higher orders (not computed in this work), is also not intuitive. I suggest the authors elaborate more on these issues, which are key to their work.
++++++++++++++++++++++++++++++++++++++++++++++
3. When computing the ensemble average over networks -expr.(2.19)-, it is assumed that the trainable parameters are distributed as factorized Gaussians (this is explicitly shown in expr.2.15). In deep networks, however, it is known that the parameters may present strong dependencies, so that the assumed i.i.d condition may not be in general a realistic assumption. How would the ensemble average change if considering multivariate Gaussians for the parameters, with generic covariance matrices? Would it be possible to extract the same conclusions for this case?
+++++++++++++++++++++++++++++++++++++++++++++++
4. I understand that this work is a contribution to the ongoing effort to develop the NN-QFT correspondence, and it is based on several other previous works along similar ideas and methods. In order to make the correspondence easier to grasp for the reader, I suggest the authors to include, for example in the form of an appendix, the "dictionary" of the correspondence, where the elements of the statistical model, neural network etc, are given an intuitive interpretation in the language of QFT. Of course it doesn't need to be exhaustive, nor a perfect match, but it should reflect the current understanding of such correspondence as much as possible. The inclusion of such a dictionary would definitely contribute to the excellence criteria for publishing at this journal.
Author: Ro Jefferson on 2022-01-03 [id 2060]
(in reply to Report 2 by Harold Erbin on 2021-12-14)Please see the attached pdf for a detailed response to the comments and suggestions provided in attachment to Report 2 above.
Attachment:
response_to_feedback.pdf
Harold Erbin on 2022-02-03 [id 2154]
(in reply to Ro Jefferson on 2022-01-03 [id 2060])I would like to thank the authors for the very detailed answer and all the explanations. I find the additional appendix A and beginning of section 2, together with all the comments at different places, extremely useful.

---

## Round 1 · Referee Report · Harold Erbin · 2021-12-14

Strengths
1. Study from first-principle the NN-QFT correspondence, which is an important topic.
2. Computations are overall clear and detailed.
Weaknesses
1. The presentation is very unbalanced.
2. Many assumptions are introduced along the way without proper discussion.
3. There are no numerical tests to support the approach of the paper.
4. It is not clear how the results are connected to concrete applications.
Report
This paper aims at deriving the quantum field theory associated with a statistical ensemble of recurrent or fully connected neural networks from first principles. The idea of the paper is to start with the description of the ensemble in terms of a stochastic differential equation to which one can associate a path integral, and reinterpret it as a field theory. Using ideas from statistical physics, it is then possible to compute different parameters such as the Lyapunov exponent and correlation length in terms of the network parameters (such as the standard deviations of the weights and biases) to characterize the properties of the network.
This is an important topic in its infancy, and this paper deserves careful attention. However, the paper in its current form is difficult to follow and the computations do not seem as general as suggested in the abstract and introduction. It is also difficult to see what concrete applications can be. As a consequence, before recommending it for publication, I would recommend a major revision of the presentation.
Requested changes
See the attached report for detailed feedback and suggestions.

---

## Round 2 · Author Response

We thank both referees for their careful reviews, and for raising several clarifying points which we have addressed below.
Response to Report 1 by Dr. Bryan Zaldivar:
1A. We see how these two statements could appear in tension, but they are not actually in contradiction: while there may exist networks (that is, a particular state or set of states for a given infinite-width network) that can approximate any given function (per the referenced theorem), there are many more states that will not. The statement about non-evolving representations is essentially that if you don’t happen to start with the correct set of weights and biases that approximates the given function, you’ll never evolve to it. In other words, the referenced theorem is an abstract statement that infinite-width networks are universal approximators, but says nothing about the learning dynamics in practice; the statement of [2] is that these dynamics are in fact trivial (specifically, that the neural tangent kernel does not evolve). This issue is first mentioned on page 8 of [2], and discussed in more detail in sections 6.3.3, 10.1.2, and chapter 11 therein, as well as in the newly added ref. [16]. We have elaborated on this in the introduction in the hopes of resolving any confusion.
1B. In the context of the previous point, we are merely making the observation that the distributions will be Gaussian at infinite width, which follows directly from the central limit theorem (CLT) regardless of the initial (potentially non-Gaussian) choice of priors from which the parameters are drawn. We have added an explicit citation to the original thesis by Neal (previously included only implicitly via ref. [7]), as requested.
2. The exercise suggested by the reviewer is indeed sensible, but in fact has already been done in [17,18]. We have alluded to and cited this discrepancy between the infinite-width prediction and the empirical results at finite N in the introduction (around page 3), as well as at the beginning of section 4 (page 30), and the end of section 4.3.3 (page 50). As the reviewer points out however, this is a central point of our work, so we have added a new paragraph in the Discussion (page 61) where we discuss this in more detail, and explicitly acknowledge this limitation of the perturbative approach. We also agree with the reviewer that the remark at the end of section 4.3 about a possible shift in the location of the critical point appearing at higher orders is not intuitive, and have removed this sentence, instead mentioning this possibility in the context of the aforementioned new paragraph.
3. While the reviewer is certainly correct that complicated dependencies may arise under training, here we are concerned with networks at initialization; we have edited the sentence above (2.15) to more clearly reflect this. In principle, one could consider a multivariate Gaussian with non-vanishing covariances between different parameters, but this would lead to an intractable increase in the number of couplings and a corresponding explosion of possible Feynman diagrams; e.g., if one coupled only two parameters, one would have a bivariate measure as in (2.46), resulting in a new coupling. With 5 parameters, there would be 26 possible couplings.
4. We thank the reviewer for this excellent suggestion to improve the manuscript, and have added a new appendix (now appendix A) enumerating the various elements of the NN-QFT dictionary. We added sentences referring the reader to this dictionary in the Introduction as well as the beginning of the Discussion.
Response to Report 2 by Dr. Harold Erbin:
Here let us first respond to the general points under "weaknesses":
As the referee points out under "strengths", we have gone to great lengths to present each step of the construction as clearly as possible. While the result is a relatively long and detailed paper, we believe this is well-worth it for the pedagogical clarity and explicitness this achieves (e.g., for the purposes of future work). Nonetheless, in the context of the next point, we agree that it is not easy for the reader to keep track of various technical assumptions in relation to the big picture, and have addressed this in our detailed response to requested changes in the attachment (see in particular points 1 and 12). While the notion of balance is inherently subjective, we believe the result is improved.
We aspired to be as general as possible for as long as possible, which is why simplifying assumptions are introduced en route, rather than restricting ourselves to some less-specific class of models at the outset. We agree with the referee that some discussion of these assumptions in relation to the generality of the work would be an improvement, and have added a new section (appendix A.1) in which we treat each of these in detail. See also our response to this point in aforementioned attachment.
We certainly agree that this is an important next step, as we have discussed in section 5. As the referee points out however, this topic is in its infancy, and the purpose of this work is explicitly theoretical in aim, namely the construction of a direct correspondence between deep neural networks and quantum field theory. We hope to see (if not perform ourselves) thorough empirical explorations of this topic in the future, but such an investigation is beyond the scope of this initial work. We would also like to point out that the complementary approach [2] also presented no numerical tests in support of their derivations, being similarly focused on theoretical explorations. In the preface of [2], the authors claim that they have performed these tests privately, but have chosen not to show them for reasons explained therein; here, we have simply been explicit about the need for empirical tests in future work.
See previous point. However, we have mentioned the main practical benefit – namely, predicting the location of the critical point – in several places, including the introduction, section 4, and section 5 (where some directions for future work are also discussed). At a more general level however, our main goal is to further the fundamental theory of deep neural networks by leveraging powerful tools from theoretical physics, especially QFT, in the spirit of previous works we have cited in the NN-QFT correspondence.
We hope that you will kindly consider the resubmitted manuscript for publication in SciPost.
Sincerely yours, K. Grosvenor and R. Jefferson

---

## Round 2 · List of Changes

Please see the pdf attached with our reply for a detailed response to the feedback and suggestions provided in Dr. Erbin's attachment. For convenience, we have copy-pasted the original text of the latter to make the document self-contained.
Ro Jefferson on 2022-01-04 [id 2065]
Here is a complete list of changes made in version 2 of our paper:
Elaborated on statements that representations do not evolve in the Introduction (page 2).
Added a new paragraph in the Discussion (page 61) on the location of the critical point in the context of the proposed experiment raised by referee #1.
Modified the language used when introducing (2.15) to make clear that this is a statement about initialisation.
Added new appendix A on the NN-QFT dictionary, following the excellent suggestion by referee #1. We believe this has substantially improved the presentation.
Added new section A.1 with a detailed, point-by-point discussion of the various technical assumptions introduced in the course of our derivations, as well as further discussion on the general conditions for which such a perturbative analysis is valid (e.g., large T,N and small T/N).
Added statements referring the reader to the new appendix A in the Introduction, beginning of section 2, and Discussion.
Added additional discussion about the large T (and large N) limit in the Introduction, as well as strengthened comparison with previous work which also identified T/N as the perturbative parameter.
Reminded the reader of the large T regime when discussing the continuum limit in footnote 12.
Added footnote 2 elaborating on the issue of boundaries in the introduction.
Corrected typo in which the stochastic function g appeared to depend on the copies, when in fact it is treated as a common external parameter.
Added further discussion about the treatment of the external stochasticity g below (2.3) as well as in appendix A.
Corrected typo below what is now (2.2), in which we had mistakenly written \gamma=0 for MLPs, when in fact the correct condition is \gamma=1 as below (3.39).
Edited paragraph below (4.48) clarifying the validity of the Taylor expansion of \phi.
Elaborated on choice of activation function below (2.2) as well as in appendix A.1, where the generic Taylor expansion suggested by referee #2 is also mentioned. We have also directed the reader to the relevant section of [2] for a more detailed discussion of activation functions in this context.
Added more discussion about the weak-coupling regime of the 2d parameter space in appendix A.
Expanded on relation to [36] at the start of section 2.
Following the excellent suggestion of referee #2, we have added a new paragraph to the start of section 2 with a roadmap of the derivation therein, so that the reader can more easily follow the analysis with the help of this "big picture". Additionally, we have here attempted to assuage the reader that the length is due to our efforts at pedagogical clarity, so that they will not be discouraged by the high level of detail.
We have slightly streamlined some the algebraic manipulations by removing trivial steps from (2.22), (4.28), (4.31), and (4.43), and added a clarifying sentence below the last of these.
We have modified the first sentence in the last paragraph of page 3 to more clearly acknowledge previous works on the present SFT approach.
Added text below (2.1) justifying this as the most generic starting point for our analysis.
Following another helpful suggestion by referee #2, we have replaced (2.2) with what was previously (2.16). We have also significantly rewritten the text below (2.2) elaborating on the various elements of this expression and providing some intuition, as well as included the reduction to an MLP here for further clarity.
Elaborated on footnote 8 to avoid any confusion about the boundaries, as previously discussed in the introduction.
Further clarified the role of the constant \gamma below (2.2) and in appendix A.
Moved equation (2.16) to just below (2.15), as suggested by referee #2, and modified text below (2.16) and (2.17) accordingly.
Corrected N^2-local to bi-local when introducing (2.24)
Removed the potentially confusing remark "subject to working with the ensemble average" from the very last sentence in sec. 2.2.
Added footnote 18 clarifying the expectation value of the external data x.
Removed potentially confusing remark "pursuant to our self-averaging assumption" above (2.44), when introducing the definition of the bivariate Gaussian.
Added text between (2.45) and (2.46), clarifying that the latter is merely a rewriting of (2.44).
Provided more intuition for the double-copy in the first paragraph of section 3, as well as pointed the reader to previous works that employed a similar strategy.
Edited footnote 22 to avoid the potentially confusing use of the commutator.
Corrected typo in which the distance was mistakenly written d(\tau) rather than d(t_1,t_2) (and similarly for c(\tau) where appropriate, i.e., for cross-correlators), and expanded the shorthand d(t) --> d(t,t).
Changed second lightcone coordinate in (3.24) and below from T to u, to avoid overloading notation.
Added text below (3.31), clarifying that we are interested in bound states.
Completely rewrote the text between (3.37) and (3.39) to make the argument more clear, and corrected text below (3.33) to match. We have also directed the reader to [36] for a more in-depth discussion about the properties of the potential.
Changed "correction" to "interaction" in the second sentence of section 4.
Rephrased sentence below (4.76) about the appearance of \delta(0).
Added explicit indices on the arguments of f in (2.5).
Removed sentence at the end of sec. 4.3 about a shift in the location of the critical point occurring at higher orders.
Changed stochastic increment in (2.1) and elsewhere from dB to dS, to avoid confusion with unrelated B introduced in (2.2).
Clarified specification to m=n=1 at beginning of section 4.4.
Added explicit citation to Neal as requested by referee #1.
Added explanation of vanishing boundary term in footnote 19.
Elaborated on footnote 25.
Edited footnote 33, since the vanishing boundary term is not actually an assumption.
Modified text below (4.28) to match use of Fourier transform.
Elaborated on footnote 40.
Reformatted some long expressions to improve page layout.
Corrected the following minor typos: missing bias in (4.10), missing parenthesis on z(t) in (4.10), use of x rather than t or \omega in (4.26) & (4.27), missing escape character resulting in appearance of comma in (4.53), xi_i --> xi_t above (2.33), O(\eta) --> \mathcal(O)(\eta) below (3.22), "relative to (3.6)" --> "relative to (2.25)" above (4.1), \hat{c} --> X in (4.69), missing 1/N on variances in (2.15).

---

## Editorial Decision

published